# Counterfactual Memorization
# in Neural Language Models

**Chiyuan Zhang**
Google Research
chiyuan@google.com

**Daphne Ippolito**
Carnegie Mellon University
daphnei@cmu.edu

**Katherine Lee**
Google DeepMind
katherinelee@google.com

**Matthew Jagielski**
Google DeepMind
jagielski@google.com

**Florian Tramèr**
ETH Zürich
florian.tramer@inf.ethz.ch

**Nicholas Carlini**
Google DeepMind
ncarlini@google.com

## Abstract

Modern neural language models that are widely used in various NLP tasks risk memorizing sensitive information from their training data. Understanding this memorization is important in real world applications and also from a learning-theoretical perspective. An open question in previous studies of language model memorization is how to filter out "common" memorization. In fact, most memorization criteria strongly correlate with the number of occurrences in the training set, capturing memorized familiar phrases, public knowledge, templated texts, or other repeated data. We formulate a notion of *counterfactual memorization* which characterizes how a model's predictions change if a particular document is omitted during training. We identify and study counterfactually-memorized training examples in standard text datasets. We estimate the influence of each memorized training example on the validation set and on generated texts, showing how this can provide direct evidence of the source of memorization at test time.

## 1   Introduction

Modern neural language models (LMs) have achieved impressive results in generating high quality text [e.g. Brown et al., 2020, Zhang et al., 2022, Chowdhery et al., 2022, OpenAI, 2023] and have led to breakthroughs in many downstream natural language processing tasks [Devlin et al., 2019, Raffel et al., 2020b, Bommasani et al., 2021]. The paradigm of taking a single large-scale pre-trained model and fine-tuning it for many tasks motivates the study of these models' ability to *generalize* by avoiding *memorizing* their training data. Moreover, memorization of sensitive user information or copyrighted materials in the training data [Carlini et al., 2020, Vyas et al., 2023, Lee et al., 2023] leads to practical concerns in real world applications.

Previous work on memorization in neural language models demonstrated the ability to extract memorized training data, including sensitive data such as phone numbers and usernames [Carlini et al., 2020, Ziegler, 2021, Carlini et al., 2019, Henderson et al., 2017, Thakkar et al., 2020, Thomas et al., 2020]. One issue with these extraction attacks is that they primarily identify "common" and frequently occurring strings in the training set. For example, as shown in the analysis of Lee et al. [2021], near-duplicate training examples, which are very common in standard text corpora, account for a large majority of the memorized content. To filter out such commonly occurring strings from all memorized texts, previous work applied various heuristic rules to distinguish frequently-occurring sequences from memorization of isolated pieces of information.

37th Conference on Neural Information Processing Systems (NeurIPS 2023).

In this paper, we propose a principled causal perspective to disentangle memorization of common vs rare data, by directly tying a model's predictions to the presence or absence of individual training examples. We define *counterfactual memorization* as a measure of the change in a model's prediction when a particular example is excluded from the training set. Counterfactual memorization accounts for the commonality of an example as removing one instance of a text that is common across multiple documents will have a minor effect on the model's prediction on that text. The mathematical formulation of counterfactual memorization extends a prior definition of label memorization in classification models [Feldman, 2020] to the context of neural language modeling.

Formally, a training document $x$ is considered counterfactually memorized, when the language model predicts $x$ accurately *if and only if* the model was trained on $x$. This allows us to construct a procedure to quantitatively measure the memorization of isolated pieces of text, whose sole presence in the training dataset have a large effect on the model's predictions.

Following Feldman and Zhang [2020], we further extend this definition to *counterfactual influence*, which measures the influence of a memorized training text sample on another text example. Counterfactual influence allows us to trace the source of information for a model's predictions, by locating the training example(s) which significantly contributed to it. With these tools, we study memorization across several standard text datasets. Our main contributions are as follows:

1. We define counterfactual memorization in neural LMs which gives us a principled perspective to distinguish memorization of "rare" and "common" information in neural LMs (Section 3).
2. We estimate counterfactual memorization on several standard text datasets, and confirm that rare memorized examples exist in all of them. We study common patterns across memorized text and the memorization profiles of individual internet domains. (Section 4).
3. We identify an inverse correlation between number of duplicates and counterfactual memorization as compared with previous definitions of memorization (Section 5).
4. We extend the definition of counterfactual memorization to counterfactual *influence*, and study the impact of memorized examples on the test-time prediction of the validation set examples and generated examples (Section 6).

## 2   Related Work

Previous work analyzed the memorization of large language models on sensitive information (e.g. phone numbers) in the training data [Carlini et al., 2020, Ziegler, 2021] or synthetically injected "canaries" [Carlini et al., 2019, Henderson et al., 2017, Thakkar et al., 2020, Thomas et al., 2020]. However, not all the memorized texts are equally interesting — as confirmed in a later study [Lee et al., 2021], near-duplicated training examples are very common in standard text corpus, and those commonly occurring phrases contribute significantly to memorized texts. In order to distinguish "common" memorization of common phrases or public knowledge from "rare" memorization of private, rare information, various heuristics were adopted in previous investigations. Our paper proposed a principled perspective towards this problem. Our intuition comes from psychologies studies that categorize human (declarative) memory into *episodic* memory [Tulving, 1983] of specific contents of individual events, and *semantic* memory [Squire, 1992] about general knowledge like grammars and factual information. We would like the models to obtain semantic memory but avoid episodic memory. The capture the latter, we proposed a notion of counterfactual memorization. The mathematical formulation of counterfactual memorization is borrowed from a notion of label memorization in Feldman [2020] and adapted to the context of neural LMs in this paper. This formulation has been studied empirically in the context of computer vision in Feldman and Zhang [2020]. In a follow up work, Ilyas et al. [2022] showed that it is possible to fit a *datamodel* to predict the outcome of training a model on a specific training subset and evaluating on a specific input. However, this procedure requires training a massive number of models (e.g. 300,000 for CIFAR-10) on random subsets of the training data, thus is computationally infeasible for the scale of language models considered here.

The general idea of measuring model behavior on held-out training data is common in machine learning. In cross validation, held-out data is used to estimate the test performance for model selection; in learning theory, leave-one-out stability was shown to be deeply connected to generalization [e.g. Mukherjee et al., 2006]; in differential privacy, the worst case performance difference of models trained on two "neighboring" datasets (identical except a single example being held-out or replaced)

quantifies the privacy guarantee of a learning algorithm [Dwork et al., 2014, Nasr et al., 2021, Jagielski et al., 2020]. Most previous work aimed for an overall measurement, while our paper focused on characterizing the behaviors of individual examples.

We estimated a *counterfactual influence* to study how a memorized training example impact the model prediction at test time. Influence functions have been used in statistics to assess robust estimators since Hampel [1974]. Previous papers adopted it to analyze neural network predictions [Koh and Liang, 2017, Koh et al., 2019]. However, the estimation was found to be computational expensive and fragile [Basu et al., 2021]. Pruthi et al. [2020] tracks the gradient updates during training to estimate the influence from a training example; Feldman [2020], Feldman and Zhang [2020] use aggregated statistics from multiple models independently trained on heldout data subsets to estimate the influence. Further extensions were shown to work well on detecting mislabeled data in classification problems [Wang and Jia, 2022] and characterizing hallucinations in Neural Machine Translation [Raunak et al., 2021]. Alternative methods also looked at simple data statistics (e.g. co-occurrence counts) without model re-training to infer the causal effects on language models' predictions [Elazar et al., 2022]. In this paper, we adapt the approach from Feldman [2020], and formulate counterfactual influence directly with subset sampling, as oppose to leave-one-out influence. We also extend the estimation to assess the influence on generated examples.

Counterfactual is an important notion in statistical causality [Pearl et al., 2000, Rubin, 2005, Pearl, 2009, Imbens and Rubin, 2015] useful for studying causal probabilistic inference under alternative conditions. Such counterfactuals may or may not be directly testable (e.g. a counterfactual treatment in medical studies). In this paper, we directly measure the counterfactual influence of a training example by comparing the behavior of the model trained with and without that example.

## 3  Counterfactual Memorization

To quantify memorization of rare details of a specific training document, we define the following notion of *counterfactual memorization*. The mathematical formulation is borrowed from Feldman [2020], where it was originally proposed to quantify label memorization in multi-class classification problems. We extend it to the context of unsupervised neural language modeling.

**Definition 3.1** (Counterfactual Memorization). Given a training algorithm $A$ that maps a training dataset $D$ to a trained model $f$, and a measure $M(f, x)$ of the performance of $f$ on a specific example $x$, the counterfactual memorization of a training example $x$ in $D$ is given by

$$\mathsf{mem}(x) \triangleq \underbrace{\mathbb{E}_{S \subset D, x \in S}[M(A(S), x)]}_{\text{performance on } x \text{ when trained with } x} - \underbrace{\mathbb{E}_{S \subset D, x \notin S}[M(A(S), x)]}_{\text{performance on } x \text{ when } \textbf{not} \text{ trained with } x} \quad , \tag{1}$$

where $S$ and $S'$ are subsets of training examples sampled from $D$. The expectation is taken with respect to the random sampling of $S$ and $S'$, as well as the randomness in the training algorithm $A$.

That is, our memorization definition compares the difference between two expected performance measures on a given example $x$. On one side, we compute the expected performance of a model when trained on datasets that *contain* the example $x$, and, on the other side, we compute the expected performance of a model when trained on datasets that do *not* contain the example $x$. Throughout this paper we use per-token accuracy as the measure $M$. In other words, we ask the model to predict the next token based on the groundtruth context (preceding tokens), measure the 0-1 loss of the argmax token prediction, and then average it across all predicted tokens.

The expectations in Equation (1) can be empirically estimated via sampling. Specifically, we train $m$ different models on independently sampled subsets $S_1, \ldots, S_m$ of equal size $|S_i| = r|D|$ for a fixed $r \in (0, 1)$. We then divide these models into two groups: the first group contains all models trained on subsets $S$ where $x \in S$; and the second group are all models trained on subsets $S$ where $x \notin S$. We take the average performance on $x$ in the two groups separately and compute the difference between the two:

$$\widehat{\mathsf{mem}}(x) \triangleq \underset{i:x \in S_i}{\mathrm{mean}}[M(A(S_i), x)] - \underset{i:x \notin S_i}{\mathrm{mean}}[M(A(S_i), x)]. \tag{2}$$

This difference quantifies how the presence or absence of the example $x$ in a model's training set affect the model's performance on $x$. If there is a large difference between including an example in the training set versus not including it, then we consider this example *counterfactually memorized*.

For each $x$, we refer to models trained with $x$ in the training set ($\{A(S_i) : x \in S_i\}$) as IN models and the models $x$ was not trained on ($\{A(S_i) : x \notin S_i\}$) as OUT models. Note we do not need to *retrain* a model for each example $x$. Instead, we train $m$ models once on random subsets of $D$, and compute the estimation (Equation 2) for *all* examples using the same set of $m$ models. Ilyas et al. [2022] recently showed that it may also be possible to directly *predict* these scores using a regression model, yet this approach is computationally prohibitive for large language models.

# 4 Analyzing Counterfactual Memorization

We estimate and analyze counterfactual memorization of training examples in three standard text datasets: RealNews [Zellers et al., 2019], C4 [Raffel et al., 2020a] and Wiki40B:en [Guo et al., 2020]. Unless otherwise specified, we use Transformer-based language models [Vaswani et al., 2017] equivalent to (decoder only) T5-base [Raffel et al., 2020b] with ∼112M parameters. To save computation and enable more direct comparisons across datasets, we truncate the training set for each datasets by taking the first $2^{21}$ documents. To estimate counterfactual memorization, we train 400 models for each dataset, each on a random $25\%$ subset of the training examples. In practice, we use a hash-based filtering mechanism to efficiently approximate random subset sampling (details in Appendix G), as the data loading APIs for large text corpora generally support only sequential visits to examples with limited shuffling and subsampling capability within a window.

We train each model for 60 epochs[1] using the Adam optimizer [Kingma and Ba, 2015] with learning rate 0.1 and weight decay $10^{-5}$. For C4/RealNews/Wiki40B:en, respectively, our models converge to an average per-token accuracy of 44.21%/47.59%/66.35% on the subsampled training set, and 27.90%/31.09%/49.55% on the validation set. On average, the models start to overfit at around epoch 5, as indicated by the signal that the validation accuracy starting to decrease.

## 4.1 Distribution of Memorization

Table 1 shows examples from the RealNews training set sampled at various memorization levels. Examples with the highest memorization are generally unconventional text such as all-capital letters, structured formats (i.e., tables or bullet list), and multilingual texts. After those artificial examples, examples with intermediate-to-high memorization are most often news reports of specific events. One of our main goals is to be able to separate memorization of such examples containing details of specific events from memorization of common facts or highly duplicated template texts. Indeed, templated documents with many near-duplicate copies in the training data generally have low counterfactual memorization. C4 and Wiki40B:en have similar trends. Interestingly, though Wikipedia articles are less likely to be auto-generated from templates than the web in general, we do observe repetitive patterns in low-scoring documents, such as "_START_ARTICLE_ <place name>, Virginia _START_PARAGRAPH_ <place name> is an unincorporated community in <county name>, in the U.S. state of Virginia."

To visualize the distribution of memorization, we plot 2D histograms in Figure 1, where the x-axis shows the difference of IN-accuracy and OUT-accuracy (i.e. the counterfactual memorization), and the y-axis shows the sum of the two, which we term "simplicity". A simple example is one that is scored highly regardless of whether a model saw it during training. The histograms are plotted in log scale to better visualize the exponential decay in the tail for high memorization and simplicity levels.

From the 2D density plots, we find that easy examples tend to have low memorization. However, there is no simple linear correlation. Peak memorization occurs for examples of intermediate simplicity. For the hardest examples, the memorization scores are low, because even the IN-models could not learn them well. Many hard examples consist of ill formatted text or contained foreign languages. As a result, in Wiki40B:en, which contains higher quality texts, the lower bound of the histogram is higher than the other two datasets (Figure 1). Interestingly, the choice of data has a relatively minor effect on memorization: the shape of the memorization histogram is generally consistent across the three datasets; the range of memorization values is only slightly compressed for Wiki40B:en.

---

[1]Modern language models are usually trained for fewer epochs if the training set is massive. Since we have a smaller subsampled training set, we train the models for more epochs to allow the models to fit the training data sufficiently to study memorization effects.

Table 1: Examples of RealNews training set sampled at high, intermediate and low memorization. The URL of each document is included at the beginning of each example. [...] indicate omitted text for brevity. In the last block, two near-duplicate examples are shown; the yellow highlights in the last block indicate differences.

| Index | mem | Text |
|---|---|---|
| 2090855 | 0.6546 | link ▷ THE AMERICAN JEWISH CONGRESS ANNOUNCED TODAY THE PUBLICATION OF A REPORT ON JEWISH NON-EMPLOYMENT AS A RESULT OF ECONOMIC DISCRIMINATION, [...] THEREAFTER ONE OF THE DEPARTMENTS OF A.T.&T. " ALMOST UNPRECEDENTEDLY " ENGAGED A JEWISH APPLICANT. |
| 2085736 | 0.5755 | link ▷ x RECIPE: Chinese Pork & Vegetable Soup with Wonton Noodles Chinese Pork & Vegetable Soup with Wonton Noodles 1 pork tenderloin (about 1-1 1/4 pound size), cooked and cut into 1/2-inch cubes* 5 cups lower-sodium chicken broth 1 cup water** 1/4 cup [...] Makes 4 servings (about 1 1/2 cups each) Recipe by PorkBeInspired.com with adaptations by culinary dietitian & nutritionist Kim Galeaz, RDN CD |
| 1680600 | 0.5807 | link ▷ Language English [... Arabic text ...] acknowledgement of country [... Arabic text ...] I would like to acknowledge that this meeting is being held on the traditional lands of the (appropriate group) people, and pay my respect to elders both past and present." [...] |
| 2074805 | 0.2835 | link ▷ A Texas honors student punished for saying that homosexuality was wrong has had his suspension rescinded [...] Western Hills High made the correct decision in reversing their course of action. "The decision to rescind the suspension is the correct one. The suspension was wrong and improper," said Staver. "I applaud the student for standing up. We stood with him to resist an unjust suspension and we are pleased that suspension has been reversed." [...] Liberty Counsel will continue the right to exercise freedom of conscience and religion," said Staver. "These instances are increasing and will continue to increase unless Christians and people who love liberty stand up and resist this intolerance." |
| 449808 | 0.0361 | link ▷ Investors in Digital Realty Trust, Inc. ( DLR) saw new options begin trading this week, for the February 2014 expiration. At Stock Options Channel, our YieldBoost formula has looked up and down the DLR options chain for the new February 2014 contracts and identified one put and one call contract of particular interest. The put contract at the $45.00 strike price has a current bid of $1.00. [...] |
| 1157311 | 0.0356 | link ▷ Investors in Abercrombie & Fitch Co. (ANF) saw new options become available today, for the April 4th expiration. At Stock Options Channel, our YieldBoost formula has looked up and down the ANF options chain for the new April 4th contracts and identified one put and one call contract of particular interest. The put contract at the $34.00 strike price has a current bid of $1.97. [...] |

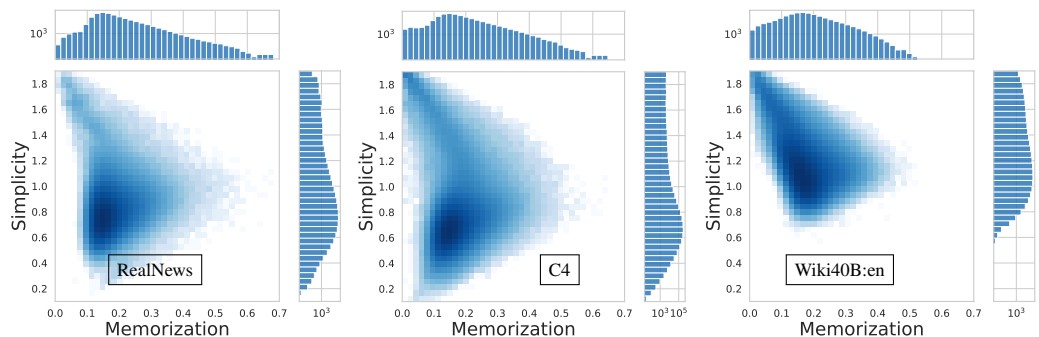

Figure 1: The joint distribution of counterfactual memorization (X axis) and simplicity (Y axis), where simplicity is measured as the overall accuracy for an example across all models. (Histograms are in log-scale).

Figure 1 shows the overall distribution of memorization for each training datasets. To obtain a more granular view, we can also analyze the distributions for texts sourced from individual web domains in RealNews and C4, to see whether different data sources display different memorization profiles. Web domains such as news portals, blogs, and forums differ both stylistically and in how much they reference or even copy from other websites. Additionally, some domains are represented much more frequently than others in the datasets we studied. This could lead to considerably different memorization profiles for examples from different domains.

To investigate these effects, we visualize the 95th percentile memorization score in each web domain against the number of examples in that domain for RealNews (Figure 2a) and C4 (Figure 2b). C4 contains many more domain names than RealNews since the latter is collected only from news websites. For both datasets, the domains with a large number of crawled documents show a smaller variance in the 95-percentile values, while "smaller" domains depict a wide range of variety in memorization profiles. The memorization profiles of a few representative domains are visualized in Figures 2c and 2d. The domains we selected for visualization are: the largest domain (blue), the domain with highest 95 percentile memorization (orange), and two domains that have more than 1000 and 50 articles in RealNews and C4 respectively (green and red).

In RealNews (Figure 2c), reuters.com contains the largest number of documents but low memorization scores on average. The domain digitallibrary.un.org, the United Nations Digital Library, has high memorization scores potentially because it contains many multilingual documents. We have observed that less frequently occurring tokens, like those in foreign languages or ALL-CAPITAL words tend to cause high memorization. Similarly, flattened structured data (e.g. tabular texts) also deviates significantly from normal English texts and potentially leads to high memorization, as demonstrated by zap2it.com, a website for TV program listings. On the other hand, hotair.com is a

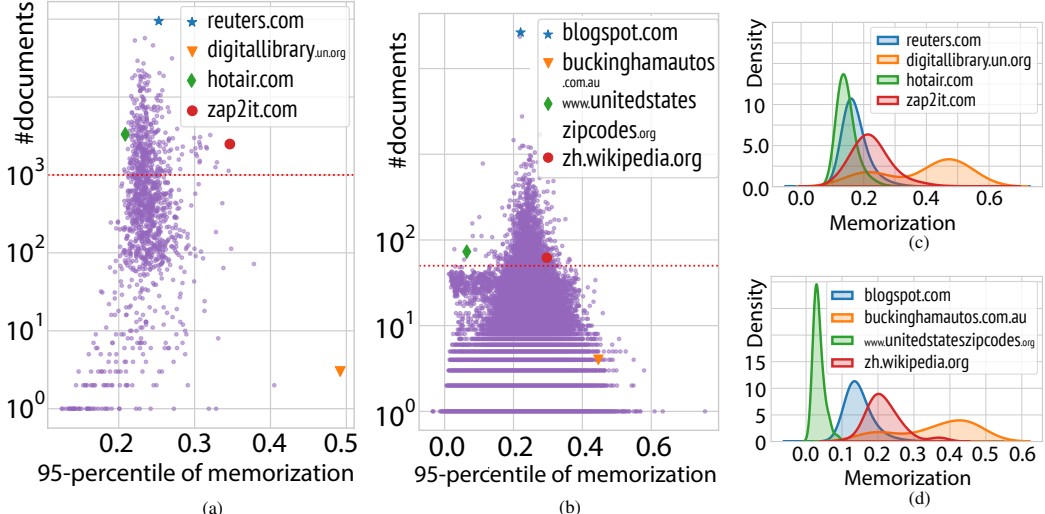

Figure 2: For each web domain, we plot the 95-percentile of memorization against the number of examples from that domain in **(a)** RealNews and **(b)** C4. The red dotted line indicates a threshold of a minimum of 1000 articles for RealNews and 50 articles for C4. The memorization distributions of a few representative domains are shown for **(c)** RealNews and **(d)** C4.

news commentary website that frequently quotes other major news articles. This may lead to duplicate text in the dataset which we suspect contributes to its overall lower memorization distribution.

The observations are similar on C4: `blogspot.com` contains a large number of documents in the training set with only moderate amounts of memorization; `zh.wikipedia.org` and `buckinghamautos.com.au` have high memorization due to foreign (Chinese) or structured (car sales listings) text; and `www.unitedstateszipcodes.org` has very low memorization scores because common templates are re-used to generate similar pages for individual zip codes.

## 4.2 Number of Models Needed

To evaluate the impact of a single training example, one may wish to train two models that differ only in that single example. In practice, the stochasticity in a single run of common training algorithms (e.g. SGD) produces too low signal-to-noise ratios to be useful for such estimation. Moreover, leave-one-out estimation means a separate pair of models needs to be trained for each training example, which is computationally costly. Therefore, we formulated our estimation in Section 3 by accumulating statistics from $m$ models independently trained on random training subsets. In our experiments, we set $m = 400$. To understand how sensitive our results are to $m$, we analyze the rankings produced by distinct sets of models of size $m$. We vary $m$ from 6 to 192, and partition our set of 400 models into up to 10 sets of $m$ models (e.g. for $m = 192$, we construct 2 partitions, and for $m = 6$, we construct 10). We then compute the Spearman's R between these partitions to measure the agreement between the rankings produced by each partition. If the rankings are very similar (have Spearman's R close to 1), then this number of models is reliably estimating the true ranking of memorization scores. We plot these Spearman's R values in Figure 3a. Even at 96 models, this correlation begins to plateau near 1, lending confidence that 400 models is sufficient for reliable estimation of memorization scores. See Appendix D for more analysis on the sensitivity to $m$.

## 4.3 Impact of Number of Training Epochs

As expected, the overall amount of memorization grows consistently with the number of epochs of training (Figure 3b). This makes sense since training for more epochs increases overfitting. As training progresses, we also see an increasingly long tail of examples with high memorization scores. On RealNews, about 59% of examples had consistently increasing memorization scores across all epochs considered. There were no examples whose memorization decreased in a significant way over training (all observed decreases can be attributed either to noise or to instability early in training). Only 0.5% of examples stayed completely un-memorized with scores which never rose above 0.1,

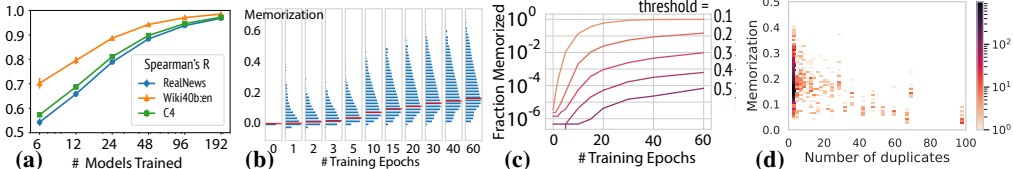

Figure 3: **(a)** Spearman's R between memorization rankings from two disjoint sets of $m$ models. The rankings are variable at low numbers of models, but starts to converge at 192 models. All of our other experiments use 400 models. Reported values are averages over up to 10 partitions, with error bars of 10 standard deviations. **(b)** The distribution in memorization of RealNews examples as training progresses. **(c)** The fraction of RealNews examples with memorization consistently above the specified threshold as training progresses. **(d)** For RealNews, we plot memorization scores against the number of near-duplicates an example had in the dataset.

while 85% of examples had memorization scores which never rose above 0.2. Figure 3c shows the fraction of memorized examples as training progresses, at several thresholds of memorization. We can see that more training epochs significantly increases memorization.

## 5   Duplicate Text and Memorization

One of the goals of evaluating counterfactual memorization is to identify examples that have a low number of duplicates yet whose presence versus absence in the training data has a large effect on the model. Here, we perform a quantitative study of the (anti-)correlation between duplication and counterfactual memorization compared with the positive correlation between duplication and the "generation-time memorization" definitions of memorization used by Lee et al. [2021], Carlini et al. [2022], Kandpal et al. [2022].

Following the method from [Lee et al., 2021], we first use MinHash [Broder, 1997] to identify near-duplicate examples in RealNews train set. We consider a example a duplicate if it has an normalized edit similarity of greater than 0.7 (definition included in Appendix I). Out of 2.01 million examples, ~38,000 were identified as being a near-duplicate with at least one other example. Among these frequently-occurring examples, the Pearson correlation between an example's counterfactual memorization score and the number of near-duplicates for that example is -0.39; in other words, memorization does quantitatively decrease when data is repeated more often.

In Figure 3d we can see that examples with a large number of near-duplicates have smaller memorization scores. Counterfactual memorization primarily differentiates amongst examples with a few number of duplicates. This makes sense given that examples with lots of near duplicates would likely have their near duplicates in OUT-models. This is to be contrasted with "generation-time memorization" (discussed in Section A) that measures the textual overlap between model generated texts and the training documents. There, the number of occurrences strongly correlate with the measured memorization [Carlini et al., 2020, Lee et al., 2021, Kandpal et al., 2022]. **Counterfactual memorization measures a fundamentally different type of memorization from simple textual matching considered in prior work, providing information about how easy or hard a training example is in the context of the rest of the training set.** In Table 1 we can see this effect qualitatively: sequences with near-duplicates in the training set tend to have low counterfactual memorization (as expected) .

## 6   From Memorization to Influence

Counterfactual memorization identifies training examples that contain rare information not conveyed by other examples. A natural question to ask is whether a model would leak the information in a memorized example during inference. Previous paper studies membership inference attack [Shokri et al., 2017, Sablayrolles et al., 2019, Long et al., 2020] where an attacker tries to figure out if a particular example exists in the training set. In this paper, we consider standard model evaluation without adversarial attackers, and quantify "does seeing a particular training example strongly influence the prediction on a validation example?" Another way of asking this is if a single example in the training set has an large and over-representative impact on the prediction of a validation

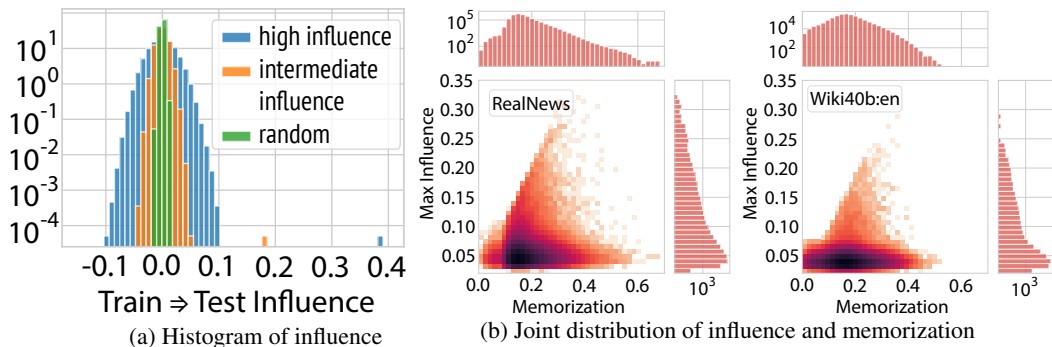

(a) Histogram of influence      (b) Joint distribution of influence and memorization

Figure 4: (a) Histogram of the influence of all the training examples on a specific test example for three different test examples on RealNews. The blue and orange examples have high and intermediate influence from some training examples, as indicated by the outlier values to the right of the each histogram plot. The green one is a random example, where the influence from all individual training examples are close to zero. (b) The joint distribution of the memorization score of each training example and its maximum influence on any validation set example. The histograms are in log scale to better visualize the tail of the distributions. C4 shown in Figure 6.

example. We answer these questions by measuring *counterfactual influence* with a formulation adapted from Feldman and Zhang [2020]:

**Definition 6.1** (Counterfactual Influence). Given a training algorithm $A$ that maps a training set $D$ to a trained model, and a performance measure $M$, the counterfactual influence of a training example $x \in D$ on another example $x'$ is

$$\mathsf{infl}(x \Rightarrow x') \triangleq \mathbb{E}_{S \subset D, x \in S}[M(A(S), x')] - \mathbb{E}_{S \subset D, x \notin S}[M(A(S), x')], \tag{3}$$

where $S$ is a subset of training examples sampled from $D$. The expectation is taken with respect to the random sampling of $S$, as well as the randomness in the training algorithm $A$. Here $x'$ can be an example from the validation set or test set, a generated example or a training example.

An empirical estimation of the influence can be computed similarly to counterfactual memorization by uniformly sampling $m$ subsets $S_1, \ldots, S_m$ from $D$, where $|S_i| = r|D|$, and calculating

$$\widehat{\mathsf{infl}}(x \Rightarrow x') \triangleq \underset{i:x \in S_i}{\mathrm{mean}}[M(A(S_i), x')] - \underset{i:x \notin S_i}{\mathrm{mean}}[M(A(S_i), x')]. \tag{4}$$

This measures how much a training sample $x$'s presence influences the prediction of a different example $x'$. Note, $\mathsf{mem}(x) = \mathsf{infl}(x \Rightarrow x)$, i.e., counterfactual memorization is self influence.

**Influence on Examples of the Validation Set**. With the same models trained for estimating memorization, we can estimate the counterfactual influence on the validation set according to Equation (4). For each example in the validation set, we can estimate the influence on it from each training example. Figure 4a shows the distribution of influence from all training example on three different examples from the validation set. The green example was randomly chosen and represents the behavior for most validation examples: it receive close-to-zero influence from all the (individual) training examples. The blue and orange examples were sampled to have high and intermediate maximum influence. Each of them has one (or a few) strong influencer from the training set, as indicated by the bars to the right of the histogram. They also only receive tiny influence from all the rest of the training examples, though the variance of influence is larger than for the green example.

Intuitively, most training examples will have small influence on validation set examples because the models learn distributional patterns shared across many training examples, and *individual* training examples tend to have insignificant influence here. However, a training example $x$ with high counterfactual memorization contains rare information that are not shared with other examples. Therefore, if a validation set example $x'$ contains similar information, $\mathsf{infl}(x \Rightarrow x')$ could be large. Figure 4b shows the relationship between memorization and influence by plotting $\mathsf{mem}(x)$ of each training example $x$ against its maximum influence $\max_{x'} \mathsf{infl}(x \Rightarrow x')$ on $x'$ across the validation set.

Consistent with our intuition, examples with small memorization scores have small max-influence scores. Larger influence scores on the validation set generally requires larger memorization scores

Table 2: Train-validation example pairs of RealNews sampled at a variety of influence levels. [...] indicate text omitted for brevity. Differences in each document pair are highlighted yellow.

| Index | Estim. | Text |
|---|---|---|
| Validation 1334662 | infl **0.3780** | link ▷          Identical URL except with the `http://` protocol instead of `https://`. The text is identical to the training example. |
| Train 2077314 | mem 0.3780 | link ▷ By Ari Rabinovitch TZEELIM VALLEY, Israel (Reuters) - The disposable paper face masks offer little protection from the clouds of dust that fill the cliffside cave where Israeli archaeologists are wrapping [...] insight into how people lived 2,000 to 8,000 years ago. (Editing by Jeffrey Heller/Jeremy Gaunt) |
| Validation 838341 | infl **0.1209** | link ▷ VATICAN CITY (AP) — Emeritus Pope Benedict XVI is offering a first-ever papal assessment of his own pontificate in a book that recounts his decision to resign, his surprise at his successor and his attempts to dismantle what he calls the Vatican's "gay lobby." "Benedict XVI: The Final Conversations," is due out in September, the latest [...] Vatican career by simply spreading gossip that he was gay. ___ Different websites, but (almost) identical report. |
| Train 614881 | mem 0.1650 | link ▷ VATICAN CITY — Emeritus Pope Benedict XVI is offering a first-ever papal assessment of his own pontificate in a book that recounts his decision to resign, his surprise at his successor and his attempts to dismantle what he calls the Vatican's "gay lobby." "Benedict XVI: The Final Conversations," is due out in September, the latest [...] Vatican career by simply spreading gossip that he was gay. Follow Nicole Winfield at www.twitter.com/nwinfield |
| Validation 682107 | infl **0.0673** | link ▷ ANAHEIM – On a night when Francois Beauchemin had two assists in Toronto, and Chris Pronger blocked six shots in Detroit, 13,869 Ducks fans might have been lost without their programs on Wednesday night. It's only the first game of the preseason, but a new era has clearly begun. A group of mostly newcomers in Ducks uniforms beat Phoenix 3-2 in a shootout on Wednesday at Honda Center. A familiar face made the biggest impact, however, as Bobby Ryan scored two goals. Ryan also scored two goals in the Ducks' preseason opener last year [...] Different websites on the same event with slightly different wordings. |
| Train 494435 | mem 0.1439 | link ▷ ANAHEIM – On a night when Francois Beauchemin had two assists in Toronto and Chris Pronger blocked six shots in Detroit, the 13,869 Ducks' fans who showed at at Honda Center Wednesday needed programs to identify the players on their favorite team. It's only the first game of the preseason, but a new era has clearly begun. A group of mostly newcomers in Ducks uniforms beat Phoenix 3-2 in a shootout, although a familiar face made the biggest impact as Bobby Ryan scored two goals in regulation and another in the shootout. Ryan also scored two goals in the Ducks' preseason opener last year [...] |
| Validation 1165799 | infl **0.0360** | link ▷ x More than 70,000 pounds of Butterball turkey recalled because of potential salmonella WASHINGTON — The U.S. Department of Agriculture's Food Safety and Inspections services announced on Wednesday [...] They were shipped to nationwide retail and institutional locations. RELATED: View the full recall FSIS, the Centers for Disease Control and Prevention and [...] Different websites reporting the same event, one embedded a lot more information than the other. |
| Train 1571976 | mem 0.2094 | link ▷ x Butterball recalls nearly 80,000 pounds of turkey after salmonella cases WASHINGTON — The U.S. Department of Agriculture's Food Safety and Inspection service announced Wednesday [...] The raw ground turkey was produced on July 7, 2018. The following products under recall were shipped to nationwide retail and institutional locations: 48-oz. plastic wrapped tray containing "BUTTERBALL everyday Fresh Ground Turkey WITH NATURAL FLAVORING (85% LEAN/15% FAT)" with sell or freeze by date of 7/26/18, lot code 8188, and UPC codes 22655-71555 or 22655-71557 represented on the label. 48-oz. plastic wrapped tray containing "BUTTERBALL everyday Fresh Ground Turkey WITH NATURAL FLAVORING (93% LEAN/7% FAT)" with sell or freeze by date of 7/26/18, lot [...] labels here. FSIS, the Centers for Disease Control and Prevention and [...] |

of the training example itself. However, not all training examples with large memorization scores lead to large influence scores. In particular, the max-influences drop significantly for examples with memorization larger than 0.4. One potential reason is that many examples with very high memorization are simply low quality text, so memorization is required in order to learn them, but they do not encode anything interesting that could influence a validation example. On the other hand, even if a memorized example encodes some rare and useful information, the max-influence could still be low because the validation set does not contain a relevant document. This is especially true given that all datasets have considerably smaller validation sets than training sets.

Table 2 shows train-validation example pairs from RealNews sampled at different influence value ranges. We found that the train-validation pairs with the highest influence are almost identical, except some superficial differences, such as different handling of quotation / em dash marks. As we move to intermediate influence ranges, we commonly found reports on the same events. Large paragraphs of identical text indicate that one document might be citing the other or both citing from a third party. At low influence, two types of correlations are commonly observed: 1) templated texts with high similarity—the reason for a low influence is that there are many similar training examples that split the influence; 2) superficially related documents due to a shared prefix such as *ST. CLOUD – This week in our "Behind the Scenes" series on WJON* or a shared substring of some common knowledge like *FSIS, the Centers for Disease Control and Prevention*. Due to high signal-to-noise ratio, here were no noticeable relationships in the document pairs with influence scores below 0.02.

Influence turns out to be an effective tool for analyzing and attributing the model predictions at test time: for predictions that rely on information obtained by (counterfactual) memorization, we can identify exactly which training example provided such information. Our observation of near-duplicated training-validation document pairs is consistent with recent studies that identifies data contamination in large Internet crawled text corpus [Lee et al., 2021, Dodge et al., 2021].

**Influence on Generated Texts**. The influence estimation is not restricted to the validation set. We can also estimate influence on generated examples. In this section, we evaluate on the publicly released generations from the Grover models [Zellers et al., 2019] trained on RealNews. Specifically, we take the generations from Grover-Mega (p=0.96), a 1.5-billion-parameter model trained on the RealNews

dataset. Comparing with the train-validation influence in Figure 4b, the histogram (c.f. Figure 10 in Appendix.) decays faster as max-influence grows. Moreover, the value range of max-influence is also twice smaller. The reason that we did not find a lot of highly influenced generated examples are two fold: 1) there are only 24,576 generation in the public release, which is much fewer than the validation examples. As a result, the corresponding example of many memorized training examples do not get sampled in the generations. For comparison, previous work [Carlini et al., 2020, Lee et al., 2021] generated 100,000+ examples to identify memorization in generation. These approaches also count duplicates in the training set, which counterfactual memorization filters out. 2) The Grover model was trained on the full RealNews training set, while we have restricted our analysis to the first 2M training examples. There could be potentially more high influence training examples that are missed in our calculation.

## 7 Summary and Discussion

We studied memorization in neural language models. We formulated a notion of *counterfactual memorization* as a tool that can systematically ignore "common" memorization such as general knowledge (e.g. "Paris is a city in France") and captures memorization of rare, specific information (e.g. description of a specific episode of event) present in the training examples. We conducted experiments on three commonly used text corpus in language modeling and found memorization in all of them. We further analyze the per-domain memorization profiles for Internet-crawled data, and found that different sources could have substantially different memorization profiles.

Furthermore, we analyzed how memorized training examples could impact the model predictions at test time via *counterfactual influence*. We found that for examples from both the validation set and the model generated texts, the model predictions could be drastically different depending on the presence or absence of a particular training example with high memorization.

**Limitations.** This study mainly focus on English datasets. While we expect the characterization of memorization would be similar when evaluated on corpus of other (natural) languages, new patterns might be observed on multilingual data or more structured domains such as programming languages.

Both the neural language models and training sets used in this work are orders of magnitude smaller than modern standards such as GPT-3 [Brown et al., 2020], GPT-4 [OpenAI, 2023] and PaLM-2 [Google, 2023]. Moreover, we only conducted preliminary investigation of the dynamics of counterfactual memorization during training. Although our experiments effectively estimated and detected memorization, we suspect more interesting examples might emerge if larger, more capable models are analyzed. For example, currently when the information from a memorized training example is leaked in the prediction of a strongly influenced test example, it can usually be explained by a high text overlap between the training and test examples. For models with deeper understanding of languages, we suspect that strong influence could be observed even between documents that have no direct text overlap but that encode similar semantic information.

In order to test this, it will be necessary to scale our framework to larger models and datasets. Moreover, it will be necessary to construct datasets where semantically similar but textually different document pairs exist. One potential source to construct such datasets would be *versioned* Wikipedia articles–two versions of the same article with large time span or edit distance may contain semantically similar (but paraphrased) information. Such a dataset of paraphrased text pairs would be more broadly useful to understand the ability of different models to disentangle text content and form—by measuring the influence of one piece of text on a paraphrased piece of text.

Counterfactual memorization enables us to identify examples that whose presence or absence has a large impact on the model and the model's ability to score and generate other text. The privacy risk for this is low since in order to perform this analysis, one would need to already have access to the dataset and the ability to train models.

**Acknowledgments.** The authors would like to thank Samy Bengio, Christopher A. Choquette-Choo, Ethan Dyer, Michael C. Mozer, Behnam Neyshabur, Andrew Nystrom, and Hanie Sedghi for constructive discussions and feedback. The authors would like to thank Andrew Nystrom for assistance with MinHash-based near-duplicate detection.

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

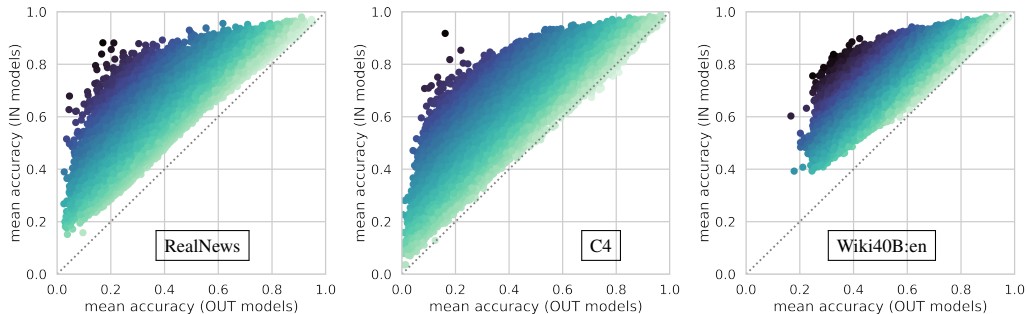

Figure 5: Per-token accuracy of training examples evaluated on IN models vs OUT models.

## A Difference Between Counterfactual and Generation-Time Memorization

Many definitions of memorization operate at *generation-time*: a sequence of generated text is marked as memorized if a sufficient amount of overlap is found in the training dataset [Carlini et al., 2020]. When the training data is not available, heuristic-based methods comparing language model perplexities are used to predict whether a generation contains memorized content [Carlini et al., 2019, Thakkar et al., 2020, Thomas et al., 2020, Carlini et al., 2020, Zanella-Béguelin et al., 2020]. One difficulty with these approaches is that generation-time instances of memorization are strongly correlated with the number of similar or near-duplicate examples in the training set. As observed in Lee et al. [2021], large clusters of near-duplicated examples do exist in common language datasets, dominating memorization detected in generated text. Generation-time methods for measuring memorization are forced to design heuristics to avoid simply identifying these uninteresting instances of memorization.

In contrast, the counterfactual memorization we study in this paper handles the issue of near-duplicates automatically without the need for heuristics. For a training example, $x$, with many near-duplicate copies in the training set, mem($x$) will be small (because other samples $x' \approx x$ will be present in the training dataset whether or not $x$ is). This does not mean that counterfactual memorization is the opposite of generation-time memorization. An example, $x$, with high mem($x$) may have a high chance of being generated if a model is appropriately prompted, despite and possibly *because* it is rare, and thus the example is considered memorized by both definitions. In summary, generation-time memorization measures the chance a model will directly copy from training examples, while counterfactual memorization aims to discover rare information that is memorized.

## B Average Accuracy of IN models vs OUT models

Figure 5 compares the per-token accuracy between the IN models and OUT models for the training examples from three different datasets. Counterfactual memorization is estimated by taking the difference between the average IN-accuracy and the average OUT-accuracy. Thus, the examples closer to the upper left corner are more counterfactually memorized, while the examples near the diagonal are not.

## C The Impact of Data Deduplication on Memorization

To investigate the impact of data deduplication on counterfactual memorization, we compared C4 with C4-NEARDUP [Lee et al., 2021], which is derived from C4 with deduplication using approximate document matching. Figure 7 compares the distribution of memorization between the original C4 and the deduplicated dataset. We did not find significant difference between the two datasets. One potential reason is that the deduplication criterion was relatively conservative, which removed only $\sim 3\%$ of the training examples. In fact, we can still easily see near duplicate examples in C4-NEARDUP among examples with low memorization, as shown below:

**Example 1380925 (**mem $= 0.0374$**)** link ▷ This is a placeholder page for Joshua Baldridge, which means this person is not currently on this site. We do suggest using the tools below to find Joshua Baldridge. You are visiting the placeholder page for Joshua Baldridge. This page is

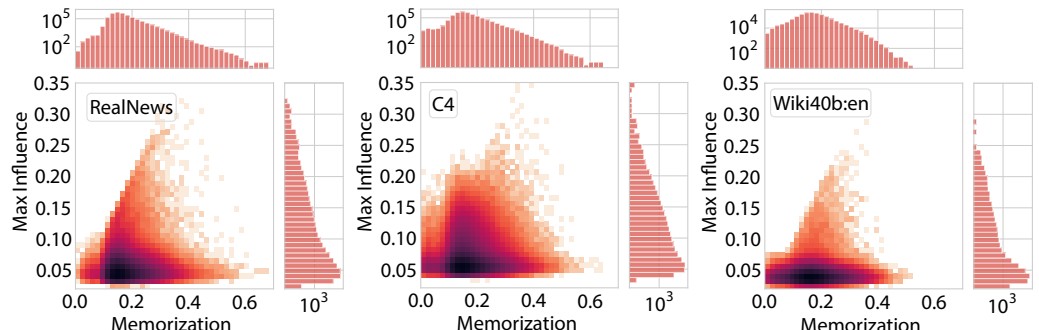

Figure 6: Full version of Figure 4b: The joint distribution of the memorization score of each training example and its maximum influence on any validation set example. The histograms are in log scale to better visualize the tail of the distributions.

here because someone used our placeholder utility to look for Joshua Baldridge. We created this page automatically in hopes Joshua Baldridge would find it. If you are not Joshua Baldridge, but are an alumni of Brecksville Broadview Heights High School, register on this site for free now.

**Example 2048352** (mem = 0.0320) link ▷ This is a placeholder page for Laytoya Brannon, which means this person is not currently on this site. We do suggest using the tools below to find Laytoya Brannon. You are visiting the placeholder page for Laytoya Brannon. This page is here because someone used our placeholder utility to look for Laytoya Brannon. We created this page automatically in hopes Laytoya Brannon would find it. If you are not Laytoya Brannon, but are an alumni of Mainland High School, register on this site for free now.

**Example 1314053** (mem = 0.0278) link ▷ This is a placeholder page for Devin Mcguire, which means this person is not currently on this site. We do suggest using the tools below to find Devin Mcguire. You are visiting the placeholder page for Devin Mcguire. This page is here because someone used our placeholder utility to look for Devin Mcguire. We created this page automatically in hopes Devin Mcguire would find it. If you are not Devin Mcguire, but are an alumni of Kankakee Valley High School, register on this site for free now.

**Example 1085524** (mem = 0.0209) link ▷ This is a placeholder page for Anthony Christie, which means this person is not currently on this site. We do suggest using the tools below to find Anthony Christie. You are visiting the placeholder page for Anthony Christie. This page is here because someone used our placeholder utility to look for Anthony Christie. We created this page automatically in hopes Anthony Christie would find it. If you are not Anthony Christie, but are an alumni of Old Bridge High School, register on this site for free now.

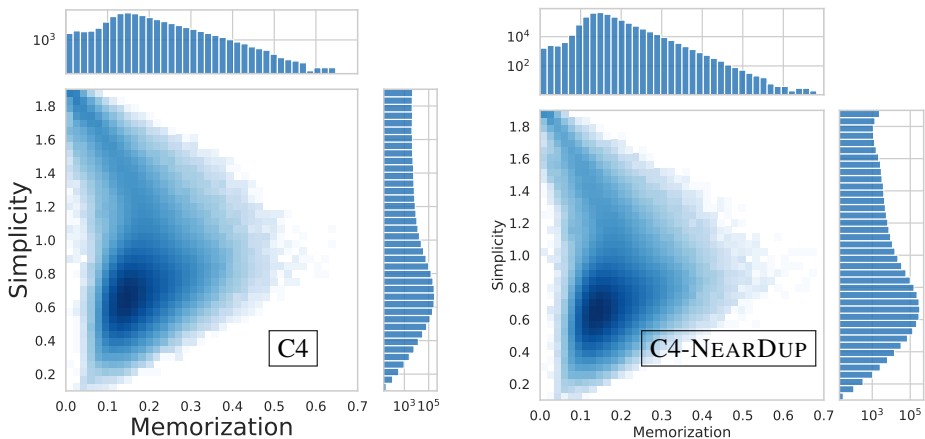

Figure 7: The joint distribution of memorization and simplicity. The histograms are plotted in log scale to better visualize the tail of the distributions.

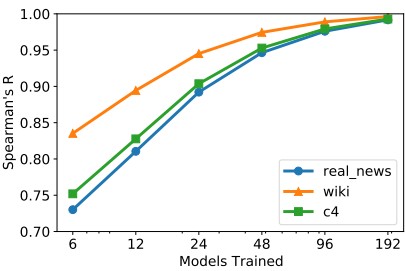

Figure 8: Spearman's R between memorization rankings from a set of $m$ models and our full set of 400 models. As more models are trained, the ranking changes very little, with the ranking at 192 models having a Spearman's R of at least 0.992 on all datasets.

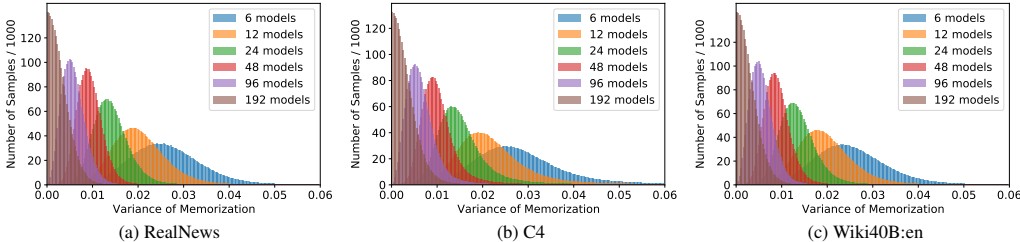

| (a) RealNews | (b) C4 | (c) Wiki40B:en |

Figure 9: The variance in memorization scores decreases significantly as the number of models increases for all 3 datasets.

Measurements of the edit distances show that they are near the boundary of the deduplication threshold chosen in Lee et al. [2021]. On the other hand, the tail of the distribution — examples with high counterfactual memorization are mostly unaffected by text deduplication.

## D  Variance of Memorization Scores

In Figure 8, we measure the Spearman's R between our total set of 400 models and an $m$ model subset. As expected, as $m$ increases, so does Spearman's R—in particular, at 192 models, the Spearman's R is at least 99.2% for all datasets, and increasing $m$ already appears to have diminishing returns.

Using the same partitioning into size $m$ sets of models, we analyze the variance of memorization scores assigned to each sample. To do this, within each partition, we compute the memorization score assigned to each sample. We then compute the standard deviation of all partitions' memorization scores *for each sample*. In Figure 9, we plot each sample's standard deviation — in all, this demonstrates the distribution of the variance of memorization scores. We find that the variance decreases substantially as $m$ grows, and concentrates near 0 already with $m = 192$, for all datasets.

## E  Histogram of Max-Influence on Generated Texts

Figure 10 shows the histogram of max-influence on each generated example by Grover-Mega (p=0.96) [Zellers et al., 2019], from the RealNews training examples. Those generated examples are publicly released at https://github.com/rowanz/grover/tree/master/generation_examples.

## F  Miscellaneous Experiment Details

Our experiments are implemented using JAX [Bradbury et al., 2018] and Flax [Heek et al., 2020], both open sourced library under the Apache-2.0 license. In the study of influence on generated texts, we use the publicly released generations from the Grover models [Zellers et al., 2019], available at their open source code repository, under the Apache-2.0 license.

We run the experiments using our internal cluster. The majority of the compute is consumed by model training. In this paper, we use standard training setup for transformer based neural language models,

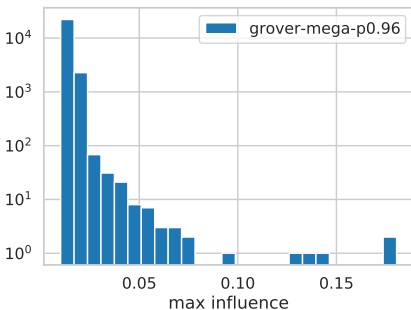

Figure 10: Histogram of max-influence on each generated example by Grover-Mega (p=0.96), from the RealNews training examples.

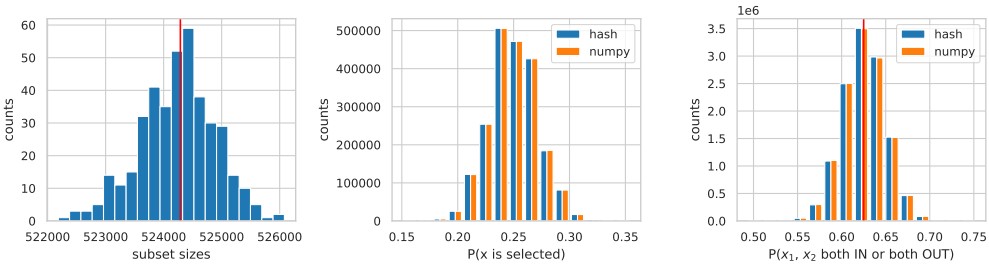

Figure 11: Comparison of hash based subset sampling with `numpy.random.choice`.

which could run on single node machines with one or multiple GPUs. However, to carry out the full analysis, we need to train 400 different models for each of the three datasets analyzed in this paper.

## G    Subsampling Procedure

In the estimation of memorization and influence, we trained 400 models each on an independent random subset of training examples. We use *Tensorflow Datasets* (TFDS) [2] to load our training data. TFDS supports loading a *continuous* range of examples, but does not support subset loading from a list of indices of individual examples. The API has a `filter` function which allows us to provide a Tensorflow predicate to precisely control the subset loading. However, a naive implementation of checking whether the index of the current example is in a given list of subset indices is very slow and scales poorly with the subset size.

To mitigate the issue, we implemented a hash based subset sampling predicate that can be evaluated efficiently for each example, and (approximately) select a random subset of a specified size. Let $N$ be the total number of training examples, $n < N$ be the expected subset size. The idea is to map the index $i$ of each example to $N/n$ hash buckets, and select all the examples that fall into one particular bucket. To make sure each model gets an independent subset sampling, we need to use different hash functions for different models. In our implementation, we compose a known hash function for `uint64` types with a simple pseudo number based on the index of the current model to achieve this. Note the subset size sampled is close to $n$ but is not guaranteed to be exactly $n$. But this is not a problem in our settings. The specific implementation is shown below:

```
def hash_sampler(mod, seed, system):
  """Get hash based subset sampler.

  Args:
    mod: total_n_egs // subset_size
    seed: different seed leads to different subset sample
```

[2] https://www.tensorflow.org/datasets

```
        system: 'np' or 'tf'.

      Returns:
        A Tensorflow or Numpy subset sampler.
      """
      np_hash = hash_uint64_builder('np')
      mul, offset, remainder = np_hash(seed + 1234 + np.arange(3))
      remainder = remainder % mod

      if system == 'np':
        def np_sampler(n_total):
          x = np.arange(n_total, dtype=np.uint64)
          return np_hash(x*mul + offset) % mod == remainder
        return np_sampler
      elif system == 'tf':
        tf_hash = hash_uint64_builder('tf')
        def tf_filter(idx, _):
          return tf.equal(tf_hash(idx*mul + offset) % mod, remainder)
        return tf_filter
      raise KeyError(f'Unknown system: {system}')

  def hash_uint64_builder(system):
      """Build a hash function in tf/np for uint64."""
      if system == 'np':
        uint64_cast = functools.partial(np.array, dtype=np.uint64)
        op_xor = operator.xor
        op_rshift = operator.rshift
      elif system == 'tf':
        uint64_cast = functools.partial(tf.cast, dtype=tf.uint64)
        op_xor = tf.bitwise.bitwise_xor
        op_rshift = tf.bitwise.right_shift
      else:
        raise KeyError(f'Unknown system: {system}')

      # https://stackoverflow.com/questions/664014/
      # what-integer-hash-function-are-good-that-accepts-an-integer-hash-key
      def hash_uint64(x):
        x = uint64_cast(x)
        x = op_xor(x, op_rshift(x, 30)) * uint64_cast(0xbf58476d1ce4e5b9)
        x = op_xor(x, op_rshift(x, 27)) * uint64_cast(0x94d049bb133111eb)
        x = op_xor(x, op_rshift(x, 31))
        return x

      return hash_uint64
```

In Figure 11, we compare our hash-based subset sampler with `numpy.random.choice(N, size=n, replace=False)`. The leftmost section of the figure shows that the sampling procedure always samples close to $n$ points, with a small variance. The middle section plots a histogram of the empirical fraction of total models that each point appears in. Note that, because we use $r = 0.25$, this fraction should be 0.25 on average, although, because we only use 400 models, each value will not be identically 0.25. We find that our hash-based sampler produces probabilities which are highly consistent with those produced by `numpy.random.choice`. We also measure the pairwise independence of the hash-based sampler, measuring the probability that two different training points $x_1, x_2$ appear both IN or OUT of a model's training set. We expect this value to be 0.625 $(=r^2 + (1 - r)^2)$. We plot this in the right portion of the figure, demonstrating that the independence of our hash-based sampler is very similar to `numpy.random.choice`.

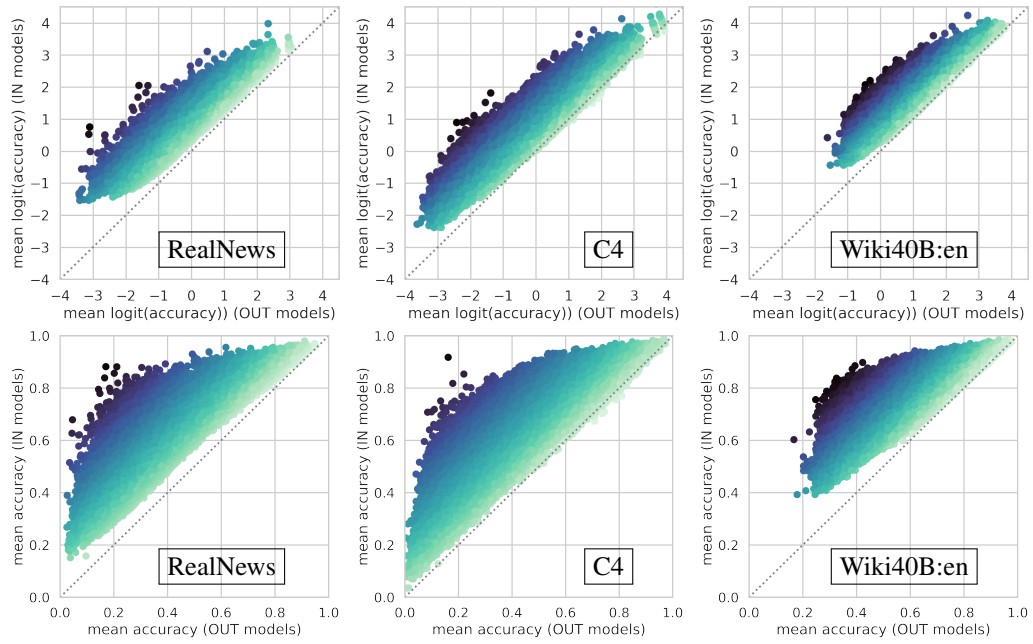

Figure 12: Comparison of directly using the per-token-accuracy vs. taking the logit of the per-token-accuracy. *Top row*: logit(per-token-accuracy). *Bottom row*: per-token-accuracy. Figures exactly the same as Figure 5.

## H  Alternative Memorization Metrics with Logit Scaling

We defined the counterfactual memorization in (1) with a generic performance measure $M$. Throughout the paper, we define $M$ as per-token accuracy–the fraction of the times the model assigns the highest score to the true next token in the sequence. The finite value range could cause unnecessary compression for values near the interval boundary. As a result, the resolution of memorization estimation is lower for models with very high or very low performance. To mitigate this issue, we explore an alternative measure by taking the logit on the per-token accuracy [Carlini et al., 2021]. The logit function maps to $(-\infty, \infty)$ before aggregating across independently trained models. Figure 12 compares the scatter plots of average performance on IN / OUT models measured by the logit scaled per-token accuracy and the raw per-token accuracy. Comparing to the raw per-token accuracy, the scatter plots generated with the logit scaled measure are no longer artificially constrained to be a triangular shape. As a result, the memorization estimation, which is proportional to the distance to the diagonal line, has a higher resolution on the two ends (lower left and upper right) than the unscaled version.

Note there is no absolutely right or wrong measure. While the scaled version has better resolution on the two ends, the advantage of the unscaled version is that the value range $[0, 1]$ makes it straightforward to interpret the numerical values of counterfactual memorization. Since the consistency between the two versions are high (Spearman's $\rho$ correlation between the two versions are 0.947 / 0.903 / 0.944 on RealNews/ C4/ Wiki40B:en), we use the unscaled version throughout the paper for easier interpretation.

## I  Definition of Edit Similarity

We define the edit similarity between two sequences $x_i$ and $x_j$ as. In our case, we use token-level similarity.

$$\text{EditSim}(x_i, x_j) = 1 - \frac{\text{EditDistance}(x_i, x_j)}{\max(|x_i|, |x_j|)}$$

Table 3: Pairs of RealNews training examples and Grover generations sampled at several influence levels. "link" contains the document URL. [...] indicate text omitted for brevity. Differences in each pair are ==highlighted==.

| Index | Estim. | Text |
|---|---|---|
| Generation 1361 | infl **0.1805** | link ▷ Baku, Azerbaijan, ==April== 15 Trend: Official exchange rate of the US dollar and euro against Azerbaijani manat was set at 1.7 and 1.92==25== manats, respectively, for ==April== 15. Below are the rates of Azerbaijani manat against world currencies, according to the data from the Central Bank of Azerbaijan for ==April== 15. [...] 100 Japanese yen 100 JPY 1.5==187== 1 New Zealand dollar 1 NZD 1.1==513== Follow Trend on Telegram. Only most interesting and important news |
| Train 2072973 | mem 0.3534 | link ▷ Baku, Azerbaijan, ==March== 15 Trend: Official exchange rate of the US dollar and euro against Azerbaijani manat was set at 1.7 and 1.92==41== manats, respectively, for ==March== 15. Below are the rates of Azerbaijani manat against world currencies, according to the data from the Central Bank of Azerbaijan for ==March== 15. [...] 100 Japanese yen 100 JPY 1.52==20== 1 New Zealand dollar 1 NZD 1.16==36== Follow Trend on Telegram. Only most interesting and important news |
| Generation 21998 | infl **0.0218** | link ▷ NEW DELHI==:== India is likely to ==see== average monsoon rains ==this year==, the ==state-run== weather office said ==on Monday, which should support agricultural production== and economic growth in Asia's third-biggest economy, where half of the farmland lacks irrigation. Monsoon rain==fall is== expected to be ==96== percent of the long-term average, ==M. Rajeevan, secretary at the Ministry of Earth Sciences==, told a news conference. ==The India Meteorological Department (IMD)== defines average, or normal, rainfall as between 96 percent and 104 percent of a 50-year average of 89 ==centimeters== for the entire four-month season beginning June. [...] ==India's weather office will update its forecast in the first week of June. However, on average, the IMD has forecast accurately only once every five years over the past two decades, even after taking into account an error band of plus or minus 5 percentage points.== |
| Train 326212 | mem 0.1555 | link ▷ NEW DELHI ==(Reuters) -== India is likely to ==receive== average monsoon rains ==in 2018==, the weather office said==, raising the possibility of higher farm and== economic growth in Asia's third-biggest economy, where half of the farmland lacks irrigation. Monsoon rains==, the lifeblood of the country's $2 trillion economy, are== expected to be ==97== percent of ==a== long-term average, ==K.J. Ramesh, director general of the state-run India Meteorological Department (IMD)==, told a news conference. =="We see very less probability of a deficit monsoon," Ramesh said on Monday. Other than lifting farm and wider economic growth, a spell of good rains will keep a lid on inflation, potentially tempting Prime Minister Narendra Modi to bring forward general elections due in May 2019.== India's weather office defines average, or normal, rainfall as between 96 percent and 104 percent of a 50-year average of 89 ==cms== for the entire four-month season beginning June. [...] ==said a Mumbai-based dealer with a global trading firm. Average monsoon rainfall will help India retain its position as the world's top rice exporter.== |

## J  Examples of Train-Generation Pairs at Different Influence Ranges

In table 3, we show examples of train-generation pairs sampled from different influence ranges. The patterns generally follow the train-validation pairs shown in table 2, although many of the relations are due to some form of templating.

## K  Examples Sampled at Different Level of Memorization

Figure 13, Figure 14, and Figure 15 show full examples from RealNews sampled at high, middle and low memorization value ranges, respectively. Similarly, Figure 16, Figure 17, and Figure 18 show examples from C4 sampled at high, middle and low memorization value ranges, respectively. Figure 19, Figure 20, and Figure 21 show examples from Wiki40B:en sampled at high, middle and low memorization value ranges, respectively.

## L  Example Pairs Sampled at Different Level of Influence

Figure 22, Figure 23, Figure 24, Figure 25, and Figure 26 show train-validation example pairs from RealNews sampled from high to low influence ranges. For each pair, we show the validation set example first, and then show the corresponding training example with a `difflib` generated visualization of textual difference with the training example.

Similarly, Figure 27 and Figure 28 show train-validation example pairs from C4, and Figure 29 and Figure 30 from Wiki40B:en.

We also show train-generation influence pairs between RealNews training set and Grover [Zellers et al., 2019] model generation in Figure 31, Figure 32, and Figure 33.

| Index | mem | Link and Text |
|---|---|---|
| 2090855 | 0.6546 | http://www.jta.org/1937/12/29/archive/american-jewish-congress-plans-drive-on-job-discrimination |

THE AMERICAN JEWISH CONGRESS ANNOUNCED TODAY THE PUBLICATION OF A REPORT ON JEWISH NON-EMPLOYMENT AS A RESULT OF ECONOMIC DISCRIMINATION, AND ALSO A NATION-WIDE PROGRAM TO DEAL WITH THE PROBLEM. THE PROGRAM OUTLINED IN THE PAMPHLET INCLUDES ESTABLISHMENT OF BUREAUS IN COMMUNITIES WHERE THE CONGRESS HAS BRANCHES TO DEAL WITH DISCRIMINATION, DEVELOPMENT OF A PROGRAM OF EDUCATING EMPLOYERS, LEGISLATIVE CORRECTION OF AGENCY ABUSES, ELIMINATION OF QUESTIONS ON RELIGION FROM EMPLOYMENT APPLICATION BLANKS AND FORMATION OF AN ADVISORY COUNCIL OF CHRISTIAN LIBERALS. THE REPORT, THE SECOND OF ITS KIND, WAS PREPARED BY RABBI J.X. COHEN, CHAIRMAN OF THE CONGRESS'S COMMISSION ON ECONOMIC PROBLEMS, AND IS ENTITLED "HELPING TO END ECONOMIC DISCRIMINATION." IT SURVEYS CASES OF DISCRIMINATION AGAINST JEWISH JOB-SEEKERS AND WARNS THIS IS "SOCIALLY DANGEROUS." THE PAMPHLET REPORTS THAT THE PRESIDENT OF AMERICAN TELEPHONE AND TELEGRAPH COMPANY, WALTER S. GIFFORD, AT THE LAST NATIONAL CONFERENCE OF PRESIDENT OF LOCAL COMPANIES, ADVISED HIS ASSOCIATES AGAINST PERMITTING PERSONNEL MANAGERS TO REJECT JEWISH APPLICANTS BECAUSE OF PREJUDICE, AND SHORTLY THEREAFTER ONE OF THE DEPARTMENTS OF A.T.& T. " ALMOST UNPRECEDENTEDLY " ENGAGED A JEWISH APPLICANT.

| Index | mem | Link and Text |
|---|---|---|
| 799 | 0.6324 | https://www.express.co.uk/sport/othersport/356764/Greyhound-racing-results-08-11-12 |

Greyhounds jostle for position at a meet in Newcastle [] ROMFORD FANCIES 10.31 Jogadusc Jasper (1-4-2) 10.46 Selkirk Grace (4-3-1) 11.00 Forest Fella (2-4-1) 11.16 Strawberry Hall (2-5-6) 11.31 Lilys Rocket (3-1-5) 11.44 Droopys Sian (3-4-1) 11.58 Dudleys Lady (Nap) (4-6-5) 12.17 Giffys Girl (3-5-6) 12.33 Clerihan Ruso (3-1-2) 12.47 Dukes Wish (3-2-5) 1.04 Cairns Rebel (5-4-1) 1.19 Borna Karma (5-3-6) 1.33 Tolo (5-2-4) 1.49 Alrita Panther (1-2-6) CRAYFORD FANCIES 10.38 Zenas Heather (Nap) (1-5-3) 10.53 Ballybane Lena (1-5-3) 11.07 Any Wonder (1-5-3) 11.23 Whitefield Maeve (1-3-4) 11.37 Denwill Andy (1-2-4) 11.51 Mountcashel Jess (1-4-3) 12.08 Newlawn Sweep (1-6-3) 12.24 Boy Wonder Oscar (1-2-3) 12.39 Lordsbury Rollie (1-4-2) 12.57 Thirsty Rooney (1-2-3) 1.11 Little Thriller (1-2-3) 1.27 Demarco (1-2-3) NEWCASTLE FANCIES 2.24 Wansbeck Magpie (2-1-6) 2.42 Cregg Rock (6-1-5) 2.58 Steel Ranger (2-3-6) 3.18 Hillview Holly (5-6-2) 3.37 Westoe Lass (6-2-1) 3.53 Lil Carpi (Nap) (1-3-2) 4.14 Sleek Joe (2-5-1) 4.28 Blackhouse Lace (2-5-1) 4.47 A Bit Of Glory (5-6-3) 5.07 Royal Cracker (6-4-2) 5.27 Seacht Gorm (5-6-3) 5.44 Bohard Lad (1-4-2) 5.58 Dark Milly (4-5-3) 6.16 Dereks Hope (6-1-4) RESULTS HOVE: 11.11 Vivendi Katie 5-2f (1-4-3 BAGS F £20.06 TC £51.94). 11.28 Tallulah Belle 7-4f (6-3-1 £13.86 TC £61.16). 11.42 Alfatwelldigger 4-1 (2-5-1 £28.48 TC £109.66). 11.57 Clune Hondo 5-1 (4-5-6 £20.55 TC £59.03). 12.12 Que Katie 4-1 (4-5-3 £32.27 TC £88.94). 12.27 Guinness Joker 5-1 (6-1-4 £23.67 TC £67.49). 12.42 Cabra Bolt 5-2jt (6-4-1 £28.39 TC £94.26). 12.58 Calano 7-4f (6-4-3 £13.75 TC £49.05). 1.12 Westway Triangle 9-4f (5-6-2 £10.44 TC £34.96). 1.27 Jazz Diamond 3-1 (4-3-1 £16.58 TC £64.44). 1.44 Ladies Watch 5-1 (4-5-1 £43.06 TC £112.38). 1.58 Albion Star 2-1jt (6-1-2 £16.97 TC £72.66). Placer £1 stake (Races 1-6) £548.00, (Races 7-12) £340.00. MONMORE: 2.18 Ardera Jack 7-1 (2-4-1 BAGS F £40.24 TC £116.18). 2.37 Hellroad Rodge 7-2 (2-1-6 £18.22 TC £67.32). 2.57 Whitehart Wonder 3-1 (1-4-5 £14.57 TC £42.03). 3.17 Bawna Antarctic 7-2 (4-2-1 £17.06 TC £53.86). 3.38 Shinto 9-4jt (4-5-6 £20.45 TC £70.50). 3.58 Marcelino 5-1 (6-4-2 £30.55 TC £95.97). 4.18 Cooneen Fly 7-2 (3-5-2 £14.14 TC £35.51). 4.37 Rock It Ramon 3-1 (6-4-3 £12.96 TC £47.67). 4.57 Combo Eva 5-1 (5-6-3 £24.42 TC £74.53). 5.17 Liosgarbh Anna 4-1 (5-4-2 £17.90 TC £43.80). 5.33 Paidis Peata 4-1 (4-6-1 £34.64 TC £112.97). 5.47 Ardera General 4-1 (6-5-2 £15.91 TC £41.36). 6.04 Seathwaite Dave 6-1 (2-5-4 £38.06 TC £89.96). 6.18 Scooby Blue 7-2 (1-5-6 £19.05 TC £52.11). Placer £1 stake (Races 1-6) £253.00, (Races 9-14) £930.00. OXFORD: 11.03 Chapelane Lucy 7-2 (1-6-3 BAGS F £25.96 TC £71.49). 11.19 Beckys Druidin 5-1 (2-5-1 £44.44 TC £128.91). 11.34 Headford Royal 7-4jt (2-5-1 £19.85 TC £65.60). 11.48 Geneva Cove 3-1 (2-5-4 £29.24 TC £89.42). 12.04 Ring The Chief 3-1 (4-1-3 £11.87 TC £39.74). 12.18 Rock Magic 6-4f (1-5-6 £11.38 TC £45.53). 12.32 Lissycasey Jose 4-1 (6-4-2 £17.55 TC £71.66). 12.47 Brykat Lola 7-2 (5-3-4 £15.29 TC £53.59). 1.04 Ballymac Tish 5-2jt (1-6-4 £15.71 TC £44.31). 1.19 Gizmo Senor 5-1 (1-5-3 £60.34 TC £138.24). 1.33 Call It Paris 8-1 (5-3-2 £41.46 TC £98.54). 1.51 Millbank Printer 4-1 (3-4-5 £25.01 TC £53.87). Placer £1 stake (Races 1-6) £554.00, (Races 7-12) £526.00. SWINDON: 2.08 Unleash Fidel 5-2f (3-5-2 BAGS F £11.47 TC £30.89). 2.27 Rushy Dusky 10-1 (1-3-4 £42.10 TC

| Index | mem | Link and Text |
|---|---|---|
| 2085736 | 0.5755 | https://fox59.com/2016/01/28/recipe-chinese-pork-vegetable-soup-with-wonton-noodles/ |

× RECIPE: Chinese Pork & Vegetable Soup with Wonton Noodles Chinese Pork & Vegetable Soup with Wonton Noodles 1 pork tenderloin (about 1-1 ¼ pound size), cooked and cut into ½-inch cubes* 5 cups lower-sodium chicken broth 1 cup water** ¼ cup rice wine vinegar 1 tablespoon lower sodium soy sauce 1 heaping teaspoon very finely minced garlic 1 heaping teaspoon grated fresh ginger ¼ teaspoon black pepper 1 can (15 ounces) whole baby corn, drained and cut into quarters 2-4 ounces fresh snow peas, halved crosswise 2/3 cup carrots, thinly biased-sliced 2/3 cup sliced mushrooms 1/3 cup chopped green onions 8-12 wonton wrappers (3 to 3.5 inch size), cut into ½-inch strips and separated (so they don't stick together) Combine chicken broth, water, rice wine vinegar, soy sauce, garlic, ginger and black pepper in large saucepan or Dutch oven. Bring to a boil. Stir in baby corn pieces, snow peas, carrots, mushrooms, onion and wonton strips. Return to a boil, then reduce heat, cover and simmer about 3 minutes or until vegetables are crisp-tender. Stir in cooked pork cubes and heat thoroughly. *Roast pork tenderloin uncovered in 425°F oven until instant read thermometer registers 145°F (about 15 – 20 minutes roughly). Let pork tenderloin rest at least 5 minutes before cutting. Makes 4 servings (about 1 ½ cups each) Recipe by PorkBeInspired.com with adaptations by culinary dietitian & nutritionist Kim Galeaz, RDN CD

| Index | mem | Link and Text |
|---|---|---|
| 1680600 | 0.5807 | https://www.sbs.com.au/yourlanguage/arabic/en/article/2017/12/13/what-acknowledgement-country |

هذه الطريقة في التحية تم اعتمادها للمرة الأولى في البرلمان الأسترالي عام 2008 ومنذ ذلك الحين بدأ العمل بهذه التحية في معظم التجمعات العامة Language English وبذلك acknowledgement of country والخاصة في أنحاء البلاد. وفي عام 2010 افتتح البرلمان الفدرالي جلسته الأولى لأول مرة بتحية السكان الاصليين أصحاب الأرض أصبح هذا البروتوكول المتعارف عليه عند افتتاح أي دورة برلمانية. وأصبح من المتعارف عليه استخدام جملة تعريفية عن حق السكان الاصليين في المكان الذي يعقد في التجمع "I would like to acknowledge that this meeting is being held on the traditional lands of the (appropriate group) people, and pay my respect to elders both past and present." ومن اللافت أنه يتم استخدام اسم القبيلة بالتحديد لنسب التحية لأحيائهم وأمواتها لذلك من المهم على صاحب التجمع. أفراد من غير السكان الاصليين اعترافاً منهم بمكانة وأصل الأرض التي Acknowledgement of Country معرفة اسم القبيلة ويذكرها بصورة صحيحة. وعادة ما يُؤدي تحية والتي يؤديها عادة شخص كبير في السن من السكان الاصليين من المجتمع المحلي ويلقي التحية Welcome to Country يقيمون ويعملون عليها، ولكن هذه التحية تختلف عن للأحياء والأموات منهم. ويمارس السكان الاصليين هذه العادة منذ الاف السنين حيث يعتبرونها شيء بديهي أن تقوم بتحية زوارك بطريقة خاصة عندما يأتون لزيارتك للمرة الأولى، وعادة ما يرافقها احتفال اشعال الدخان Welcome to Country Hello my name is [name of speaker] a و إليكم مثالاً على خطاب Smoking ceremony representative/Elder of the [insert organisation or local Indigenous group]. I would like to begin by paying my respect to the local Indigenous people [or insert name of Indigenous people], the traditional custodians of this land where we are meeting upon today. On behalf of the traditional custodians [local Indigenous group's name] I welcome you all. خطاب على شمل احتفال أول عقد وتم 'Welcome to Country' في العليا المحكمة اقامة حفل تذكاري خلال 1999 عام ولاية نيوساوث ويلز احتفالا بمرور 175 عاماً على انشاءها. ويُذكر أن أستراليا على خلاف الجارة نيوزيلاندا و كل من كندا والولايات المتحدة فلم تعقد اتفاقية صلح مع السكان كاعتراف بسيط عن حقهم في الارض والتاريخ ولكنها ليست بديلاً عن الاتفاقية بالتأكيد ليست Acknowledgement of Country الاصليين ولذلك تعتبر تحية.

Figure 13: Text examples from RealNews with high memorization.

| Index | mem | Link and Text |
|-------|-----|---------------|
| 2074805 | 0.2835 | http://www.christianpost.com/news/texas-students-suspension-for-anti-gay-remark-reversed-94734/ |

A Texas honors student punished for saying that homosexuality was wrong has had his suspension rescinded after a meeting with the mother and her attorney. The 14-year-old, Dakota Ary, from Western Hills High School of the Fort Worth Independent School District was initially given a suspension of one day in-school and two days full. After hearing about the suspension the mother, Holly Pope, reached out to Matt Krause of the Liberty Counsel to be her son's legal representative. Matt Staver, founder and chairman of the Liberty Counsel, told The Christian Post that he believed Western Hills High made the correct decision in reversing their course of action. "The decision to rescind the suspension is the correct one. The suspension was wrong and improper," said Staver. "I applaud the student for standing up. We stood with him to resist an unjust suspension and we are pleased that suspension has been reversed." Dakota Ary was in a German language class at Western Hills when class conversation shifted to the issues of religion and homosexuality. The student reportedly said that "being a homosexual was wrong." Upon hearing his remark, the teacher proceeded to take administrative action. Ary himself is an honors student, noted for being a member of the high school's football team and a volunteer at his church. The rescinding of the suspension allows for him to play in an upcoming football game. In a publicly released statement, the Fort Worth Independent School District said that it does not comment on student matters. In keeping with this, Fort Worth ISD did not return comment to The Christian Post by press time. "As a matter of course, Fort Worth ISD does not comment on specific employee or student-related issues. Suffice it to say that we are following district policy in our review of the circumstances and any resolution will likewise be in accordance with district policy," said Fort Worth ISD. Staver also told CP that in recent years incidents like this one are only increasing as the gay rights movement continues to advance in American society. "Unfortunately, we have seen similar situations. The homosexual agenda is aggressive and intolerant to people who do not agree. Liberty Counsel will continue the right to exercise freedom of conscience and religion," said Staver. "These instances are increasing and will continue to increase unless Christians and people who love liberty stand up and resist this intolerance."

| Index | mem | Link and Text |
|-------|-----|---------------|
| 710814 | 0.1536 | http://in.reuters.com/article/idINIndia-55725620110320 |

MANAMA Bahraini security forces detained the outspoken head of the main human rights group early on Sunday, a colleague said, days after a crackdown that drove mainly Shi'ite protesters off the street. "I want to update you that they arrested Nabeel Rajab at 1.30 am and the riot police came to my home and I'm not in my home right now," Said Yousif of the Bahrain Center for Human Rights said in an email sent overnight. Rajab had spoken to media about the crackdown and the Bahrain Center for Human Rights put out statements this week accusing Bahraini forces and their Saudi and Emirati allies of "massacres" including some using Apache helicopters. Bahraini state television has said that Bahrain does not own Apache helicopters. Four protesters and three police have been confirmed killed in Wednesday's crackdown. Neither Rajab nor Yousif could be reached by phone early on Sunday, but some Twitter feeds suggested Rajab had already been released after questioning by police. Rajab's arrest would bring to 10 the number of people believed to have been arrested since the crackdown began on Wednesday. Among the others are opposition leaders who had called for the overthrow of the monarchy and doctors who had complained of excessive use of force against protesters. (Reporting by Frederik Richter, Writing by Lin Noueihed; Editing by Matthew Jones)

| Index | mem | Link and Text |
|-------|-----|---------------|
| 1263261 | 0.1533 | http://wwd.com/accessories-news/handbags/stella-dot-expands-handbag-selection-6840891/ |

Stella & Dot, the social commerce-based accessories company, will add to its offerings on April 8 with a full range of handbags. The nine-piece collection will retail from $22 for a cosmetics case to $138 for a weekender bag. Founder and chief executive officer Jessica Herrin called this the nearly 10-year-old company's "launch into a lifestyle brand," as well as a chance to double its market opportunity. "In the $30 billion accessories market in the U.S., bags and jewelry each account for approximately $10 billion," Herrin told WWD. "We see the same opportunity in bags as we did in jewelry — giving women on the go a simple way to be chic."Other styles include a $59 technology case that serves double duty as a clutch and wallet, a $39 jewelry roll, a classic $89 tote and a $128 convertible bag that has zippers up the side to let the wearer decide if she wants a sleeker or more expanded silhouette. A small handbag collection was released in 2011 — containing two cross-body and two convertible cross-body to clutch styles — but this is the first significant push for handbags geared for daytime use. With sales expected to surpass $200 million this year — business grew from $175 million to $200 million from 2011 to 2012 — Herrin projected the new category will take in about $25 million through the end of the year. In addition to the brand being carried exclusively online at stelladot.com, the brand is sold via 14,000 active stylists around the world. "We're really investing in this as a huge growth category. We just opened a dedicated in-house design studio for this in Sausalito [Calif.]," Stella & Dot chief creative officer Blythe Harris said. She added that the bags are comprised of a combination of coated canvas, waffle poly and Saffiano leather and come in ikat, snakeskin and colorful multistripe prints.Herrin added: "Broadening our accessories line by launching the new and massive category of bags allows us to make that opportunity for stylists bigger than ever before. It's a ground-floor opportunity with a proven company."

| Index | mem | Link and Text |
|-------|-----|---------------|
| 854230 | 0.1271 | https://www.nbcwashington.com/news/local/Man-Outside-Home-Fatally-Shot-By-Resident-Police-Say-466564043.html |

Police said a man was shot to death outside a Chillum, Maryland, home by the homeowner. News4's Darcy Spencer reports. A homeowner in Maryland shot and killed a man who he caught breaking into his vehicle hours after Christmas Day, police say. Sources tell News4 the shooting, which occurred in an area repeatedly hit by car break-ins, appears to be justified. Deontae Parker, of no fixed address, died after he was shot the day early Tuesday outside a home in Chillum, Maryland, Prince George's County police said. He was 32. Police said Tuesday that they were investigating what happened. "We're still working through all the details to determine what led up to the shooting," Cpl. Tyler Hunt said. A resident of a home on the 6200 block of Sligo Parkway called 911 about 2:15 a.m. Tuesday to report a shooting. Officers arrived and found a man with a gunshot wound to the upper body. He was pronounced dead. According to the preliminary investigation, detectives believe the homeowner -- who police have not identified -- shot Parker from inside his home after Parker "had broken into the homeowner's vehicle," police said in an update Wednesday morning. News4 footage shows a bullet hole in a window. Police now have video from the home's surveillance system, "which shows Parker's actions prior to the shooting." A second suspect is believed to have run from the scene before police arrived. Parker and the homeowner did not know each other, police believe. Sources told News4 the homeowner has lived there for 17 years, and that about 10 people were inside the house at the time of the shooting, including children. A neighbor who asked News4 not to identify him said his car had been broken into multiple times. The Prince George's County State's Attorney's Office will determine whether any charges should be filed against the homeowner. Anyone with information on the case is asked to call 301-772-4925. Anyone who wants to remain anonymous can call 866-411-TIPS, go to www.pgcrimesolvers.com or use the P3 Tips app.

Figure 14: Text examples from RealNews with intermediate memorization.

| Index | mem | Link and Text |
|-------|-----|---------------|
| 449808 | 0.0361 | https://www.thestreet.com/story/12179505/1/first-week-of-dlr-february-2014-options-trading.html |
| | | Investors in Digital Realty Trust, Inc. ( DLR) saw new options begin trading this week, for the February 2014 expiration. At Stock Options Channel, our YieldBoost formula has looked up and down the DLR options chain for the new February 2014 contracts and identified one put and one call contract of particular interest. The put contract at the $45.00 strike price has a current bid of $1.00. If an investor was to sell-to-open that put contract, they are committing to purchase the stock at $45.00, but will also collect the premium, putting the cost basis of the shares at $44.00 (before broker commissions). To an investor already interested in purchasing shares of DLR, that could represent an attractive alternative to paying $49.02/share today. Because the $45.00 strike represents an approximate 8% discount to the current trading price of the stock (in other words it is out-of-the-money by that percentage), there is also the possibility that the put contract would expire worthless. The current analytical data (including greeks and implied greeks) suggest the current odds of that happening are 74%. Stock Options Channel will track those odds over time to see how they change, publishing a chart of those numbers on our website under the contract detail page for this contract . Should the contract expire worthless, the premium would represent a 2.22% return on the cash commitment, or 13.98% annualized — at Stock Options Channel we call this the START SLIDESHOW : Top YieldBoost Puts of the REITs » Below is a chart showing the trailing twelve month trading history for Digital Realty Trust, Inc., and highlighting in green where the $45.00 strike is located relative to that history: |
| 1157311 | 0.0356 | https://www.thestreet.com/story/12425372/1/anf-april-4th-options-begin-trading.html |
| | | Investors in Abercrombie & Fitch Co. (ANF) saw new options become available today, for the April 4th expiration. At Stock Options Channel, our YieldBoost formula has looked up and down the ANF options chain for the new April 4th contracts and identified one put and one call contract of particular interest. The put contract at the $34.00 strike price has a current bid of $1.97. If an investor was to sell-to-open that put contract, they are committing to purchase the stock at $34.00, but will also collect the premium, putting the cost basis of the shares at $32.03 (before broker commissions). To an investor already interested in purchasing shares of ANF, that could represent an attractive alternative to paying $34.60/share today. Because the $34.00 strike represents an approximate 2% discount to the current trading price of the stock (in other words it is out-of-the-money by that percentage), there is also the possibility that the put contract would expire worthless. The current analytical data (including greeks and implied greeks) suggest the current odds of that happening are 57%. Stock Options Channel will track those odds over time to see how they change, publishing a chart of those numbers on our website under the contract detail page for this contract . Should the contract expire worthless, the premium would represent a 5.79% return on the cash commitment, or 49.23% annualized — at Stock Options Channel we call this the START SLIDESHOW : Top YieldBoost Puts of the S&P 500 » Below is a chart showing the trailing twelve month trading history for Abercrombie & Fitch Co., and highlighting in green where the $34.00 strike is located relative to that history: |
| 1551143 | 0.0708 | https://www.fool.com/investing/general/2012/08/07/national-instruments-beats-on-revenue-matches-exp.aspx |
| | | National Instruments (Nasdaq: NATI) reported earnings on Aug. 3. Here are the numbers you need to know. The 10-second takeaway For the quarter ended June 30 (Q2), National Instruments beat expectations on revenues and met expectations on earnings per share. Compared to the prior-year quarter, revenue improved significantly and GAAP earnings per share stayed the same. Margins dropped across the board. Revenue details National Instruments reported revenue of $292.3 million. The five analysts polled by S&P Capital IQ expected a top line of $279.7 million on the same basis. GAAP reported sales were 15% higher than the prior-year quarter's $253.3 million. Source: S&P Capital IQ. Quarterly periods. Dollar amounts in millions. Non-GAAP figures may vary to maintain comparability with estimates. EPS details EPS came in at $0.21. The four earnings estimates compiled by S&P Capital IQ anticipated $0.21 per share. GAAP EPS of $0.22 were the same as the prior-year quarter. Source: S&P Capital IQ. Quarterly periods. Non-GAAP figures may vary to maintain comparability with estimates. Margin details For the quarter, gross margin was 75.8%, 210 basis points worse than the prior-year quarter. Operating margin was 12.0%, 160 basis points worse than the prior-year quarter. Net margin was 9.0%, 150 basis points worse than the prior-year quarter. Looking ahead Next quarter's average estimate for revenue is $288.1 million. On the bottom line, the average EPS estimate is $0.24. Next year's average estimate for revenue is $1.14 billion. The average EPS estimate is $0.96. Investor sentiment The stock has a five-star rating (out of five) at Motley Fool CAPS, with 387 members out of 408 rating the stock outperform, and 21 members rating it underperform. Among 154 CAPS All-Star picks (recommendations by the highest-ranked CAPS members), 149 give National Instruments a green thumbs-up, and five give it a red thumbs-down. Of Wall Street recommendations tracked by S&P Capital IQ, the average opinion on National Instruments is outperform, with an average price target of $32.50. Can your portfolio provide you with enough income to last through retirement? You'll need more than National Instruments. Learn how to maximize your investment income and "Secure Your Future With 9 Rock-Solid Dividend Stocks." Click here for instant access to this free report. |
| 1257699 | 0.0643 | https://www.fool.com/investing/general/2012/02/27/merge-healthcare-goes-negative.aspx |
| | | Merge Healthcare (Nasdaq: MRGE) reported earnings on Feb. 15. Here are the numbers you need to know. The 10-second takeaway For the quarter ended Dec. 31 (Q4), Merge Healthcare met expectations on revenues and earnings per share. Compared to the prior-year quarter, revenue improved significantly and GAAP earnings per share dropped to a loss. Gross margins expanded, operating margins grew, and net margins shrank. Revenue details Merge Healthcare booked revenue of $64.1 million. The five analysts polled by S&P Capital IQ foresaw net sales of $64.0 million on the same basis. GAAP reported sales were 39% higher than the prior-year quarter's $46.2 million. Source: S&P Capital IQ. Quarterly periods. Dollar amounts in millions. Non-GAAP figures may vary to maintain comparability with estimates. EPS details Non-GAAP EPS came in at $0.04. The five earnings estimates compiled by S&P Capital IQ forecast $0.04 per share on the same basis. GAAP EPS were -$0.01 for Q4 against $0.10 per share for the prior-year quarter. Source: S&P Capital IQ. Quarterly periods. Non-GAAP figures may vary to maintain comparability with estimates. Margin details For the quarter, gross margin was 66.3%, 730 basis points better than the prior-year quarter. Operating margin was 13.7%, 500 basis points better than the prior-year quarter. Net margin was -2.0%, 2,380 basis points worse than the prior-year quarter. Looking ahead Next quarter's average estimate for revenue is $66.3 million. On the bottom line, the average EPS estimate is $0.05. Next year's average estimate for revenue is $289.2 million. The average EPS estimate is $0.29. Investor sentiment The stock has a four-star rating (out of five) at Motley Fool CAPS, with 101 members out of 121 rating the stock outperform, and 20 members rating it underperform. Among 34 CAPS All-Star picks (recommendations by the highest-ranked CAPS members), 31 give Merge Healthcare a green thumbs-up, and three give it a red thumbs-down. Of Wall Street recommendations tracked by S&P Capital IQ, the average opinion on Merge Healthcare is outperform, with an average price target of $8.13. Over the decades, small-cap stocks like Merge Healthcare have produced market-beating returns, provided they're value-priced and have solid businesses. Read about a pair of companies with a lock on their markets in "Too Small to Fail: 2 Small Caps the Government Won't Let Go Broke." Click here for instant access to this free report. |

Figure 15: Text examples from RealNews with low memorization.

| Index | mem | Link and Text |
|---|---|---|
| 324899 | 0.7560 | http://nfodb.net/view_329432_AI-War-Fleet-Command-v3.0.0.0-RIP-serial-regged-nocd-cracked-keygen.html |

ÜÜÜÜÜÜÜÜÜÜÜÜÜÜÜÜÜÜÜÜÜÜÜÜÜÜÜÜÜÜÜÜÜÜÜÜÜÜÜÜÜÜÜÜÜÜÜÜÜÜÜÜÜÜÜÜÜÜÜÜÜÜÜÜÜÜÜÜÜÜÜÜÜÜÜÜÜÜÜÜÜÜÜ
ÛßßßßßßßßßßßßßßßßßßßßßßßßßßßßßßßßßßßßßßßßßßßßßßßßßßßßßßßßßßßßßßßßßßÛÙÙÛ Ù ÛÙÙ ÛÙÛ ÛÙÛÛßßÛÙÛ ÛÙÛ ÛÙÛßßßß
ßßßßßÛÙÛ ÛÙÛÛßßßßßßßßß ÛÙÛ ÛÙÛÛßßßßßßßßßßßß ßÙ Û ÛÙÛ ÛÙÛ ÛÙÛ ÛÙÛ ÛÙÛ ÛÙÛ ÛÙÛ ßßßÛÙÛ ÛÙÛ ÛÙÛ ÛÙÛßßßÛÙÛ Û Û ÛÙÛ ÛÙÛ ÛÙÛ
ÛÙÛ ÛÙÛ ÛÙÛ ÛÙÛ ÛÙÛ ÛÙÛ ÛÙÛ ÛÙÛ Û Û ÛÙÛ ÛÙÛ ÛÙÛ ÛÙÛ ÛÙÛ ÛÙÛ ÛÙÛ ÛÙÛ ÛÙÛ ÛÙÛ Û Û ÛÙÛ ÛÙÛ ÛÙÛ ÛÙÛ
ÛÙÛ ÛÙÛ ÛÙÛ ÛÙÛ ÛÙÛ ÛÙÛ ÛÙÛ ÛÙÛ Û Û ÛÙÛ ÛÙÛ ÛÙÛ ÛÙßß ÛÙÛÛÙÛÙÛ ßßßßÛÙÛ ÛÙÛÛßÛÙÛ ÛÙÛßßß ÛÙÛ ÛÙÛ Û Û ÛÙÛ ÛÙÛ
ÛÙÛ ÛÙÛ ÛÙÛ ÛÙÛ ÛÙÛ ÛÙÛ ÛÙÛ ÛÙÛ ÛÙÛ Û Û ÛÙÛ ÛÙÛ ÛÙÛ ÛÙÛ ÛÙÛ ÛÙÛ ÛÙÛ ÛÙÛ ÛÙÛ ÛÙÛ Û Û ÛÙÛ ÛÙÛ
ÛÙÛ ÛÙÛ ÛÙÛ ÛÙÛ ÛÙÛ ÛÙÛ ÛÙÛ ÛÙÛ ÛÙÛ Û Û ßßßßßßßßß ÛÙÛ ÛÙÛ ßßßßßß ÛÙÛ ÛÙÛ ßßßßßßß ÛÙÛ ÛÙÛ ÛÙÛ ßßßßßßß ÛÙÛ ÛÙÛ Û ÛÙ
ßßßßßßßßßßßß ßßßßßßßßßß ßßßßßßß ßßß ßßßßßßßßßß ßßß ßßßßßßßßßß ßßßßßß ÜÙ
ßÜÜÜÜÜÜÜÜÜÜÜÜÜÜÜÜÜÜÜÜÜÜÜÜÜÜÜÜÜÜÜÜÜÜÜÜÜÜÜÜÜÜÜÜÜÜÜÜÜÜÜÜÜÜÜÜÜÜÜÜÜÜÜÜÜÜÜÜÜÜÜÜÜÜÜÜÜÜÜÜÜÜÜÜÜÜÜÜÜÙÜß
ßßßßßßßßßßßßßßßßßßßßßßßßßßßßßßßßßßßßßßßßßßßßßßßßßßßßßßßßßßßßßßßßßßßßßßßßßß P R E P A R E T O B E U N L E A S H E D ! ! AI War Fleet Command v3.0.0.0 RIP ÄÄÄÄÄÄÄÄÄÄÄÄÄÄÄÄÄÄÄÄÄÄÄÄÄÄÄÄÄÄÄÄÄÄÄÄÄÄÄÄÄÄÄÄÄÄÄÄÄÄÄÄÄÄÄÄÄÄÄÄÄ Release Date : 16/01/2010 Protection : Serial Release Type : RIP 23x5.00MB Company : Arcen Games ÄÄÄÄÄÄÄÄÄÄÄÄÄÄÄÄÄÄÄÄÄÄÄÄÄÄÄÄÄÄÄÄÄÄÄÄÄÄÄÄÄÄÄÄÄÄÄÄÄÄÄÄÄÄÄ Release Notes: AI War is an entirely unique large-scale RTS with aspects of TBS, tower defense, and grand strategy. It features single or cooperative play with as many as 8 humans against a pair of powerful, intelligent AIs. These AIs are driven by an AI Progress stat that players contribute to through aggressive actions such as taking control of planets and destroying key units, forcing tough decisions regarding which targets are worth capturing or destroying. Human fleets are usually several thousand ships in size or larger, with games often featuring 30,000 units at any given time across galaxies of anywhere from 10 to 120 planets. Some games have reached as many as 90,000 units! Despite this, players retain powerful control over their fleets in a macro-sense, ignoring painful micromanagement present in other games in favor of actual tactics like flanking. Meanwhile, ships themselves attempt to target enemies as intelligently as possible so the player can focus on the tasks that really matter. Scouting also plays a major role in the game, supplying an intel 'snapshot' about important capturable units, enemy fleet sizes, and planet ownership. Designed by RTS veterans and backed by strong, responsive support on the official forums and elsewhere, AI War is an exciting and refreshing experience. RIPPED: Nothing ÄÄÄÄÄÄÄÄÄÄÄÄÄÄÄÄÄÄÄÄÄÄÄÄÄÄÄÄÄÄÄÄÄÄÄÄÄÄÄÄÄÄÄÄÄÄÄÄÄÄÄÄÄÄÄÄÄÄÄÄÄÄÄ Install Notes: 1. Unpack the release with WinRAR or equivalent 2. Install with AIWarSetupv3.0.0.0.exe 3. Use the included keygen to generate a valid serial to register the game 4. Enjoy! Remember to support good video game development! ÄÄÄÄÄÄÄÄÄÄÄÄÄÄÄÄÄÄÄÄÄÄÄÄÄÄÄÄÄÄÄÄÄÄÄÄÄÄÄÄÄÄÄÄÄÄÄÄÄÄÄÄÄÄÄÄÄÄÄÄÄ Unleashed salutes all friends of the family, contacts and competition! ÄÄÄÄÄÄÄÄÄÄÄÄÄÄÄÄÄÄÄÄÄÄÄÄÄÄÄÄÄÄÄÄÄÄÄÄÄÄÄÄÄÄÄÄÄÄÄÄÄÄÄÄÄÄÄÄÄÄ " Signed but not pleased! "

| 2076649 | 0.6390 | https://webstatsdomain.org/d/vespa.bg |

Vespa: световен лидер в производството на скутери за индивидуално придвижване. Дизайнерски модели на стилно превозно средство, предназначено за масовия пазар. vespa.bg is a safe website. This information is from Google, AVG Threat Labs, McAfee SiteAdvisor, Wot. Alexa traffic rank shows the popularity of your site relative to other sites. Vespa.bg is ranked 8,030,558th in the world (among the 30 million domains). A low-numbered rank means that your website gets a lot of visitors. The top queries driving traffic to vespa.bg from search engines. Мотороллеры и Скутеры. Мототехника, - Её Описание и История Развития. Piezas, recambios y boutique de vespa primavera ¡vespizaos! Bij vespaparts vindt u alles voor uw vespa! van vespa gadgets en vespa boeken tot complete vespa scooters! wij hebben een groot assortiment onderdelen en accessoires voor alle vespa modellen! What websites are linking to Vespa.bg ? Website load time is an important factor, because Google is taking the site's loading speed into consideration in determining its ranking. Even though this will not have a big impact, it is still something we (webmasters) should really look into. The reason is pretty simple – the majority of visitors are usually in a rush and no one is fond of waiting half a century before the website finally loads its content or fails to load. At the last check on 2017-04-30, website load time was 17.32. The highest load time is 26.32, the lowest load time is 14.73, the average load time is 19.84.

| 2080878 | 0.5603 | http://repository.uin-suska.ac.id/view/creators/ASKINA=3A=3A=3A.html |

Items where Author is "ASKINA, " ASKINA, (2013) PENERAPAN STRATEGI PEMBELAJARAN AKTIF TIPE INVESTIGASI TIM UNTUK MENINGKATKAN HASIL BELAJAR SISWA PADA MATA PELAJARAN PENDIDIKAN AGAMA ISLAM DI KELAS IV SEKOLAH DASAR NEGERI 006 PASIR SIALANG KECAMATAN BANGKINANG SEBERANG KABUPATEN KAMPAR. Skripsi thesis, Universitas Islam Negeri Sultan Syarif Kasim Riau. This list was generated on Thu Apr 25 19:48:32 2019 WIB.

| 2088697 | 0.4794 | https://www.baindeterre.com/product/color-enhancing-blood-orange-and-passion-fruit-shampoo |

Color Enhancing Blood Orange & Passion Fruit Shampoo enhances red tones on red, light brown, and dark blonde hair. Apply shampoo to wet hair. Lather, leave on 3-5 minutes & rinse. Water/Aqua/Eau, Sodium Laureth Sulfate, Cocamide MIPA, Cocamidopropyl Betaine, Glycol Stearate, Citrus Aurantium Dulcis (Orange) Fruit Extract, Passiflora Edulis Fruit Extract, Ribes Nigrum (Black Currant) Fruit Extract, Lycium Barbarum Fruit Extract, Cocos Nucifera (Coconut) Oil, Argania Spinosa Kernel Oil, Gardenia Tahitensis Flower Extract, Laureth-12, Laureth-23, Guar Hydroxypropyltrimonium Chloride, PEG/PPG-4/12 Dimethicone, Polyquaternium-11, Citric Acid, Tetrasodium EDTA, PEG-150 Pentaerythrityl Tetrastearate, PEG-6 Caprylic/Capric Glycerides, Ethylhexyl Salicylate, Ethylhexyl Methoxycinnamate, Butyl Methoxydibenzoylmethane, PEG-40 Hydrogenated Castor Oil, Glycerin, PPG-26-Buteth-26, Polysorbate 20, Imidazolidinyl Urea, Limonene, Fragrance/Parfum, Orange 4 (CI 15510), Red 33 (CI 17200).

Figure 16: Text examples from C4 with high memorization.

| Index | mem | Link and Text |
|---|---|---|
| 2001601 | 0.2974 | https://www.environment.nsw.gov.au/news/quollity-selfie-spotting-from-citizen-scientists |

Citizen scientists from across Australia and the world have helped researchers count Illawarra's elusive spotted-tailed quoll, Environment Minister Gabrielle Upton said 3 March. "Six months ago we asked citizen scientists to jump online and help us analyse more than 80,000 photos taken from across the Illawarra region," Ms Upton said. "More than 300 volunteers answered the call and we can now confirm at least 20 individual quolls call this region home. "This is a great milestone for this citizen science project – in time to celebrate World Wildlife Day!" The Quollidor project is funded under the NSW Government's $100 million Saving our Species project and is providing a fascinating glimpse into the hidden world of this endangered carnivorous marsupial. The 29 motion sensor cameras in the region collect 20,000 to 30,000 images every 8 weeks that the enthusiastic citizen scientists have sorted through. "We have images of quolls posing for selfies, exploring the camera and jumping on and around the monitoring station," Ms Upton said. Quolls have huge home ranges and move across the landscape making it difficult to otherwise monitor their population and behavioural patterns. "It's great to see people being able to use technology to make a real contribution to the conservation of NSW's unique animals," Ms Upton said. The spotted-tailed quoll is the only remaining quoll species in the state. The project is ongoing and will help the NSW Government increase the resilience and size of the local quoll population.

| 2045446 | 0.2457 | http://www.beerlycoherent.com/2010/04/floreffe-blond.html |

Yesterday's trip to our local, Le Chatelain, was a welcome reprieve, and I took the opportunity to sample a new beer, Floreffe Blond. Hard to believe I haven't had everything on offer there yet, but that's Belgian pubs for you: even a run-of-the-mill one will have 30 or so types. Le Châtelain probably has closer to 60. Floreffe's dark beers, Floreffe Dubbel and Floreffe Prima Melior, are included in my list of top ten Belgian beers, so I was expecting great things with Floreffe Blond. I wasn't disappointed. Floreffe Blond is lighter than normal strong blond ales. It's lemony, summery. Fiona says it's smooth and has a bubble gum taste -- "that powdery, old-style bubble gum" -- but I'm sticking with citrus. It has very little bitterness, but it's not sweet. At 6.3% alcohol, it's not too brain-bending either. Which is just as well, because the wood treatment is doing my head in plenty enough.

| 1795710 | 0.2217 | https://www.snohomishcountywa.gov/CivicAlerts.aspx?AID=1619 |

EVERETT, Wash., April 15 – Snohomish County's Department of Planning and Development Services (PDS) today announced an on-line survey to gather public input for the Southwest Urban Growth Area Boundary Planning Study. Obtaining feedback from those who live, work, and play in Snohomish County about their quality of life and improvements they would like to see is at the heart of everything the county does. The Southwest Urban Growth Area (SWUGA) Boundary Planning Study (BPS) is no different. It is a high-level study of existing conditions and opportunities and constraints which can inform future planning choices. As part of the BPS, an online survey is available until April 26 for community members in the SWUGA BPS area, and outside the area, to give their feedback. The study area includes lands currently designated and zoned rural uses extending eastward from the SWUGA to Broadway Ave, north to Cathcart and south to the county line. It includes the Maltby Urban Growth Area (UGA) for context, as a key center. No changes to land use or boundaries are proposed for Maltby UGA or any other portion of the Boundary Planning Area as part of this study. The area has been chosen for study given it is adjacent to a fast-growing portion of the SWUGA. The study provides an opportunity to develop comprehensive information on environmental, socio-economic, infrastructure cost, and policy implications to inform future planning efforts. Using technical information and public outreach results, the study can help identify what currently works well and what doesn't in the SWUGA BPS area (e.g. roads within and beyond the study area). The study will also identify opportunities and constraints under different future growth scenarios in the currently designated-rural lands in the study area. This includes a scenario with no change to existing land use and zoning designations in the project study area. The scenarios allow the county to consider implications for environmental, social, capital/infrastructure, and fiscal conditions. The study itself is not proposing actions to change land uses, zoning or UGA boundaries. Information generated in the study will be available when future planning initiatives are developed in the county such as the 2023 Snohomish County GMA Comprehensive Plan Update. For more information on the SWUGA BPS, please visit the study webpage.

| 1130970 | 0.1636 | https://www.electionportal.org/about |

The International Republican Institute (IRI) has deployed a Long-Term Observation Mission to observe preparations for the Moldovan parliamentary elections scheduled for February 24, 2019. Long-term observers (LTOs), who were accredited by the Moldovan Central Election Commission on December 11, 2018, are be based in Edinet, Ungheni, Orhei, Anenii Noi, Hancesti, Comrat and Chisinau. IRI observers will remain on the ground to observe the pre-election preparations, Election Day and the post-election period, concluding their mission on March 26, 2019. The mission is led by Andrea Keerbs, with the assistance of three analysts who will provide in-depth analysis of the media, electoral and legal landscape. The experienced LTOs represent 11 different countries: the U.S., Poland, Canada, the U.K., France, the Philippines, Uganda, Portugal, Montenegro, Slovenia and Georgia. IRI has organized more than 200 international election observation missions around the globe, earning a reputation for impartiality and professionalism. The IRI mission to Moldova is funded by the United States Agency for International Development and will conduct its activities on a strictly independent and nonpartisan basis, without interfering in the election process and conforming to the laws of Moldova. Click here for more information on IRI's work in Moldova.

Figure 17: Text examples from C4 with intermediate memorization.

| Index | mem | Link and Text |
|---|---|---|
| 653722 | 0.0119 | http://odessahighschool.org/odessa-mo/alumni/7950336/lewis-thompson.html |
| | | This is a placeholder page for Lewis Thompson, which means this person is not currently on this site. We do suggest using the tools below to find Lewis Thompson. You are visiting the placeholder page for Lewis Thompson. This page is here because someone used our placeholder utility to look for Lewis Thompson. We created this page automatically in hopes Lewis Thompson would find it. If you are not Lewis Thompson, but are an alumni of Odessa High School Odessa, MO, register on this site for free now. |
| 1498720 | 0.0087 | http://anaheimhighschool.org/alumni/6122068/david-baldwin.html |
| | | This is a placeholder page for David Baldwin, which means this person is not currently on this site. We do suggest using the tools below to find David Baldwin. You are visiting the placeholder page for David Baldwin. This page is here because someone used our placeholder utility to look for David Baldwin. We created this page automatically in hopes David Baldwin would find it. If you are not David Baldwin, but are an alumni of Anaheim High School, register on this site for free now. |
| 1481511 | -0.0018 | http://www.showerreplacementguys.com/pa/shower-replacement-in-south-gibson/ |
| | | At Shower Replacement Guys, our experts understand the process of Shower Replacement. This is an important training that ensures that your tub or shower so that it discards the old, worn out and outdated look for a breathtaking and refreshing replacement shower doors. The process starts with removing panels, loosening the jambs, removing the bottom track, installing the side tracks and leveling the track. They will then expertly fasten the jambs, expand the holes for the panels, installing back the bottom track and the storage column. As the process nears the end stage, our experts will then fasten the jambs to the column before restoring the top track. Once the shelving units are expertly restored, the caulk frames will be put in place and finally, you will have the door back in place. Talk to our experts in Shower Replacement Guys in South Gibson, PA so that you can learn more on how all these stages can be carried out efficiently and conveniently for your Shower Replacement needs. One of the key requirements for quality services in our South Gibson, PA is that the services must be sanctioned by the authorities. We uphold provision of quality services that meet the needs and requirements of people so that they can enjoy services provided without fear of substandard services. We have complied with all the provisions of the law so that you get superior quality services. As a result, you can be sure that compare favorably with other providers of Shower Replacement services in South Gibson, PA. You can therefore approach us for our services with full confidence that you will get the quality services. We guarantee you that you will get the most trusted and reliable shower door replacement services from Shower Replacement Guys in South Gibson, PA. This replacement shower doors will transform your experience with your shower or tub. It will spot a new look replete with elegance, style and an air of sophistication never experienced before. Call us on 888-398-0573 and discover the extent to which your old tub r shower will be changed from the unpleasant look to a most appealing and attractive finish unprecedented. |
| 175431 | -0.0033 | http://www.showerpanguys.com/wi/shower-pan-in-greenbush/ |
| | | At Shower Pan Guys, you will find that our personnel have the requisite training, skills and expertise in the design of the Shower Pan liner or Shower Pan. Whether you are looking for a flexible shower liner, PVC,One liner, prefabricated shower liner, roll on membrane among other options, you will be at home with our experts to advice you on which Shower Pan is most suitable for your shower. There are some specific details that only professionals can handle. The reason why this issue is important is that you would to use shower liner for water proof environment or other flexible options available. You will require people with expertise to guide you on all these. Thank fully, we do not run out of these exceptionally skilled people to assist you in making the right decision. At Shower Pan Guys in Greenbush, WI, all your shower liner needs will be catered for in a most professional way. Our personnel will implement your proposals in an effortless way. You can be guaranteed of fully bonded, insured and licensed service in Greenbush, WI. We take our work seriously and therefore ensure that all the requirements as set by the local government are adhered to. The safety of any installation is as good as the installer. We do not like putting our clients at a disadvantage with leaking Shower Pan liner such that they cannot enjoy their bathroom experience. As per set rules and regulations, we follow the right procedure in ensuring that you get the full worth of your investment from procurement to the final installment stage. We also rank favorably with other similar service providers in Greenbush, WI. Have the confidence to approach at any time for a quality and high standard services. You will find that from Shower Pan Guys in Greenbush, WI, we help alleviate the shipping cost for our customers. For purchases worth some predetermined amount, you will receive shipping to your destination. Call us on 888-670-3340 and learn more the amount of purchases that will enable you to receive free shipping services. Shower Pan Guys in Greenbush, WI can be contacted over 888-670-3340 to have any aspect of the shower pans clarified to you in way you will understand. Issues such as procurement, quotation, design, inspection, installation, repair and maintenance will be explained to you so that you understand the procedure. You will be able to reach us through email or the online option. Our personnel will be very helpful to you so that you can make the right decision regarding your requirements. |

Figure 18: Text examples from C4 with low memorization.

| Index | mem | Link and Text |
| --- | --- | --- |
| 2091427 | 0.5121 | wikidata_id=Q4060703, version_id=13476149602771921760 |
| | | _START_ARTICLE_ Rasim Alguliyev _START_SECTION_ Monographs (4) _START_PARAGRAPH_ •Аббасов А.М., Алгулиев Р.М., Касумов В.А. Проблемы информационной безопасности в компьютерных сетях. Баку, "Элм", 1998, 235 с._NEWLINE_•Алгулиев Р.М. Теоретические основы построения виртуальных частных сетей с перестраиваемой структурой. Москва, "ИСА РАН", 1999, 104 с. _NEWLINE_•Алгулиев Р.М. Методы синтеза адаптивных систем обеспечения информационной безопасности корпоративных сетей. Москва, "УРСС", 2001, 248 с._NEWLINE_• Alguliyev R.M., Ağayev N.B., Alıquliyev R.M. Plagiatlıqla mübarizə texnologiyaları.Bakı, "İnformasiya Texnologiyaları", 2015, 165 s. |
| 2084824 | 0.4990 | wikidata_id=Q1017392, version_id=8637645553702126390 |
| | | _START_ARTICLE_ Malebranche (Divine Comedy) _START_PARAGRAPH_ The Malebranche (Italian: [ˌmaleˈbraŋke]; "Evil Claws") are the demons in the Inferno of Dante's Divine Comedy who guard Bolgia Five of the Eighth Circle (Malebolge). They figure in Cantos XXI, XXII, and XXIII. Vulgar and quarrelsome, their duty is to force the corrupt politicians (barrators) to stay under the surface of a boiling lake of pitch. |
| 2085791 | 0.4947 | wikidata_id=Q2893296, version_id=8125185859215438057 |
| | | _START_ARTICLE_ Beautiful Soup (HTML parser) _START_SECTION_ Code example _START_PARAGRAPH_ #!/usr/bin/python3# Anchor extraction from html documentfrom bs4 import BeautifulSoupfrom urllib.request import urlopenwith urlopen('https://en.wikipedia.org/wiki/Main_Page') as response:_NEWLINE_ soup = BeautifulSoup(response, 'html.parser')_NEWLINE_ for anchor in soup.find_all('a'):_NEWLINE_ print(anchor.get('href', '/')) |
| 2083026 | 0.4640 | wikidata_id=Q3559668, version_id=14703273748516274673 |
| | | _START_ARTICLE_ 320 mm Model 1934 naval gun _START_SECTION_ M1934 guns _START_PARAGRAPH_ The Conte di Cavour-class battleships Conte di Cavour and Giulio Cesare originally mounted 13 guns as built in 1915 and 1914. These ships were rebuilt between 1933 and 1937 by removing the Q triple turret amidships and replacing guns in triple A turret, twin B and X turrets, and triple Y turret. Conte di Cavour carried Vickers Mk G guns while Giulio Cesare carried Pattern T guns built by Elswick Ordnance Company. Odero Terni Orlando (OTO) re-bored the Vickers guns while the Elswick guns were re-bored by Gio. Ansaldo & C.. Maximum gun elevation was increased to 27°. _START_SECTION_ M1936 guns _START_PARAGRAPH_ The Andrea Doria-class battleships Andrea Doria and Caio Duilio originally mounted 13 guns as built in 1916 and 1915. These ships were similarly rebuilt between 1937 and 1940 by removing Q turret and replacing guns in A, B, X, and Y turrets. Pattern T guns aboard Caio Duilio were similarly rebuilt by Ansaldo while Mk G guns aboard Andrea Doria were rebuilt by OTO. Maximum gun elevation was increased to 30°. _START_SECTION_ Ammunition _START_PARAGRAPH_ The gun was loaded with four cloth bags each containing 44 kilograms (97 lb) of smokeless powder. High explosive (HE) shells weighed only 458 kilograms (1,010 lb). Anticipated useful barrel life was 150 effective full charges (EFC). |
| 2065942 | 0.4448 | wikidata_id=Q161622, version_id=10162152339355055078 |
| | | _START_ARTICLE_ 1-Ethyl-3-(3-dimethylaminopropyl)carbodiimide _START_SECTION_ Mechanism _START_PARAGRAPH_ EDC couples primary amines to carboxylic acids by creating an activated ester leaving group. First, the carbonyl of the acid attacks the carbodiimide of EDC, and there is a subsequent proton transfer. The primary amine then attacks the carbonyl carbon of the acid which forms a tetrahedral intermediate before collapsing and discharging the urea byproduct. The desired amide is obtained. |

Figure 19: Text examples from Wiki40B:en with high memorization.

| Index | mem | Link and Text |
|---|---|---|
| 1834612 | 0.2829 | wikidata_id=Q4601742, version_id=13717366317222518793 |

_START_ARTICLE_ 2003 Polish Air Force Mi-8 crash _START_PARAGRAPH_ On 4 December 2003, a Polish Mi-8 helicopter operated by the 36th Special Aviation Regiment carrying Poland's Prime Minister Leszek Miller crashed near Piaseczno, just outside Warsaw. The pilot performed an autorotation landing in a forest following the failure of both engines. The helicopter suffered extensive damage and was written off as a total loss, but despite the severity of the crash there were no fatalities. Fourteen of the 15 people on board were injured, including Leszek Miller, who had two of his thoracic vertebrae broken. _START_SECTION_ Aircraft and pilot _START_PARAGRAPH_ The helicopter was 26 years old at the time of the crash, and was close to the end of its service. It belonged to the 36th Special Aviation Regiment responsible for transporting Polish government officials. The pilot of the helicopter was major Marek Miłosz, later promoted to lieutenant colonel. _START_SECTION_ Cause _START_PARAGRAPH_ The cause of the engine failure was determined to be icing. _START_SECTION_ Trial _START_PARAGRAPH_ On 10 March 2004, Miłosz was criminally charged with violating flight safety rules and causing the crash. Specifically, the pilot was blamed for not manually turning on the deicing equipment during the flight. The pilot argued that the meteorologic information available to him at the time did not indicate that icing was likely, and hence he was not required to turn on the deicing equipment. He was consulting a thermometer during the flight, but it suffered from a systematic measurement error and hence was unable to warn of icing. In addition, during the flight an unusual thermal inversion occurred; the temperature rose with altitude, which the pilot could not have predicted._NEWLINE_In March 2010, the 6-year trial ended with a verdict of not guilty. The judge in the case noted the pilot expertly carried out the difficult autorotation landing and that the passengers survived because of his superb piloting skills. Leszek Miller declared that if he had to fly again in a helicopter in difficult atmospheric conditions, he would choose Miłosz as his pilot.

| 610 | 0.2664 | wikidata_id=Q440261, version_id=3093157553250021392 |

_START_ARTICLE_ Inga Bejer Engh _START_SECTION_ Early life and career _START_PARAGRAPH_ Engh is educated Candidate of Law. Following law school she worked with international law for the United Nations in New York City. Upon return to Norway her first job was at Drammen District Court later Oslo District Court and Asker and Bærum Police District. At 32 she began working as a prosecutor. She has prosecuted several major criminal cases including the much publicized methanol distribution case in Østfold in the early 2000s, a case where a man received ten years prison after having thrown acid on his wife, several major drugs and sexual abuse cases. She also prosecuted Tore W. Tvedt, one of the witnesses in the Breivik case, in 2002. _START_SECTION_ Personal life _START_PARAGRAPH_ Inga Bejer Engh is married and has two young boys. The youngest was born in 2009 and was very premature but has sustained no lasting medical problems. Her decision to become a lawyer was made in the gymnasium (Norwegian equivalent to high school, now called videregående).

| 173213 | 0.2531 | wikidata_id=Q54085254, version_id=10019130216726221228 |

_START_ARTICLE_ RTV silicone _START_PARAGRAPH_ RTV Silicone (Room-Temperature-Vulcanizing silicone) is a type of silicone rubber made from a two-component system (base plus curative; A+B) available in a hardness range of very soft to medium--usually from 15 to 40 Shore A. RTV silicones can be cured with a catalyst consisting of either platinum or a tin compound such as dibutyltin dilaurate. Applications include low-temperature over-molding, making molds for reproducing, and lens applications for some optically clear grades. _START_SECTION_ Applications _START_PARAGRAPH_ To produce the material, the silicone rubber is mixed with the curing agent or vulcanizing agent. Usually, the mixing ratio is a few percent. In order for the RTV silicone to reproduce the surface texture, attention is paid to the cleanliness of the original. Vacuum de-airing removes entrained air bubbles from the mixed silicone and catalyst to ensure optimal tensile strength, which affects reproduction times. In casting and mold-making, RTV silicone rubber reproduces fine details and is suitable for a variety of industrial and art-related applications including prototypes, furniture, sculpture, and architectural elements. RTV silicone rubber can be used to cast materials including wax, gypsum, low melt alloys/metals and urethane, epoxy or polyester resins (without using a release agent). A more recent innovation is the ability to 3D print RTV silicones. RTV silicones' industrial applications include aviation, aerospace, consumer electronics, and microelectronics. Some aviation and aerospace product applications are cockpit instruments, engine electronics potting, and engine gasketing. RTV silicones are used for their ability to withstand mechanical and thermal stress. _START_SECTION_ Advantages and disadvantages _START_PARAGRAPH_ RTV silicone rubber has excellent release properties compared to mold rubbers, which is especially an advantage when doing production casting of resins (polyurethane, polyester, and epoxy). No release agent is required, obviating post-production cleanup. Silicones also exhibit good chemical resistance and high-temperature resistance (205 °C, 400 °F and higher). For this reason, silicone molds are suitable for casting low-melt metals and alloys (e.g. zinc, tin, pewter, and Wood's metal). _NEWLINE_RTV silicone rubbers are, however, generally expensive--especially platinum-cure. They are also sensitive to substances (sulfur-containing modelling clay such as Plastilina, for example) that may prevent the silicone from curing (referred to as cure inhibition). Silicones are usually very thick (high viscosity), and must be vacuum degassed prior to pouring, to minimize bubble entrapment. If making a brush-on rubber mold, the curing time factor between coats is long (longer than urethanes or polysulfides, shorter than latex). Silicone components (A+B) must be mixed accurately by weight (scale required) or they do not work. Tin catalyst silicones shrink somewhat and do not have a long shelf life. Acetoxysilane-based RTV release acetic acid during the curing process, and this can attack solder joints, causing the solder to detach from the copper wire.

| 1478633 | 0.1693 | wikidata_id=Q6697935, version_id=5711381530412598390 |

_START_ARTICLE_ Lucky Landing Marina and Seaplane Base _START_SECTION_ Facilities and aircraft _START_PARAGRAPH_ Lucky Landing Marina and Seaplane Base has one landing area designated 2/20 which measures 15,000 x 4,000 ft (4,572 x 1,219 m). For the 12-month period ending July 31, 2006, the airport had 1,850 aircraft operations, an average of 154 per month: 86% general aviation and 14% air taxi.

Figure 20: Text examples from Wiki40B:en with intermediate memorization.

| Index | mem | Link and Text |
|---|---|---|
| 351056 | 0.0213 | wikidata_id=Q6724201, version_id=7759240659351916823 |
| | | _START_ARTICLE_ Mack Creek Village, Virginia _START_PARAGRAPH_ Mack Creek Village is an unincorporated community in Pulaski County, in the U.S. state of Virginia. |
| 771076 | 0.0200 | wikidata_id=Q6673673, version_id=2013268011129477536 |
| | | _START_ARTICLE_ Longdale, Virginia _START_PARAGRAPH_ Longdale is an unincorporated community in Alleghany County, Virginia, United States. |
| 727548 | 0.0115 | wikidata_id=Q5551607, version_id=15473757212411819927 |
| | | _START_ARTICLE_ German submarine U-295 _START_SECTION_ Design _START_PARAGRAPH_ German Type VIIC/41 submarines were preceded by the shorter Type VIIB submarines. U-295 had a displacement of 759 tonnes (747 long tons) when at the surface and 860 tonnes (850 long tons) while submerged. She had a total length of 67.10 m (220 ft 2 in), a pressure hull length of 50.50 m (165 ft 8 in), a beam of 6.20 m (20 ft 4 in), a height of 9.60 m (31 ft 6 in), and a draught of 4.74 m (15 ft 7 in). The submarine was powered by two Germaniawerft F46 four-stroke, six-cylinder supercharged diesel engines producing a total of 2,800 to 3,200 metric horsepower (2,060 to 2,350 kW; 2,760 to 3,160 shp) for use while surfaced, two AEG GU 460/8–27 double-acting electric motors producing a total of 750 metric horsepower (550 kW; 740 shp) for use while submerged. She had two shafts and two 1.23 m (4 ft) propellers. The boat was capable of operating at depths of up to 230 metres (750 ft)._NEWLINE_The submarine had a maximum surface speed of 17.7 knots (32.8 km/h; 20.4 mph) and a maximum submerged speed of 7.6 knots (14.1 km/h; 8.7 mph). When submerged, the boat could operate for 80 nautical miles (150 km; 92 mi) at 4 knots (7.4 km/h; 4.6 mph); when surfaced, she could travel 8,500 nautical miles (15,700 km; 9,800 mi) at 10 knots (19 km/h; 12 mph). U-295 was fitted with five 53.3 cm (21 in) torpedo tubes (four fitted at the bow and one at the stern), fourteen torpedoes, one 8.8 cm (3.46 in) SK C/35 naval gun, (220 rounds), one 3.7 cm (1.5 in) Flak M42 and two 2 cm (0.79 in) C/30 anti-aircraft guns. The boat had a complement of between forty-four and sixty. _START_SECTION_ Service history _START_PARAGRAPH_ The boat's service life began with training with the 8th U-boat Flotilla in October 1943. She was then transferred to the 9th flotilla for operations on 1 August 1944. She was reassigned to the 13th flotilla on 1 October and moved again to the 14th flotilla on 1 April 1945. _START_SECTION_ 1st and 2nd patrols _START_PARAGRAPH_ U-295's first patrol was uneventful._NEWLINE_She then embarked on a series of short journeys between Bergen, Kristiansand, Stavanger and Trondheim._NEWLINE_Her second foray, between Trondheim and Harstad was the most successful. She damaged the British frigate HMS Mounsey east northeast of Murmansk on 2 November 1944. _START_SECTION_ 3rd and 4th patrols _START_PARAGRAPH_ The submarine's third sortie took her into the Barents and Norwegian Seas. She returned to Harstad on 18 December 1944._NEWLINE_Her fourth patrol started in Harstad and finished in Narvik. She had spent three days off Murmansk, to no avail. _START_SECTION_ 5th patrol _START_PARAGRAPH_ Her fifth effort was just as barren, even though it was longer. _START_SECTION_ 6th patrol and fate _START_PARAGRAPH_ The boat departed Narvik on 15 April 1945. Her route took her once again to the Barents Sea. She returned to the Nordic port on 7 May._NEWLINE_She was then moved to Skjomenfjord on 12 May 1945 and in accordance with the surrender terms, she was transferred to Loch Eriboll in northern Scotland for Operation Deadlight on the 19th. She was sunk on 17 December by the guns of ORP Blysawica. |
| 799605 | 0.0086 | wikidata_id=Q2024611, version_id=3681129031547882940 |
| | | _START_ARTICLE_ German submarine U-528 _START_SECTION_ Design _START_PARAGRAPH_ German Type IXC/40 submarines were slightly larger than the original Type IXCs. U-528 had a displacement of 1,144 tonnes (1,126 long tons) when at the surface and 1,257 tonnes (1,237 long tons) while submerged. The U-boat had a total length of 76.76 m (251 ft 10 in), a pressure hull length of 58.75 m (192 ft 9 in), a beam of 6.86 m (22 ft 6 in), a height of 9.60 m (31 ft 6 in), and a draught of 4.67 m (15 ft 4 in). The submarine was powered by two MAN M 9 V 40/46 supercharged four-stroke, nine-cylinder diesel engines producing a total of 4,400 metric horsepower (3,240 kW; 4,340 shp) for use while surfaced, two Siemens-Schuckert 2 GU 345/34 double-acting electric motors producing a total of 1,000 shaft horsepower (1,010 PS; 750 kW) for use while submerged. She had two shafts and two 1.92 m (6 ft) propellers. The boat was capable of operating at depths of up to 230 metres (750 ft)._NEWLINE_The submarine had a maximum surface speed of 18.3 knots (33.9 km/h; 21.1 mph) and a maximum submerged speed of 7.3 knots (13.5 km/h; 8.4 mph). When submerged, the boat could operate for 63 nautical miles (117 km; 72 mi) at 4 knots (7.4 km/h; 4.6 mph); when surfaced, she could travel 13,850 nautical miles (25,650 km; 15,940 mi) at 10 knots (19 km/h; 12 mph). U-528 was fitted with six 53.3 cm (21 in) torpedo tubes (four fitted at the bow and two at the stern), 22 torpedoes, one 10.5 cm (4.13 in) SK C/32 naval gun, 180 rounds, and a 3.7 cm (1.5 in) SK C/30 as well as a 2 cm (0.79 in) C/30 anti-aircraft gun. The boat had a complement of forty-eight. _START_SECTION_ Patrol and loss _START_PARAGRAPH_ The boat departed Kiel on 15 April 1943, moved through the North Sea, negotiated the gap between Iceland and the Faroe Islands and entered the Atlantic Ocean. There, she was intercepted by the escorts of Convoy ON (S) 5 and damaged. She was sunk on her way to the French Atlantic bases._NEWLINE_U-528 was 'destroyed' on 11 May 1943 southwest of Ireland by depth charges dropped from a Handley Page Halifax of No. 58 Squadron RAF and the British sloop HMS Fleetwood._NEWLINE_Eleven men went down with the U-boat; there were 45 survivors. |

Figure 21: Text examples from Wiki40B:en with low memorization.

| Index | Estim. | Link and Text |
|---|---|---|
| valid 1334662 | infl 0.3780 | http://www.metro.us/news/archaeologists-vs-robbers-in-israel-s-race-to-find-ancient-scrolls/kZgpfb---HhAktviTsXh_U3HpnRICjA |

By Ari Rabinovitch TZEELIM VALLEY, Israel (Reuters) - The disposable paper face masks offer little protection from the clouds of dust that fill the cliffside cave where Israeli archaeologists are wrapping up the largest excavation in the Judean desert of the past half-century. Clipped into safety harnesses, volunteers stand at the cave opening, 250 meters (820 feet) above a dry river bed that leads to the lowest spot on earth, the Dead Sea. They sift through an endless supply of dirt-filled buckets, and the dust they throw in the air reaches the far corners of the cave where a dozen workers crawling on hands and knees can't help but cough. The three-week excavation was the first part of a national campaign to recover as many artefacts as possible, particularly scrolls, left behind by Jewish rebels who hid in the desert some 2,000 years ago, before they are snatched up by antiquity robbers. "These looters that operate in the area are experts at finding scrolls. We go after them, look for what they are looking for and try to catch them," said Guy Fitoussi, head of the Israel Antiquities Authority robbery prevention unit in southern Israel. "This is the game. Like cat and mouse." The Dead Sea Scrolls, a collection of ancient texts written on papyrus and parchment, have already been rescued by scholars. They are among the earliest texts written in the Hebrew language and are on display in the Israel Museum in Jerusalem as a national treasure. Now Israel wants to uncover whatever may remain in the desert hideouts before it is destroyed or ends up on the black market. For a Wider Image photo essay on the excavation click: http://reut.rs/25zc4ZK PISTOL-PACKING ARCHAEOLOGIST According to Israeli law, all relics found on land or at sea belong to the state. Fitoussi, a pistol-packing archaeologist with authority to arrest looters, and his team catch about 100 of them each year. Most are fined; some are sent to jail. In 2014, they arrested six people who were plundering this particular cavern, known as the Cave of Skulls, where seven skulls had been found from Jews of the Bar Kokhba rebellion against Rome in the 2nd century. That raid, Fitoussi said, helped spur the multi-year government-backed excavation program and focused their initial efforts at this site, about a two-hour drive southeast from Jerusalem. To access the cave, diggers don climbing gear and descend 20 minutes from their campsite along a steep path that hugs the rocky cliff. Inside, the grotto expands 160 square meters (1,720 square feet), including a number of cramped tunnels that extend deep into the mountain. The limestone walls of the dry desert cave are perfect for preservation, said Uri Davidovich, an archaeologist from Tel Aviv University who was one of the dig's directors. One 19-year-old volunteer, working flat on her belly in a dark crawl space, dirt mixed with sweat covering her face and digging with her fingers, unearthed a thin, 25 centimeter (10 inch)-long rope that most likely was used by the Bar Kokhba rebels. A rope this length was a rare discovery, Davidovich said. They haven't found any scrolls yet, he said, but the artefacts found in this cave, and countless others nearby, will provide historians rare insight into how people lived 2,000 to 8,000 years ago. (Editing by Jeffrey Heller/Jeremy Gaunt)

| Index | Estim. | Link and Text |
|---|---|---|
| train 2077314 | men 0.3780 | https://www.metro.us/news/archaeologists-vs-robbers-in-israel-s-race-to-find-ancient-scrolls/kZgpfb---HhAktviTsXh_U3HpnRICjA |

By Ari Rabinovitch TZEELIM VALLEY, Israel (Reuters) - The disposable paper face masks offer little protection from the clouds of dust that fill the cliffside cave where Israeli archaeologists are wrapping up the largest excavation in the Judean desert of the past half-century. Clipped into safety harnesses, volunteers stand at the cave opening, 250 meters (820 feet) above a dry river bed that leads to the lowest spot on earth, the Dead Sea. They sift through an endless supply of dirt-filled buckets, and the dust they throw in the air reaches the far corners of the cave where a dozen workers crawling on hands and knees can't help but cough. The three-week excavation was the first part of a national campaign to recover as many artefacts as possible, particularly scrolls, left behind by Jewish rebels who hid in the desert some 2,000 years ago, before they are snatched up by antiquity robbers. "These looters that operate in the area are experts at finding scrolls. We go after them, look for what they are looking for and try to catch them," said Guy Fitoussi, head of the Israel Antiquities Authority robbery prevention unit in southern Israel. "This is the game. Like cat and mouse." The Dead Sea Scrolls, a collection of ancient texts written on papyrus and parchment, have already been rescued by scholars. They are among the earliest texts written in the Hebrew language and are on display in the Israel Museum in Jerusalem as a national treasure. Now Israel wants to uncover whatever may remain in the desert hideouts before it is destroyed or ends up on the black market. For a Wider Image photo essay on the excavation click: http://reut.rs/25zc4ZK PISTOL-PACKING ARCHAEOLOGIST According to Israeli law, all relics found on land or at sea belong to the state. Fitoussi, a pistol-packing archaeologist with authority to arrest looters, and his team catch about 100 of them each year. Most are fined; some are sent to jail. In 2014, they arrested six people who were plundering this particular cavern, known as the Cave of Skulls, where seven skulls had been found from Jews of the Bar Kokhba rebellion against Rome in the 2nd century. That raid, Fitoussi said, helped spur the multi-year government-backed excavation program and focused their initial efforts at this site, about a two-hour drive southeast from Jerusalem. To access the cave, diggers don climbing gear and descend 20 minutes from their campsite along a steep path that hugs the rocky cliff. Inside, the grotto expands 160 square meters (1,720 square feet), including a number of cramped tunnels that extend deep into the mountain. The limestone walls of the dry desert cave are perfect for preservation, said Uri Davidovich, an archaeologist from Tel Aviv University who was one of the dig's directors. One 19-year-old volunteer, working flat on her belly in a dark crawl space, dirt mixed with sweat covering her face and digging with her fingers, unearthed a thin, 25 centimeter (10 inch)-long rope that most likely was used by the Bar Kokhba rebels. A rope this length was a rare discovery, Davidovich said. They haven't found any scrolls yet, he said, but the artefacts found in this cave, and countless others nearby, will provide historians rare insight into how people lived 2,000 to 8,000 years ago. (Editing by Jeffrey Heller/Jeremy Gaunt)

Figure 22: Validation / training example pair from RealNews with high influence. Red / green highlighted text indicate deleted / added text in the training example comparing to the corresponding validation example, generated using Python `difflib`.

| Index | Estim. | Link and Text |
|-------|--------|---------------|
| valid 838341 | infl 0.1209 | https://www.dailymail.co.uk/wires/ap/article-3670409/Retired-pope-offers-assessment-papacy.html |
| | | VATICAN CITY (AP) — Emeritus Pope Benedict XVI is offering a first-ever papal assessment of his own pontificate in a book that recounts his decision to resign, his surprise at his successor and his attempts to dismantle what he calls the Vatican's "gay lobby." "Benedict XVI: The Final Conversations," is due out in September, the latest book-length interview that Benedict has conducted with German journalist Peter Seewald. Italian daily Corriere della Sera, which has the book's newspaper rights, provided a brief overview Friday. Corriere said Benedict recounts in the book that he decided to announce his resignation in Latin because he feared making a mistake in Italian. He recalls his "surprise" that Jorge Mario Bergoglio was elected pope and his "joy" at seeing Pope Francis mingle with crowds. Benedict also claims to have dismantled a group of four or five gay prelates, dubbed the "gay lobby" by the Italian media, who exercised power and influence in the Vatican. The existence of this group of gay prelates — who purportedly used blackmail to promote and preserve their interests — has been mythologized in Italian media. Soon after he was elected pope and was asked about the so-called "gay lobby," Francis quipped that he had yet to encounter any priest who had "gay" written on his business card. That said, just this week a gay monsignor who was fired from the Vatican and suspended as a priest after he came out, boyfriend by his side, published a book about his experiences as a gay official in the Vatican's doctrine office. In "The First Stone," Polish-born Krzysztof Charamsa recounts the absolute "obsession" with homosexuality in the halls of the Holy See. He details the "hypocrisy" of its functionaries who profess a celibate life but live quite another, and writes that it was enough to destroy someone's Vatican career by simply spreading gossip that he was gay. ___ |
| train 614881 | men 0.1650 | https://www.deseretnews.com/article/765687763/Retired-pope-offers-first-ever-assessment-of-his-own-papacy.html |
| | | VATICAN CITY (AP)— Emeritus Pope Benedict XVI is offering a first-ever papal assessment of his own pontificate in a book that recounts his decision to resign, his surprise at his successor and his attempts to dismantle what he calls the Vatican's "gay lobby." "Benedict XVI: The Final Conversations," is due out in September, the latest book-length interview that Benedict has conducted with German journalist Peter Seewald. Italian daily Corriere della Sera, which has the book's newspaper rights, provided a brief overview Friday. Corriere said Benedict recounts in the book that he decided to announce his resignation in Latin because he feared making a mistake in Italian. He recalls his "surprise" that Jorge Mario Bergoglio was elected pope and his "joy" at seeing Pope Francis mingle with crowds. Benedict also claims to have dismantled a group of four or five gay prelates, dubbed the "gay lobby" by the Italian media, who exercised power and influence in the Vatican. The existence of this group of gay prelates — who purportedly used blackmail to promote and preserve their interests — has been mythologized in Italian media. Soon after he was elected pope and was asked about the so-called "gay lobby," Francis quipped that he had yet to encounter any priest who had "gay" written on his business card. That said, just this week a gay monsignor who was fired from the Vatican and suspended as a priest after he came out, boyfriend by his side, published a book about his experiences as a gay official in the Vatican's doctrine office. In "The First Stone," Polish-born Krzysztof Charamsa recounts the absolute "obsession" with homosexuality in the halls of the Holy See. He details the "hypocrisy" of its functionaries who profess a celibate life but live quite another, and writes that it was enough to destroy someone's Vatican career by simply spreading gossip that he was gay. —Follow Nicole Winfield at www.twitter.com/nwinfield |

Figure 23: Validation / training example pair from RealNews with relatively high influence. Red / green highlighted text indicate deleted / added text in the training example comparing to the corresponding validation example, generated using Python `difflib`.

| Index | Estim. | Link and Text |
|---|---|---|
| valid 682107 | infl 0.0673 | https://www.pasadenastarnews.com/2009/09/16/ducks-ushering-in-a-new-era-for-2009-10-season/ |

ANAHEIM – On a night when Francois Beauchemin had two assists in Toronto, and Chris Pronger blocked six shots in Detroit, 13,869 Ducks fans might have been lost without their programs on Wednesday night. It's only the first game of the preseason, but a new era has clearly begun. A group of mostly newcomers in Ducks uniforms beat Phoenix 3-2 in a shootout on Wednesday at Honda Center. A familiar face made the biggest impact, however, as Bobby Ryan scored two goals. Ryan also scored two goals in the Ducks' preseason opener last year, when he was trying to make the team's opening-day roster as a rookie. His bid failed and Ryan was forced to start the season in the American Hockey League, mostly due to salary-cap constraints. The circumstances could not be much different this year. Ryan finished the season as a Calder Trophy candidate, and began this season wearing a temporary alternate captain's "A" on his jersey along with Joffrey Lupul and Sheldon Brookbank. Nick Boynton, a 30-year-old defenseman signed in the offseason, was the eldest player in uniform. Of the 20 players chosen to play, only goaltender Jonas Hiller and center Ryan Carter spent all of last season with the Ducks. Hiller started in goal and looked sharp, stopping 18 of the 19 shots he faced. Timo Pielmeier, acquired in the trade that sent Kent Huskins and Travis Moen to San Jose, came in halfway through the third period and stopped 22 of 23 shots before the game went into overtime. A phantom interference penalty sent Ducks defenseman Steve Eminger to the box, and set up a power-play goal by the Coyotes' Lauri Korpikowski with 20.2 left in the period. Korpikowski deflected a long-range shot by Keith Yandle, and Hiller did well just to get his glove on it. Ryan got the goal back early in the second period, poking in a rebound of a Dan Sexton shot past former Kings goalie Jason LaBarbera. With 1:08 elapsed in the third period, Ryan made it 2-1 on a breakaway shot that caromed in off a Phoenix stick. Phoenix tied the game at 2 at 12:30 of the third period, when Sheldon Brookbank made a costly turnover in front of his own net. Chad Kolarik batted down Brookbank's pass just a few feet in front of Pielmeier and tucked the puck in behind the goaltender.

| Index | Estim. | Link and Text |
|---|---|---|
| train 494435 | men 0.1439 | https://www.dailybreeze.com/2009/09/17/newcomers-help-lead-ducks-to-exhibition-win/ |

Beauchemin had two assists in Toronto and Chris Pronger blocked six shots in Detroit, and Chris Pronger blocked six shots in Detroit,the 13,869 Ducks fans might h' fans who showed at at Honda Center Wednesday needed programs to identify the players on their fave been lost without their programs on Wednesday nightorite team. It's only the first game of the preseason, but a new era has clearly begun. A group of mostly newcomers in Ducks uniforms beat Phoenix 3-2 in a shootout on Wednesday at Honda Center. A, although a familiar face made the biggest impact, however, as Bobby Ryan scored two goals in regulation and another in the shootout. Ryan also scored two goals in the Ducks' preseason opener last year, when he was trying to make the team's opening-day roster as a rookie. His bid failed and Ryan was forced to start the season in the American Hockey League, mostly due to salary-cap constraints. The circumstances could not be much different this yearore different now. Ryan finished thelast season as a Calder Trophy candidate, and began this season wearing a temporary alternate captain's "A" on his jersey along with Joffrey Lupul and Sheldon Brookbank. Nick Boynton, a 30-year-old defenseman signed in the offseason, was the eldest player in uniform. Of the 20 players chosen to play, only goaltender Jonas Hiller and center Ryan Carter spent all of last season with the Ducks. Hiller started in goal and looked sharp, stopping 18 of the 19 shots he faced. Timo Pielmeier, acquired in the trade that sent Kent Huskins and Travis Moen to San Jose, came in halfway through the third period and stopped 27 of 2 of 23 shots before the game went into overtime8. A phantom interference penalty sent Ducks defenseman Steve Eminger to the box, and set up a power-play goal by the Coyotes' Lauri Korpikowski with 20.2 left in the period. Korpikowski deflected a long-range shot by Keith Yandle, and Hiller did well just to get his glove on itopening period. Ryan got the goal back early in the second period, poking in a rebound of a Dan Sexton shot past former Kings goalie Jason LaBarbera at the 3:09 mark. With 1:08 elapsed in the third period, Ryan made it 2-1 on a breakaway shot that caromed in off a Phoenix stick. Phoenix tied the game at 2 at 12:30 of the third period, when Sheldon Brookbank made a costly turnover in front of his own net. Chad Kolarik batted down Brookbank's pass just a few feet in front of Pielmeier and tucked the puck in behind the goaltender. Chad KolarikIn the shootout, Pielmeier stopped batted down Brookoth shots he faced, while Ryan and Joffrey Lupul bankeat Montoya on the Ducks's pass just a few feet in front of Pielmeier and tucked the puck in behind the goaltender two attempts.

Figure 24: Validation / training example pair from RealNews with intermediate influence. Red / green highlighted text indicate deleted / added text in the training example comparing to the corresponding validation example, generated using Python `difflib`.

| Index | Estim. | Link and Text |
|-------|--------|---------------|
| valid 1165799 | infl 0.0360 | https://kdvr.com/2019/03/14/more-than-70000-pounds-of-butterball-turkeys-recalled-because-of-potential-salmonella/ |

× More than 70,000 pounds of Butterball turkey recalled because of potential salmonella WASHINGTON — The U.S. Department of Agriculture's Food Safety and Inspections services announced on Wednesday that approximately 78,164 pounds of raw ground Butterball turkey might be contaminated with Salmonella. Product numbers "EST. P-7345" inside the USDA mark of inspection are subject to recall. They were shipped to nationwide retail and institutional locations. RELATED: View the full recall FSIS, the Centers for Disease Control and Prevention and its public health partners are investigating a multistate outbreak of salmonella schwarzengurn involving five case-patients from two states. According to the USDA, Wisconsin collected three intact Butterball brand ground turkey samples from a residence where four of the case-patients live. Signs of salmonellosis, also known as salmonella foodborne illnesses, appear within one to three days after consumption of contaminated products. They include diarrhea, abdominal cramps and fever. The illness lasts four to seven days. According to the USDA, most people recover without treatment, however, they suggest that if diarrhea is severe the person might need to be contaminated. They also suggest those with weakened immune systems are more susceptible to illness. Consumers with food safety questions can "Ask Karen," the FSIS virtual representative available 24 hours a day at AskKaren.gov or via smartphone at m.askkaren.gov. The toll-free USDA Meat and Poultry Hotline 1-888-674-6854 is available in English and Spanish and can be reached from 8 a.m. to 4 p.m. MDT Monday through Friday. Recorded food safety messages are available 24 hours a day. The online Electronic Consumer Complaint Monitoring System can be accessed 24 hours a day. RELATED: USDA RECALLS

| Index | Estim. | Link and Text |
|-------|--------|---------------|
| train 1571976 | men 0.2094 | https://wreg.com/2019/03/14/butterball-recalls-nearly-80000-pounds-of-turkey-after-salmonella-cases/ |

× More than 7Butterball recalls nearly 80,000 pounds of Butterball turkey recalled because of potential salmonellaafter salmonella cases WASHINGTON — The U.S. Department of Agriculture's Food Safety and Inspections services announced on announced Wednesday that approximately 78,164 pounds of raw ground Butterball turkey might be contaminated with Salmonella. Product numbers "EST. P-7345" inside the USDA mark of inspection are subject to recall. The raw ground turkey was produced on July 7, 2018. They followere shipped to nationing products under recall wide retail and institutional locations.ere shipped to nationwide retail and institutional locations: 48-oz. plastic wrapped tray containing "BUTTERBALL everyday Fresh Ground Turkey WITH NATURAL FLAVORING (85% LEAN/15% FAT)" with sell or freeze by date of 7/26/18, lot code 8188, and UPC codes 22655-71555 or 22655-71557 represented on the label. 48-oz. plastic wrapped tray containing "BUTTERBALL everyday Fresh Ground Turkey WITH NATURAL FLAVORING (93% LEAN/7% FAT)" with sell or freeze by date of 7/26/18, lot code 8188 and UPC code 22655-71556 represented on the label. 16-oz. plastic wrapped tray containing "BUTTERBALL everyday Fresh Ground Turkey WITH NATURAL FLAVORING (85% LEAN/15% FAT)" with sell or freeze by date of 7/26/18, lot code 8188 and UPC code 22655-71546 represented on the label. 16-oz. plastic wrapped tray containing "BUTTERBALL everyday Fresh Ground Turkey WITH NATURAL FLAVORING (93% LEAN/7% FAT)" with sell or freeze by date of 7/26/18, lot code 8188 and UPC codes 22655-71547 or 22655-71561 represented on the label 48-oz. plastic wrapped tray containing "Kroger GROUND TURKEY FRESH 85% LATEAN – 15% FAT" with sell or freeze by date of 7/26/18, lot code 8188, and UPC code 111141097993 represented on the label. 48-oz. plastic wrapped tray containing "FOOD: Vie LION 15% fat ground turkey wthe full recallith natural flavorings" with sell or freeze by date of 7/26/18, lot code 8188 and UPC code 3582609294 represented on the label. See images of the recalled product labels here. FSIS, the Centers for Disease Control and Prevention and its public health partners are investigating a multistate outbreak of salmonella sSchwarzengurnrund involving five case-patients from two states. According to the USDA, Wisconsin officials collected three intact Butterball brand ground turkey samples from a residence where four of the case-patients live. Signs of salmonellosis, also known as salmonella foodborne illnesses, appear within one to three days after consumption of contaminated products. They include diarrhea, abdominal cramps and fever. The illness lasts four to seven days. According to the USDA, most people recover without treatment, however, they suggest that if diarrhea is severe the person might need to be contaminated. They also suggest those with weakened immune systems are more susceptible to illness. Consumers with food safety questions can "Ask Karen," the FSIS virtual representative available 24 hours a day at AskKaren.gov or via smartphone at m.askkaren.gov. The toll-free USDA Meat and Poultry Hotline 1-888-674-6854 is available in English and Spanish and can be reached from 8 a.m. to 4 p.m. MDT Monday through Friday. Recorded food safety messages are available 24 hours a day. The online Electronic Consumer Complaint Monitoring System can be accessed 24 hours a day. RELATED: USDA RECALLS

Figure 25: Validation / training example pair from RealNews with relatively low influence. Red / green highlighted text indicate deleted / added text in the training example comparing to the corresponding validation example, generated using Python difflib.

| Index | Estim. | Link and Text |
|---|---|---|
| valid 1556405 | infl 0.0155 | https://www.deccanchronicle.com/nation/current-affairs/280218/after-darwin-union-min-targets-newtons-theory-says-mantras-coded.html |

Earlier in January, Satyapal Singh had claimed that Charles Darwin's theory of evolution of man was 'scientifically wrong' and it needed to be changed in school and college curriculum. (Photo: File) New Delhi: Minister of State for Human Resource Development, Satyapal Singh speaking at a meeting of the Central Advisory Board of Education (CABE) on January 15 and 16 said that mantras codified the 'laws of motion' much before they were framed by Issac Newton. He also suggested that Vaastu compliance of educational buildings was important for learning, according to a report in Hindustan Times. "There are mantras which codified 'laws of motion' much before it was discovered by the Newton. Hence it is essential that traditional knowledge must be incorporated in our curriculum," Singh was quoted as saying by the minutes of the meeting of the government's highest advisory body for policymaking in education. Earlier in January, Satyapal Singh had claimed that Charles Darwin's theory of evolution of man was "scientifically wrong" and it needed to be changed in school and college curriculum. Singh said our ancestors have nowhere mentioned that they saw an ape turning into a man.

| train 1903637 | men 0.1927 | https://www.deseretnews.com/article/29058/MUBARAK-SAYS-HE-WILL-VISIT-ISRAEL-IF-IT-SHOWS-FLEXIBILITY.html |

President Hosni Mubarak said he is willing to visit Israel if the Jewish state shows flexibility in promoting peace and agrees to take part in an international Middle Earlier in Januarst peace conference, according to an interview published Saturday. "I am still saying that I am ready to travel to Israel provided that this would lead to real progress on the road to solving the Middle East problem, Sat" Mubarak told the Egyapal Singh had claiptian newspaper Al Massa. The remed that Charlesarks represented the second time in less than a week that Mubarak had expressed readiness to make a peace visit to Israel. In an interview Darwinec. 24 with the Kuwaiti newspaper l,Anba, Mubarak said he was prepared to travel to Israel if the visit would promote a solution of the Palestinian problem and a just Middle East peace. The offer was initially embraced by Israel, but later two of Mubarak's theorsenior foreign policy of eaides said the volution of man was isit was contingent on the Jewish state opening negotiations with the Palestine Liberation Organization. Israel rejects the PLO as a negotiating partner despite the United States'scientifically wrong' and it needed to be changed in school and college curriculum decision Dec. (Photo: File) New Delhi: Minister of State for Human Resource Development, Satyapal Singh speaking at a meeting of the Central Advisory Board of Education (CABE) on January 15 and 16 said that mantras codified the 'laws of motion' much before they were framed by Issac Newton4 to open a dialogue with the group. He also suggested that Vaastu compliance of educational buildings was important for learning, according to a report in Hindustan Times. "There are mantras which codified 'laws of motion' much before it was discovered by the Newton. Hence it is essential that traditional knowledge must be incorporated in our curriculum," Singh was quoted as saying by the minutes of the meeting of the government's highest advisory body for policymaking in education. Earlier in January, Satyapal Singh had claimed that Charles Darwin's theory of evolution of man was "scientifically wrong" and it needed to be changed in school and college curriculum. Singh said our ancestors have nowhere mentioned that they saw an ape turning into a man.

Figure 26: Validation / training example pair from RealNews with low influence. Red / green highlighted text indicate deleted / added text in the training example comparing to the corresponding validation example, generated using Python `difflib`.

| Index | Estim. | Link and Text |
|---|---|---|
| valid
262420 | infl
0.3755 | https://expertappliancerepairorangecounty.com/appliances/range-repair-orange-county-service?z=92811 |
| | | Range Repair 92811 Expert repairs and services all types ranges, whether you are in need of electric range repair or gas range repair. If your range is having problems like the range surface Element won't work, range burner has spark problems, range surface element won't turn off, range burners spark all the time. Range Repair Service will put you right back where you need to be. Our range repair technicians carry most range parts. Expert Appliance Repair 92811 will have your range repaired or serviced in no time flat. We will have your range up in running in no time and you back to cooking. |
| train
2059217 | men
0.4406 | https://expertappliancerepairorangecounty.com/appliances/range-repair-orange-county-service?z=92871 |
| | | Range Repair 928**71** Expert repairs and services all types ranges, whether you are in need of electric range repair or gas range repair. If your range is having problems like the range surface Element won't work, range burner has spark problems, range surface element won't turn off, range burners spark all the time. Range Repair Service will put you right back where you need to be. Our range repair technicians carry most range parts. Expert Appliance Repair 928**71** will have your range repaired or serviced in no time flat. We will have your range up in running in no time and you back to cooking. |
| | | |
| valid
274413 | infl
0.1267 | http://vannafrosoni.featuredwebsite.com/free_home_valuation.asp |
| | | Want to know what your home in ENGLISHTOWN is worth? I'll provide you with a FREE home evaluation at no obligation. Simply fill in your information below and I'll get back to you shortly with a detailed report of comparable homes in your area that have recently sold or are currently for sale. |
| train
28486 | mem
0.2073 | http://cnyrealestate.com/free_home_valuation.asp |
| | | Want to know what your home in ENGLI~~SH~~~~TOWN~~**yracuse** is worth? I'll provide you with a FREE home evaluation at no obligation. Simply fill in your information below and I'll get back to you shortly with a detailed report of comparable homes in your area that have recently sold or are currently for sale. |
| | | |
| valid
27342 | infl
0.0603 | https://www.sciway.net/hotels/beaches.html |
| | | South Carolina SC Hotels SC Beach Hotels Also see: SC Coastal Hotels \| US Hotels Want to stay at the beach? Below is an accommodation list for all the beaches in South Carolina – plus descriptions of what those beaches are like. Find hotels located on SC beaches, inlcuding Myrtle Beach, shown here. Our list starts at South Carolina's border with North Carolina and heads south towards Savannah, Georgia. |
| train
1520782 | mem
0.2299 | https://www.sciway.net/hotels/coast.html |
| | | South Carolina SC Hotels SC **Coastal Hotels Also see: SC** Beach Hotels ~~Also see: SC Coastal~~ \| **US** Hotels ~~\| US Hotels~~ Want to stay at ~~the beach~~**long the coast**? Below is an accommodation list for all the ~~beache~~**coastal area**s in South Carolina – plus descriptions of what those ~~beache~~**area**s are like. Find hotels located ~~on SC beaches~~**along the SC coast**, in**c**luding ~~Myrtle Beach~~**Charleston**, shown here. ~~Our list starts at South Carolina's border with North Carolina and heads south towards Savannah, Georgia.~~ |

Figure 27: Validation / training example pair from C4 with high to intermediate influence. Red / green highlighted text indicate deleted / added text in the training example comparing to the corresponding validation example, generated using Python `difflib`.

| Index | Estim. | Link and Text |
|---|---|---|
| valid 114853 | infl 0.0316 | http://antiquefinejewelry.info/1920s-antique-art-deco-solid-platinum-90ctw-old-mine-cut-diamond-navette-ring.php |

This is a 1920s Antique Art Deco Solid Platinum. 90ctw Old Mine Cut Diamond Navette Ring. Diamond - Old Mine Cut - GH VS1-VS2. 90%PLAT10%IRID; item has been tested and guaranteed to be solid PLATINUM. This ring is in excellent pre-owned condition. Please disregard the two characters at the end of the title, they are used for inventory purposes. As a courtesy, please notify us of any return. Always fast & free unless otherwise stated. Collectors Coins & Jewelry has been family owned and operated on Long Island, NY since 1946. We have four brick and mortar locations and offer the highest quality products with unbeatable customer service. The item "1920s Antique Art Deco Solid Platinum. 90ctw Old Mine Cut Diamond Navette Ring" is in sale since Thursday, November 30, 2017. This item is in the category "Jewelry & Watches\Fine Jewelry\Fine Rings\Diamond". The seller is "collectorsbuysell" and is located in Huntington, New York. This item can be shipped worldwide.

| train 504624 | men 0.1665 | http://goldbroochpin.com/vintage_estate_14k_solid_yellow_gold_carved_red_orange_coral_ship_brooch_pin.php |

Men's / Women's. 32mm (length) x 42mm (width) approx. 14K; Item has been tested and guaranteed to be solid GOLD. This is a 1920s Antique Art Deco Solid Platinum. 90ctw Old Mine Cut Diamond Natem is in vette Ring. Diamond ry good pre- Old Mine Cut - GH VS1-VS2. 90%PLAowned condition. T10%IRID; item has been tested and guaranteed to be solid PLATINUM. This ring is in excellent pre-owned condition. here is a slight chip on the coral. Please disregard the two characters at the end of the title, they are used for inventory purposes. As a courtesy, please notify us of any return. Always fast & free unless otherwise stated. Collectors Coins & Jewelry has been family owned and operated on Long Island, NY since 1946. We have four brick and mortar locations and offer the highest quality products with unbeatable customer service. The item "Vintage Estate 1920s Anti4k Solid Yellow Gold Carved Red Orange Coral Ship Brooch Pin&que Art Deco Solid Platinum. 90ctw Old Mine Cut Diamond Navette Ring" is in sale since Thursot; is in sale since Friday, November 30June 8, 2017 8. This item is in the category "Jewelry & Watches\Fine Jewelry\Fine RingPins & Brooches\Diamonds & Gemstones". The seller is "collectorsbuysell" and is located in Huntington, New York. This item can be shipped worldwide.

| | | |

| valid 278369 | infl 0.0156 | http://www.sandwell.gov.uk/info/200208/crime_prevention_and_emergencies/3155/the_community_trigger-request_a_review_on_reports_of_anti_social_behaviour |

A Community Trigger gives victims and communities the right to demand action on problems with anti social behaviour (ASB) they have reported in the past. The Community Trigger can be used by anyone who has reported ASB but feels no action has been taken. The Community Trigger is aimed at putting victims first and to hold agencies responsible for managing anti social behaviour to account. Agencies including councils, the police, local health teams and registered providers of social housing who receive a Community Trigger report will then need to conduct a case review. Who can raise a Community Trigger and when? Anyone can raise the trigger on behalf of the victim - for example a family member, friend, carer, councillor, Member of Parliament or other professional person. It doesn't matter who you originally reported the ASB to (the council, the police or your landlord) - please use the Community Trigger form here on our website. What happens after a Community Trigger is raised? When you complete the Community Trigger form, we will contact you to say we have received it and let you know what will happen next. Your completed form will be sent to Sandwell Council's Community Safety and Anti Social Behaviour Manager. A member of the ASB team will contact you, within two working days. The Community Trigger is not an alternative method of complaining about the service you have received. If you are not satisfied with the service you have received there is still an independent complaints process that you should follow. Feedback on the use of the Community Trigger.

| train 1897623 | mem 0.1837 | http://www.reddal.com/insights/reddal-talks-vietnam-building-a-strong-domestic-industrial-backbone-to-sustain-growth/ |

FDI attraction, export growth and a thriving consumer market have been characteristics of the Vietnamese growth story but what is next for this emerging market in South-East Asia? In this video, we discuss the prospect of the Vietnamese economy, possible setback of its growth strategy and potential remedies. Communitapturing FDI spillovers to build a strong back-bone local manufacturing industry Trigger gives victims and communities the right to demand action on problems with anti social behaviour (ASB) themay have reported in the pastold key to sustain growth. The Community Trigger can be used by anyone who has reported ASB but feels no action has been taken. The Community Trigger is aimed at putting Read more about the topic in our Reddal Insights article - Drivictims first and to hold agencies responsible for managing anti social behaviour to account. Agencies including councils, the police, local health teams and registered providers of social housing who receive a Community Trigger report will then need to conduct a case review. Who can raise a Community Trigger and when? Anyone can raise the trigger on behalf of the victim - for example a family member, friend, carer, councillor, Member of Parliament or other professional person. It doesnng Vietnam't matter who you originally reported the ASB to (the council, the police or your landlord) - please use the Community Trigger form here on our websites economic growth. What happens after a Community Trigger is raised? When you complete the Community Trigger form, we will contact you to say we have received it and let you know what will happen next. Your completed form will be sent to Sandwell Council's Community Safety and Anti Social Behaviour Manager. A member of the ASB team will contact you, within two working days. The Community Trigger is not an alternative method of complaining about the service you have received. If you are not satisfied with the service you have received there is still an independent complaints process that you should follow. Feedback on the use of the Community Trigger.

Figure 28: Validation / training example pair from C4 with intermediate to low influence. Red / green highlighted text indicate deleted / added text in the training example comparing to the corresponding validation example, generated using Python `difflib`.

| Index | Estim. | Wiki ID and Text |
|---|---|---|
| valid 196 | infl 0.2863 | wikidata_id=Q48845100, version_id=15957308656362122644 |
| | | _START_ARTICLE_ 2017 Missouri Valley Conference Men's Soccer Tournament _START_SECTION_ Background _START_PARAGRAPH_ The 2017 Missouri Valley Conference Men's Soccer Tournament is the culmination of the regular season. The regular season conference matches determine the seeding in the tournament, which determines the conference's automatic berth into the NCAA Tournament. All teams in the Missouri Valley Conference, or MVC, play each other once during the season. Teams play certain teams at home during even number years, and then will play those teams on the road during odd number years. Teams are awarded three points for a win, a point for a draw and no points for a loss._NEWLINE_In the event that teams are tied on points, the first tiebreaker is head-to-head record. If that tiebreaker is tied, goal differential is applied, followed by goals scored, then away goals, then RPI._NEWLINE_Missouri State won the regular season with a 5-2-1 record. |
| train 2044546 | mem 0.3659 | wikidata_id=Q65120533, version_id=1274679701798322632 |
| | | _START_ARTICLE_ 2018 Missouri Valley Conference Men's Soccer Tournament _START_SECTION_ Background _START_PARAGRAPH_ The 2018 Missouri Valley Conference Men's Soccer Tournament is the culmination of the regular season. The regular season conference matches determine the seeding in the tournament, which determines the conference's automatic berth into the NCAA Tournament. All teams in the Missouri Valley Conference, or MVC, play each other once during the season. Teams play certain teams at home during even number years, and then will play those teams on the road during odd number years. Teams are awarded three points for a win, a point for a draw and no points for a loss._NEWLINE_In the event that teams are tied on points, the first tiebreaker is head-to-head record. If that tiebreaker is tied, goal differential is applied, followed by goals scored, then away goals, then RPI._NEWLINE_Missouri StateCentral Arkansas won the regular season with a 5-2-4-1-1 record. |
| | | |
| valid 131821 | infl 0.1143 | wikidata_id=Q2150068, version_id=13935027521411622651 |
| | | _START_ARTICLE_ HNLMS K XIV _START_SECTION_ Service history _START_PARAGRAPH_ The submarine was laid down in Rotterdam at the shipyard of Rotterdamsche Droogdok Maatschappij on 31 May 1930. The launch took place on 11 July 1931. On 6 July 1933 the boat was commissioned in the Dutch navy._NEWLINE_On 7 February 1934 K XIV and K XV left the Netherlands for the Dutch East Indies. The route they took led through the Suez Canal. On 6 September 1938 she participated in a fleet show at Surabaya. The show was held in honor of the Dutch Queen Wilhelmina of the Netherlands who celebrating her 40th year as head of state. More than twenty navy ships participated in the show._NEWLINE_In the war K XIV sank several Japanese ships. She survived the war and was decommissioned on 23 April 1946. 1 June 1946 she was stricken. |
| train 1102598 | mem 0.1773 | wikidata_id=Q2789027, version_id=2769557335126998411 |
| | | _START_ARTICLE_ HNLMS K XIV _START_SECTION_ Service history _START_PARAGRAPH_ The submarine was laid down in Rotterdam at the shipyard of Rotterdamsche Droogdok Maatschappij on 31 May 1930. The launch took place on 11 July0 December 1931932. On 6 July30 December 1933 the boat was commissioned in the Dutch navy._NEWLINE_On 7 February 1934 K XV and K XIVK XV and K XV left the Netherlands for the Dutch East Indies. The route they took led through the Suez Canal. On 6 September 1938 she participateds in a fleet show at Surabaya. The show was held in honor of the Dutch Queen Wilhelmina of the Netherlands who celebrating herwas than 40th year as years the head of state. More than twenty navy ships participatede in the show._NEWLINE_In the war K XIV sank several Japanese ships. She survived the war and was decommissioned on 23 April 1946. 1 June 1946 she was stricken. and sold for scrap in December 1950. |
| | | |
| valid 28865 | infl 0.0631 | wikidata_id=Q25059205, version_id=4123980408724644851 |
| | | _START_ARTICLE_ Wanim Island _START_SECTION_ Geography _START_PARAGRAPH_ The island has an area of 3.56 km², it is part of the Pana Tinani Group. The island is hilly, rising to 119 m at Mt. Wanim._NEWLINE_The island is 0.9 km south of Pana Tinani, and separated from it with the Bulami Channel. _START_SECTION_ History _START_PARAGRAPH_ The island was discovered in the late 18th century. _START_SECTION_ Population _START_PARAGRAPH_ At the census of population in 2014, the island had 600 inhabitants, spread across 3 small villages. _NEWLINE_The main town is Bunbun, located on the northwest point. |
| train 1339144 | mem 0.1241 | wikidata_id=Q25059204, version_id=15365851189558929194 |
| | | _START_ARTICLE_ WanimNimoa Island _START_SECTION_ Geography _START_PARAGRAPH_ The island has an area of 3.56 km², it is part of the Pana Tinani Group. The island is hilly, rising to 11940 m at Mt. WanimNimoa._NEWLINE_The island is 0.91.7 km sounorth of Pana TinanVanatinai, and separated from it with the Bulami Channel. _START_SECTION_ History _START_PARAGRAPH_ The island was discovered in the late 18th century. _START_SECTION_ Population _START_PARAGRAPH_ At the census of population in 2014, the island had 600395 inhabitants, spread across 35 small villages. _NEWLINE_The main town is BunbunSoluwo, located on the norsouthwest point. |

Figure 29: Validation / training example pair from Wiki40B:en with high to intermediate influence. Red / green highlighted text indicate deleted / added text in the training example comparing to the corresponding validation example, generated using Python `difflib`.

| Index | Estim. | Link and Text |
|---|---|---|
| valid 127940 | infl 0.0348 | wikidata_id=Q58932464, version_id=4799254013558324766 |

_START_ARTICLE_ Alexis Gutiérrez _START_SECTION_ Youth career _START_PARAGRAPH_ Gutiérrez at a young age was scouted and joined Guadalajara's youth academy in 2012. He then continued through Chivas Youth Academy successfully going through U-13, U-15, U-17 and U-20. Until finally receiving attention to join Cruz Azul, Pedro Caixinha being the coach promoting Gutiérrez to the first team. _START_SECTION_ Cruz Azul _START_PARAGRAPH_ Gutiérrez made his professional debut in the Liga MX on the 28 of April 2019. He was subbed in by coach Pedro Caixinha in the 81 minute which ended in a 4-1 Win against Lobos BUAP.

| Index | Estim. | Link and Text |
|---|---|---|
| train 1104654 | mem 0.1723 | wikidata_id=Q35493379, version_id=5930618974605611002 |

_START_ARTICLE_ Alexis Gutiérrez Brian Figueroa _START_SECTION_ Youth career _START_PARAGRAPH_ Gutiérrez Figueroa at a young age was scouted and joined Guadalajara's Pumas youth academy in 2012 1. He then continued through Chiv Pumas Youth Academy successfully going through U-13, U-15, U-17 and U-20. Until finally receiving attention to join Cruz Azul the first team, Pedro Caixinh Francisco Palenci a being the coach promoting Gutiérrez Figueroa to the first team. _START_SECTION_ Cruz Azul Pumas UNAM _START_PARAGRAPH_ Gutiérrez Figueroa made his professional debut in the Liga MX on the 28 of April 3 of July 2019 7. He started w as subbed in by coach Pedro Caixinha in the 81 minute ith the first team which ended in a 4-1 1-0 Win against Lobos BUAP Pachuca.

| Index | Estim. | Link and Text |
|---|---|---|
| valid 48624 | infl 0.0154 | wikidata_id=Q20994343, version_id=3654160548380714055 |

_START_ARTICLE_ Rodrigo Tarín _START_SECTION_ Club career _START_PARAGRAPH_ Born in Chiva, Valencian Community, Tarín joined FC Barcelona's youth categories in 2011, from Valencia CF. On 18 September 2014, he renewed his contract until 2018, and was promoted to the reserves in Segunda División B the following July._NEWLINE_Tarín made his senior debut on 22 August 2015, starting in a 1–2 away loss against UE Cornellà. He scored his first senior goal on 17 September of the following year, netting the winner in a 2–1 home success over CD Atlético Baleares; in November, however, he suffered a knee injury which took him out for six months._NEWLINE_Tarín made his professional debut on 19 August 2017, starting in a 2–1 away win against Real Valladolid for the Segunda División championship. The following 27 June, he signed a three-year deal with La Liga side CD Leganés._NEWLINE_Tarín made his debut in the main category of Spanish football on 26 September 2018, starting in a 2–1 home defeat of former side Barcelona.

| Index | Estim. | Link and Text |
|---|---|---|
| train 281892 | mem 0.2074 | wikidata_id=Q29467626, version_id=2545406209863252269 |

_START_ARTICLE_ Rodrigo Tarín 2017 Tamil Nadu Farmers Protest _START_SECTION_ Club career History _START_PARAGRAPH_ Born in Chiva, Valencian Community, Tarín The severe 2016 drought and the hydrocarbon pro joined FC Barcelona ect's youth categories i implementation in Tamil Nadu frustrated Tamil farmers. They started protests in Jantar Mantar, New Delhi._NEWLINE_ In 2016, Tamil Nadu faced the worst rainfall for the prior 140 years. Average annual rainfall decreased 62% in 2016. The Northeast monsoon season failed over Tamil Nadu with the worst rainfall ever, from Valencia CF. On with scattered rain in some areas. The highest deficit of rainfall in Indian states in 2016 was in Tamil Nadu with an 82% reduction. Se ptem eing their crops failing, many farmers be r-gan to die of heart attacks and suicides. The Tamil Nadu government requested a Rs.40,000 crores relief fund. The Government approved only Rs.2014, he renewed his contract until 2018, and was promoted to the reserves in Segunda División B the following July crores._NEWLINE_Tarín made his senior de he Hydrocar but on 22 August 2015, starting in a 1–2 away loss against UE on extracting project implemented in the agricultural fields of Tamil villages like Neduvasal by the C ornellà. He scored his first senior goal on 17 September of the following year, netting the winner in a 2–1 home success over CD Atlético Baleares; in November, however, he suffered a knee injury which took him out for six months. entral government annoyed the Tamil farmers. The people of Neduvasal protested _NEWLINE_Ta rín made his professional de mil Nadu farmers protested, led b but on 19 y Tamil Farmer August yyakannu. They invented protesting methods such as shaving half their beards and hai and eating rats and snakes. Farmers at Jantar Mantar vowed to drink their urine on April 22, 2017. They promised to eat feces the next day. On the 22nd day of the protest, starting in a 2–1 a the Tamil Nadu High Court gave judgement way win against Real Valladolid for the Segunda División championship. The following 27 June, he signed a three-year deal with La Liga side CD Leganés._NEWLINE_Tarín made his de iving farmers loan in cooperative b ut in the main category of Spanish football on 26 September 2018, starting in a 2–1 home defeat of former side Barcelona. anks.

Figure 30: Validation / training example pair from Wiki40B:en with intermediate to low influence. Red / green highlighted text indicate deleted / added text in the training example comparing to the corresponding validation example, generated using Python difflib.

| Index | Estim. | Link and Text |
|---|---|---|
| generated 1361 | infl 0.1805 | https://en.trend.az/business/finance/3046505.html |

Baku, Azerbaijan, April 15 Trend: Official exchange rate of the US dollar and euro against Azerbaijani manat was set at 1.7 and 1.9225 manats, respectively, for April 15. Below are the rates of Azerbaijani manat against world currencies, according to the data from the Central Bank of Azerbaijan for April 15. Currencies April 15, 2019 1 US dollar 1 USD 1.7 1 euro 1 EUR 1.9225 1 Australian dollar 1 AUD 1.2188 1 Argentine peso 1 ARS 0.0403 100 Belarus rubles 1 BYN 0.8055 1 Brazil real 1 BRL 0.4379 1 UAE dirham 1 AED 0.4628 1 South African rand 1 ZAR 0.1218 100 South Korean won 100 KRW 0.15 1 Czech koruna 1 CZK 0.0751 1 Chilean peso 100 CLP 0.2572 1 Chinese yuan 1 CNY 0.2534 1 Danish krone 1 DKK 0.2576 1 Georgian lari 1 GEL 0.6309 1 Hong Kong dollar 1 HKD 0.2168 1 Indian rupee 1 INR 0.0245 1 British pound 1 GBP 2.226 100 Indonesian rupiah 100 IDR 0.0121 100 Iranian rials 100 IRR 0.004 1 Swedish krona 1 SEK 0.1837 1 Swiss franc 1 CHF 1.6967 1 Israeli shekel 1 ILS 0.4763 1 Canadian dollar 1 CAD 1.2756 1 Kuwaiti dinar 1 KWD 5.5887 1 Kazakh tenge 1 KZT 0.0045 1 Kyrgyz som 1 KGS 0.0238 100 Lebanese pounds 100 LBP 0.1126 1 Malaysian ringgit 1 MYR 0.4134 1 Mexican peso 1 MXP 0.0906 1 Moldovan leu 1 MDL 0.0963 1 Egyptian pound 1 EGP 0.0982 1 Norwegian krone 1 NOK 0.2003 100 Uzbek soums 100 UZS 0.0203 1 Polish zloty 1 PLN 0.4492 1 Russian ruble 1 RUB 0.0264 1 Singapore dollar 1 SGD 1.2566 1 Saudi riyal 1 SAR 0.4533 1 SDR (Special Drawing Rights of IMF) 1 SDR 2.3635 1 Turkish lira 1 TRY 0.294 1 Taiwan dollar 1 TWD 0.0551 1 Tajik somoni 1 TJS 0.1801 1 New Turkmen manat 1 TMM 0.4857 1 Ukrainian hryvna 1 UAH 0.0637 100 Japanese yen 100 JPY 1.5187 1 New Zealand dollar 1 NZD 1.1513 Follow Trend on Telegram. Only most interesting and important news

| Index | Estim. | Link and Text |
|---|---|---|
| train 2072973 | men 0.3534 | https://en.trend.az/business/finance/3033358.html |

Baku, Azerbaijan, ~~April~~ March 15 Trend: Official exchange rate of the US dollar and euro against Azerbaijani manat was set at 1.7 and 1.92~~25~~41 manats, respectively, for ~~April~~March 15. Below are the rates of Azerbaijani manat against world currencies, according to the data from the Central Bank of Azerbaijan for ~~April~~March 15. Currencies ~~April~~March 15, 2019 1 US dollar 1 USD 1.7 1 euro 1 EUR 1.92~~25~~41 1 Australian dollar 1 AUD 1.2~~188~~037 1 Argentine peso 1 ARS 0.04~~03~~17 100 Belarus rubles 1 BYN 0.79~~68~~055 1 Brazil real 1 BRL 0.4~~379~~420 1 UAE dirham 1 AED 0.4628 1 South African rand 1 ZAR 0.12~~18~~173 100 South Korean won 100 KRW 0.15~~493~~ 1 Czech koruna 1 CZK 0.075~~1~~0 1 Chilean peso 100 CLP 0.25~~72~~32 1 Chinese yuan 1 CNY 0.25~~34~~40 1 Danish krone 1 DKK 0.25~~76~~9 1 Georgian lari 1 GEL 0.63~~09~~33 1 Hong Kong dollar 1 HKD 0.216~~8~~86 1 Indian rupee 1 INR 0.0245 1 British pound 1 GBP 2.22~~6~~516 100 Indonesian rupiah 100 IDR 0.012~~1~~419 100 Iranian rials 100 IRR 0.004~~0~~ 1 Swedish krona 1 SEK 0.183~~7~~729 1 Swiss franc 1 CHF 1.696~~7~~755 1 Israeli shekel 1 ILS 0.47~~63~~ 1 Canadian dollar 1 CAD 1.27~~56~~670 1 Kuwaiti dinar 1 KWD 5.58~~87~~959 1 Kazakh tenge 1 KZT 0.0045 1 Kyrgyz som 1 KGS 0.023~~8~~841 100 Lebanese pounds 100 LBP 0.1126 1 Malaysian ringgit 1 MYR 0.413~~4~~456 1 Mexican peso 1 MXP 0.09~~06~~06880 1 Moldovan leu 1 MDL 0.096~~3~~390 1 Egyptian pound 1 EGP 0.098~~2~~279 1 Norwegian krone 1 NOK 0.200~~3~~31986 100 Uzbek soums 100 UZS 0.020~~3~~3199 1 Polish zloty 1 PLN 0.449~~2~~270 1 Russian ruble 1 RUB 0.026~~4~~0 1 Singapore dollar 1 SGD 1.25~~66~~646 1 Saudi riyal 1 SAR 0.4533 1 SDR (Special Drawing Rights of IMF) 1 SDR 2.363~~5~~546 1 Turkish lira 1 TRY 0.294~~3110~~ 1 Taiwan dollar 1 TWD 0.055~~1~~0 1 Tajik somoni 1 TJS 0.1801 1 New Turkmen manat 1 TMM 0.4857 1 Ukrainian hryvna 1 UAH 0.063~~7~~71 100 Japanese yen 100 JPY 1.5~~187~~187220 1 New Zealand dollar 1 NZD 1.15~~13~~1636 Follow Trend on Telegram. Only most interesting and important news

| Index | Estim. | Link and Text |
|---|---|---|
| generated 20228 | infl 0.0540 | https://www.capitalfm.com/northeast/terms-conditions/i-saw-it-first-1000-april19/ |

I Saw It First Capital Yorkshire, Capital Manchester and Capital North East April 2019 RULES 1. The following rules ("Specific Rules") together with the general competition terms and conditions (which can be found at https://www.capitalfm.com/terms-conditions/competition/) (the "Competition Terms and Conditions") of Global (as defined in the Competition Terms and Conditions) apply to the 'I Saw It First' radio competition (the "Competition") which will run from Monday 8th April 2019 to Saturday 27th April 2019 on Capital Yorkshire, Capital Manchester and Capital North East (the "Radio Station"). 2. Anyone who enters the Competition (an "Entrant") will be deemed to have read and accepted the Specific Rules and the Competition Terms and Conditions and will be bound by them. Details of the Competition: 3. To enter the Competition an Entrant must go online to www.capitalfm.com, answer the multiple choice question and register their details. Online entry will open at 00:00 on Monday 8th April 2019 and close at 17:30 on Saturday 27th April 2019. 4. The winner will be picked at random from all correct entries and notified via telephone or email. Eligibility: 5. Entrants must be 18 or over. 6. Entry is restricted to one entry per person; duplicate entries will be excluded from the Competition. Prize: 7. One winner will receive the £1000 prize package which consists of: A £600 I Saw It First voucher £400 cash 8. The £600 voucher will be sent as an e-code for the winner to spend online and will have no expiry date. 9. £400 will be sent as a cheque to the winners address within 28 days of winning. 10. Prize is none transferable. 11. Global reserves the right to substitute the prize with another prize at its absolute discretion.

| Index | Estim. | Link and Text |
|---|---|---|
| train 1072243 | mem 0.2117 | https://www.capitalfm.com/yorkshire/terms-conditions/leeds-city-college-march19/ |

I Saw It First Leeds City College Capital Yorkshire, ~~Capital Manchester and Capital North East April~~rch 2019 RULES 1. The following rules ("Specific Rules") together with the general competition terms and conditions (which can be found at https://www.capitalfm.com/terms-conditions/competition/) (the "Competition Terms and Conditions") of Global (as defined in the Competition Terms and Conditions) apply to ~~the 'I Saw It First'~~ "Apprenticeship' radio competition (the "Competition") which will run from ~~Saturday 2nd~~ Monday 8th ~~April~~rch 2019 ~~to~~until Saturday 27~~th April~~rd March 2019 (the "Duration") on Capital Yorkshire, (the "Radio Station"). 2. Anyone who enters the ~~Capital Manchester and Capital North~~ competition (an "~~East (the "Radio Station~~ntrant"). 2. An will be deemed to have read and accepted the Specific Rules and the Competition Terms and Conditions and will be bound by ~~one who enters~~ them. Details of the Competition ~~(an ":~~ 3. To enter the Competition an ~~Entrant")~~ will be deemed to have read and accepted the Speci must go online to www.capital~~fic Rules and the Competition Terms and Conditions and will be bound by them. Details of the Competition: 3. To enter the Competition an Entrant must~~ m.com/ and re~~go online to www.capitalfm.com, answer the multiple choice question and register their detail~~gister their details. 4. Online entry will open at 00:00 on Saturday 2nd ~~Monday 8th April~~rch 2019 and close at ~~1723~~:59 on Saturday 2~~3~~0 ~~on Saturday 27th April~~rd March 2019. ~~4. The winner will be picked at random~~5. Entrants must answer the question and register their details f~~rom all correct entries and noti~~or the chance to win. 6. One winner will be selected at random fi~~erom all correct and vi~~a telephone or email~~alid entries and contacted by Capital FM. Eligibility: 7. Entrants must be age 15~~. Entrants must b~~ or over. 8. Adult consent (someone age 18 or over~~-6)~~ will be required for winners under the age of 18. 9. Entry is restricted to one entry per person~~; duplicate entries will be excluded from the Competition. Prize: 7. One winner will recei~~or the duration of the promotion. Duplicate entries will be remove~~the £d.~~ 10. Entrants must be from Yorkshire postcodes- DN, BD, LS, YO, WF, HD, HX, S, HG and HU Prize~~ package which consists of: A £600 I Saw It First voucher £400 cash 8. The £600 voucher will be sent as an e-code for the winner to spend online and will have no expiry date. 9. £400 will be sent as a cheque to the winners address within 28 days of winning. 10. Prize is none transferable.~~ 11. One winner will receive 1 x Apple Airpods and 1 x Apple iPad Pro 11-inch display. 12. Prize delivery will be discussed with the winner when they are contacted by a member of the Capital FM Yorkshire team. 13. The prize won't be substituted for any other prize or cash equivalent. 14. Global reserves the right to substitute the prize ~~with another prize at its absolute discretion.~~ at its absolute discretion.

Figure 31: Generated / training example pair from RealNews with high to intermediate influence. The generated examples are directly taken from publicly released generations of the Grover-Mega (p=0.96) model [Zellers et al., 2019]. Red / green highlighted text indicate deleted / added text in the training example comparing to the corresponding validation example, generated using Python `difflib`.

| Index | Estim. | Link and Text |
|---|---|---|
| generated 19128 | infl 0.0357 | https://www.globalresearch.ca/us-plans-designate-iran-revolutionary-guards-irgc-terrorists/5673967 |

Notes Disclaimer: The contents of this article are of sole responsibility of the author(s). The Centre for Research on Globalization will not be responsible for any inaccurate or incorrect statement in this article. The Centre of Research on Globalization grants permission to cross-post Global Research articles on community internet sites as long the source and copyright are acknowledged together with a hyperlink to the original Global Research article. For publication of Global Research articles in print or other forms including commercial internet sites, contact: [email protected] www.globalresearch.ca contains copyrighted material the use of which has not always been specifically authorized by the copyright owner. We are making such material available to our readers under the provisions of "fair use" in an effort to advance a better understanding of political, economic and social issues. The material on this site is distributed without profit to those who have expressed a prior interest in receiving it for research and educational purposes. If you wish to use copyrighted material for purposes other than "fair use" you must request permission from the copyright owner. For media inquiries: [email protected]

| train 1244174 | men 0.1904 | https://www.globalresearch.ca/the-american-media-empire-of-managed-news/20568 |

Operation FALCONotes mass arrests; suspension of habeas corpus by the Chief Executive; US and NATO as protectors of global capital; structure of inequality maintained by global capital and military empire worldwide; repression that addresses us all; the 1948 Declaration of Human Rights. 

| generated 7818 | infl 0.0207 | https://en.trend.az/business/energy/3046526.html |

Baku, Azerbaijan, April 15 By Ali Mustafayev – Trend: Italy would have to import almost 50 percent of its natural gas through Germany without the Trans Adriatic Pipeline (TAP), managing director of TAP AG Beat Rathmann told Trend April 15. TAP is a part of the Southern Gas Corridor project, aimed at diversifying the European energy supply sources and routes. The 930-kilometer long TAP project envisages transportation of gas from Azerbaijan's Shah Deniz Stage 2 to the EU countries. The pipeline will connect to the Trans Anatolian Natural Gas Pipeline (TANAP) on the Turkish-Greek border, run through Greece, Albania and the Adriatic Sea, before coming ashore in Italy's south. TAP's shareholding is comprised of BP (20 percent), SOCAR (20 percent), Snam S.p.A. (20 percent), Fluxys (19 percent), Enagás (16 percent) and Axpo (5 percent). --- Follow the author on Twitter: @Ali_Mustafayev Follow Trend on Telegram. Only most interesting and important news

| train 1246306 | mem 0.1520 | https://en.trend.az/business/energy/2791327.html |

Baku, Azerbaijan, April 15ugust 29 By Ali MustafaLeman Zeyenalova – Trend: ItaNearly 60 percent of pipes have been would have to import almost 50 percent of its natural gas through Germany withoutelded as part of construction of the Trans Adriatic Pipeline (TAP), managing director of in Greece and Albania, TAP AG Beat Rathmann told Trend April 15consortium said in a message on its Twitter page. This accounts for 458 kilometers of the total of 765 kilometers of TAP is a part of the Southern Gas Corridor project, aimed at diversifying the European energy supply sources and routroute in those two countries. The 930-kilometer long TAP is a part of the Southern Gas Corridor, which is one of the priority energy projects for the European Union. The project envisages transportation of gas from Azerbaijan's Shah Deniz Stage 2 to the EU countries. The pipeline will connect to the Trans Anatolian Natural Gas Pipeline (TANAP) on the Turkish-Greek border, run through Greece, Albania and the Adriatic Sea, before coming ashore in Italy's south. TAP will be 878 kilometers in length (Greece 550 kilometers, Albania 215 kilometers, Adriatic Sea 105 kilometers, and Italy 8 kilometers). TAP's shareholding is comprised of BP (20 percent), SOCAR (20 percent), Snam S.p.A. (20 percent), Fluxys (19 percent), Enagás (16 percent) and Axpo (5 percent). --- Follow the author on Twitter: @AliLyaman_MustafaZeyevn Follow Trend on Telegram. Only most interesting and important news

Figure 32: Generated / training example pair from RealNews with intermediate to low influence. The generated examples are directly taken from publicly released generations of the Grover-Mega (p=0.96) model [Zellers et al., 2019]. Red / green highlighted text indicate deleted / added text in the training example comparing to the corresponding validation example, generated using Python difflib.

| Index | Estim. | Link and Text |
|---|---|---|
| generated 21998 | infl 0.0218 | http://www.arabnews.com/node/1483111/world |

NEW DELHI: India is likely to see average monsoon rains this year, the state-run weather office said on Monday, which should support agricultural production and economic growth in Asia's third-biggest economy, where half of the farmland lacks irrigation. Monsoon rainfall is expected to be 96 percent of the long-term average, M. Rajeevan, secretary at the Ministry of Earth Sciences, told a news conference. The India Meteorological Department (IMD) defines average, or normal, rainfall as between 96 percent and 104 percent of a 50-year average of 89 centimeters for the entire four-month season beginning June. "Overall, the country is expected to have well distributed rainfall scenario during the 2019 monsoon season, which will be beneficial to farmers in the country during the ensuing Kharif (summer-planting) season," the IMD said in its forecast. Skymet, the country's only private weather forecasting agency, earlier this month forecast rainfall could be below normal this year. Monsoon rains, the lifeblood for India's farm-dependent $2.6 trillion economy, arrive on the southern tip of Kerala state around June 1 and retreat from the desert state of Rajasthan by September. After a wet spell, sowing of summer-sown crops gets off to a strong start, boosting crop yields and output which in turn raises rural incomes and usually lifts consumer spending in India. If plentiful monsoon rains lift agricultural production this year, that could keep food prices under control. Subdued overall inflation could also add to pressure on India's central bank to cut interest rates. "IMD's prelim forecast, showing near-normal and well distributed rainfall, will bode well for near-term food inflation," said Madhavi Arora, lead economist at Edelweiss Securities, FX and Rates. The next policy review by India's central bank is scheduled for June 6, after the country's election. Millions of Indians are casting their votes in a mammoth general election, spread over seven weeks. On the downside, higher production could mean farmers continue to get hit by low crop prices, a major cause for concern in rural India, where most Indians live, in the past two years. After falling for five straight months, retail food prices in India rose 0.30 percent in March from a year earlier. Last month, a senior IMD official told Reuters that this year's monsoon was likely to be robust and healthy provided there wasn't a surprise El Nino phenomenon. "El Nino is weakening and we expect that El Nino will get weakened further. There is no reason to be worried about El Nino," Rajeevan said. A strong El Nino, marked by a warming of the sea surface on the Pacific Ocean, can cause severe drought in Australia, Southeast Asia and India, while drenching other parts of the world such as the US Midwest and Brazil in rains. The emergence of a strong El Nino triggered back-to-back droughts in 2014 and 2015, for only the fourth time in over a century, driving some Indian farmers to penury and suicide. ECONOMIC GROWTH Good rains will spur the planting of crops such as rice, corn, cane, cotton and soybeans. Stronger agricultural production would help support India's economy. It is still the world's fastest-growing major economy, but annual growth slowed to 6.6 percent in the December quarter, from 7.0 percent in the previous period and the slowest in five quarters. The monsoon usually covers the half of the country in the first 15 days. The rains reach central India's soybean areas by the third week of June and western cotton-growing areas by the first week of July. India's weather office will update its forecast in the first week of June. However, on average, the IMD has forecast accurately only once every five years over the past two decades, even after taking into account an error band of plus or minus 5 percentage points.

| Index | Estim. | Link and Text |
|---|---|---|
| train 326212 | men 0.1555 | https://www.reuters.com/article/us-india-monsoon/india-set-to-receive-average-monsoon-rains-in-boost-for-poll-bound-modi-idUSKBN1HN1AQ |

NEW DELHI (Reuters) - India is likely to see average monsoon rains this year average monsoon rains in 2018, the state-run weather office said on Monday weather office said, which should support agricultural production raising the possibility of higher farm and economic growth in Asia's third-biggest economy, where half of the farmland lacks irrigation. Monsoon rainfall is, the lifeblood of the country's $2 trillion economy, are expected to be 96 percent of the 7 percent of a long-term average, M.K.J. Rajeevanmesh, secretary at the Ministry of Earth Sciences, told a news conference. The director general of the state-run India Meteorological Department (IMD) defines average, or normal, rainfall as between 96 percent and 104 percent of a 50-year average of 89 centimeters for the entire four-month season beginning June told a news conference. "We see very less probability of a deficit monsoon," Ramesh said on Monday. Overall ther than lifting farm and wider economic growth, the country is expected to have well distributed rainfall scenario during the a spell of good rains will keep a lid on inflation, potentially tempting Prime Minister Narendra Modi to bring forward general elections due in May 2019 monsoon season. India's weather office defines average, which will be beneficial to farmers in the country during the ensuing Kharif (summer or normal, rainfall as between 96 percent and 104 percent of a 50-planting) season year average of 89 cms for the entire four-month season beginning June. "The moderate La Nina conditions developed in the equatorial Pacific during last year started weakening in the early part of this year and currently have turned to weak La Nina conditions," the IMD said in its forecas statement. Skymet La Nina is a weather pattern that brings equatorial Pacific Ocean temperatures, the country's only pri rainfall patterns and winds closer to a vate weather forecasting agency, earlier this month forecast rainfall could be below normal this year erage. The latest forecasts from global models indicate conditions over the Pacific will turn neutral before the beginning of monsoon season, the IMonsoon rains D said. Good rains will spur the planting of crops such as rice, the lifeblood for corn, cotton and soybeans, accelerating economic growth that rose 7.2 percent in the December quarter, its fastest in five quarters, compared with China's 6.8 percent in that quarter. Growth in the December quarter restored India's farm status as the world's fastest growing major economy. RURAL INCOME Average monsoon rains, with good distribution in July and August would support rural demand, said Rupa Rege Nitsure, group chief economist at L&T Finance Holdings, a Mumbai-dependent $2 based non-banking finance company. 6 trillion economy Good rains boost rural incomes, arrive on the southern tip of Kerala state around June 1 and retreat from the desert state of Rajasthan by September lifting the demand for an array of consumer goods ranging from lipsticks to refrigerators. After a wet spell n average monsoon would keep food inflation lower, sowing of summer-sown crops gets off to a strong start Nitsure said. On Monday, boosting crop yields and output which in turn raises rural incomes and usually lifts consumer spending in government data showed India's wholesale food prices fell 0.07 percent in March 2018 from a year earlier. If plentiful monsoon rains lift agricultural production this year ndia's weather office will update its forecast in June. On an average, that could e IMD has forecast accurately only once in every five years over the past two decades, even after take keep food prices under control ing into account an error band of plus or minus 5 percentage points. Subdued Rains usually lash Kerala state on the south coast around June 1, and cover all inflation could also add to pressure on the whole country by mid-July. Timely rains trigger planting of crops such as rice, soybeans and cotton. The monsoon usually covers the half of the country in the first 15 days. The rains reach central India's central ban soybean areas by the third week to cut interest rates of June and western cotton-growing areas by the first week of July. Good rains would help boost soybean output which in turn could cut expensive vegetable oil imports by India, the world's biggest importer of edible oils, which is the third-biggest import item after crude oil and gold. Currently India is struggling with huge amounts of sugar and good rains could further bump up the supply of the sweetener. "IMD's prelim forecast A good monsoon will help bring down edible oil imports, showing near-normal and well distributed rainfall, will bode well for near-term food inflation but it could also create a problem of plenty in pulses and sugar," said a Madhavi Arora, lead economist at Edelweiss Securities, FX and Rates. The next policy review by India's central bank is scheduled for June 6, after the country's election. Millions of Indians are casting their votes in a mammoth general election, spread over seven weeks. On the downside, higher production could mean farmers continue to get hit by low crop prices, a major cause for concern in rural India, where most Indians live, in the past two years umbai-based dealer with a global trading firm. After falling for five straight months, retail food prices in rage monsoon rainfall will help India rose 0.30 percent in March from a year earlie retain its position as the world's top rice exporter. Last month, a senior IMD official told Reuters that this year's monsoon was likely to be robust and healthy provided there wasn't a surprise El Nino phenomenon. "El Nino is weakening and we expect that El Nino will get weakened further. There is no reason to be worried about El Nino," Rajeevan said. A strong El Nino, marked by a warming of the sea surface on the Pacific Ocean, can cause severe drought in Australia, Southeast Asia and India, while drenching other parts of the world such as the US Midwest and Brazil in rains. The emergence of a strong El Nino triggered back-to-back droughts in 2014 and 2015, for only the fourth time in over a century, driving some Indian farmers to penury and suicide. ECONOMIC GROWTH Good rains will spur the planting of crops such as rice, corn, cane, and soybeans. Stronger agricultural production would help support India's economy. It is still the world's fastest-growing major economy, but annual growth slowed to 6.6 percent in the December quarter, from 7.0 percent in the previous period and the slowest in five quarters. The monsoon usually covers the half of the country in the first 15 days. The rains reach central India's soybean areas by the third week of June and western cotton-growing areas by the first week of July. India's weather office will update its forecast in the first week of June. However, on average, the IMD has forecast accurately only once every five years over the past two decades, even after taking into account an error band of plus or minus 5 percentage points.

Figure 33: Generated / training example pair from RealNews with low influence. The generated examples are directly taken from publicly released generations of the Grover-Mega (p=0.96) model [Zellers et al., 2019]. Red / green highlighted text indicate deleted / added text in the training example comparing to the corresponding validation example, generated using Python difflib.

