## A  Extended Related Work

The relation to previous work are briefly explained in the Introduction and Section B. In this section, we present an extended discussion.

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

1.922541 1 Australian dollar 1 AUD 1.2188037 1 Argentine peso 1 ARS 0.040317 100 Belarus rubles 1 BYN 0.79680055 1 Brazil real 1 BRL 0.4379420 1 UAE dirham 1 AED 0.4628 1 South African rand 1 ZAR 0.1218173 100 South Korean won 100 KRW 0.15493 1 Czech koruna 1 CZK 0.07510 1 Chilean peso 100 CLP 0.25732 1 Chinese yuan 1 CNY 0.25340 1 Danish krone 1 DKK 0.25769 1 Georgian lari 1 GEL 0.630933 1 Hong Kong dollar 1 HKD 0.21686 1 Indian rupee 1 INR 0.0245 1 British pound 1 GBP 2.22516 100 Indonesian rupiah 100 IDR 0.012419 100 Iranian rials 100 IRR 0.0040 1 Swedish krona 1 SEK 0.183729 1 Swiss franc 1 CHF 1.696755 1 Israeli shekel 1 ILS 0.47063 1 Canadian dollar 1 CAD 1.275670 1 Kuwaiti dinar 1 KWD 5.5887959 1 Kazakh tenge 1 KZT 0.0045 1 Kyrgyz som 1 KGS 0.023841 100 Lebanese pounds 100 LBP 0.1126 1 Malaysian ringgit 1 MYR 0.413456 1 Mexican peso 1 MXP 0.0906880 1 Moldovan leu 1 MDL 0.096390 1 Egyptian pound 1 EGP 0.098279 1 Norwegian krone 1 NOK 0.20031986 100 Uzbek soums 100 UZS 0.0203199 1 Polish zloty 1 PLN 0.449270 1 Russian ruble 1 RUB 0.02640 1 Singapore dollar 1 SGD 1.25646 1 S

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

Notes Disclaimer: The contents of this article are of sole responsibility of the author(s). The Centre for Research on Globalization will not be responsible for any inaccurate or incorrect statement in this article. The Centre of Research on Globalization grants permission to cross-post Global Research articles on community internet sites as long the source and copyright are acknowledged together with a hyperlink to the original Global Research article. For publication of Global Research articles in print or other forms including commercial internet sites, contact: [email protected] www.globalresearch.ca contains copyrighted material the use of which has not always been specifically authorized by the copyright owner. We are making such material available to our readers under the provisions of "fair use" in an effort to advance a better understanding of political, economic and social issues. The material on this site is distributed without profit to those who have expressed a prior interest in receiving it for research and educational purposes. If you wish to use copyrighted material for purposes other than "fair use" you must request permission from the copyright owner. For media inquiries: [email protected]

| train 1244174 | men 0.1904 | https://www.globalresearch.ca/the-american-media-empire-of-managed-news/20568 |

Operation FALCON Notes mass arrests; suspension of habeas corpus by the Chief Executive; US and NATO as protectors of global capital; structure of inequality maintained by global capital and military empire worldwide; repression that addresses us all; the 1948 Declaration of Human Rights. Disclaimer: The contents of this article are of sole responsibility of the author(s). The Centre for Research on Globalization will not be responsible for any inaccurate or incorrect statement in this article. The Centre of Research on Globalization grants permission to cross-post Global Research articles on community internet sites as long the source and copyright are acknowledged together with a hyperlink to the original Global Research article. For publication of Global Research articles in print or other forms including commercial internet sites, contact: [email protected] www.globalresearch.ca contains copyrighted material the use of which has not always been specifically authorized by the copyright owner. We are making such material available to our readers under the provisions of "fair use" in an effort to advance a better understanding of political, economic and social issues. The material on this site is distributed without profit to those who have expressed a prior interest in receiving it for research and educational purposes. If you wish to use copyrighted material for purposes other than "fair use" you must request permission from the copyright owner. For media inquiries: [email protected]