# OpenReview forum: "Counterfactual Memorization in Neural Language Models"
_NeurIPS.cc/2023/Conference — NeurIPS 2023 spotlight_

### Official Review · Reviewer_dGf9 · 2023-07-06

**Soundness:** 3 good
**Presentation:** 2 fair
**Contribution:** 3 good
**Rating:** 6
**Confidence:** 3

**Summary:**

This paper defines counterfactual memorization as the difference between a model's expected performance on a sample when it is included in the training data and that when it is not. The performance is measured by per-token prediction accuracy. The paper studies common patterns across counterfactually memorized samples. It further defines counterfactual influence similarly to counterfactual memorization, except that the performance is measured on any sample, and studies the correlation between counterfactual memorization of a sample and its counterfactual influence maximized over some set of data.

**Strengths:**

- It is interesting to study whether the existence of one training document will affect model predictions.
- The mathematical definition in Eq. (1) - (4) are reasonable.
- Ablation studies including 3.3 and 3.4 are well-explained.

**Weaknesses:**

Overall:

- The research problem can be more crisply defined. It is unclear what type of information the paper aims to study. Overall per-token top-1 accuracy is not sophisticated enough to be reflective of memorization of "sensitive information" (L2) or "details of specific events" (L111).
- There are technically mistaken statements. E.g. L32 says counterfactual memorization is a measure of an expected change. But it is indeed the difference between two expected values where the expectations are over different distributions.
- The quantitative definition about which information is considered "common" or "rare" is unclear (L48). While the paper uses 2^21 documents (L95), only 38000 are identified as being a near-duplicate with at least one other example (L199). Is two near-duplicated examples considered common, and one considered rare? The 38k seems to be a small portion and it's unclear if the notion of "common" vs. "rare" is significant enough to study.
- It is unclear what the common patterns found across counterfactually memorized text are (L50).
- Please use "counterfactual memorization" instead of simply "memorization", whenever proper, to avoid confusion.

Analyzing Counterfactual Memorization:

- There are contradicting claims. E.g. L107 says high (counterfactual) memorization happens for unconventional texts and multilingual texts, while L126 says ill-formatted text or foreign languages are hard and L125 says hard examples have low counterfactual memorization scores.
- There are non-refutable claims. E.g. L123 says easy examples tend to have low scores. "Simple samples" are defined as those with high accuracy, and two high accuracy scores should apparent have a relatively small difference.

From Memorization to Influence:

- Authors claim that small memorization scores have small max-influence scores and larger influence scores requires larger memorization scores (L253-254), but there doesn't seem to be a concentration of densities around the x=y line in Figure 4(b).

Some typos:

- L21: revious $\rightarrow$ Previous
- L38: considerded $\rightarrow$ considered
- L109: most often $\rightarrow$ are most often

**Questions:**

- L22-25: Are sensitive data, such as, phone numbers and usernames, "common" and frequently occurring?
- L30: How do you quantitatively define "common" and "rare"?
- L38-39: Why is the memorization of $x$ $\textit{counterfactual}$?
- L38-39: Could you clarify what "predicts $x$ accurately" means? It is implausible that given an initial token, an autoregressive language model will be able to generate an entire document $x$ perfectly via greedy decoding.
- L75: Does |D| refer to the number of documents?
- L93: But T5-base has ~220M parameters, instead of ~112M.
- There seems to be no control over the number of subsets S and the number of subsets S' being similar and both big. It is possible that a training sample appears in all 400 samples. Then the expected measure M is a bad approximation. What's a theoretical error bound?

Please also see Weaknesses. If the review misses important aspects, please feel free to address them.

**Limitations:**

The authors discussed limitations but not ethical concerns. Both are reasonable.

---

> ### Author Rebuttal · Authors · 2023-08-08
>
> Thank the reviewer for the helpful suggestions and detailed questions. It seems there is some confusion regarding the definition and analysis of counterfactual memorization. Due to the 6000-character limit, we focus on the main questions below. We hope the reviewer could increase the rating if the answers clarify those confusions. We are happy to answer any further questions during the discussion period if anything is still unclear.
>
> ---
>
> **Q1**: per-token top-1 accuracy is not sophisticated enough
>
> We used teacher-forcing when evaluating under per-token accuracy, so the evaluation is similar to perplexity in the sense that we are measuring the likelihood of predicting the next token accurately given the *true* context, instead of asking the model to construct the entire document accurately. We have evaluated the per-document perplexity to measure memorization, and the ranking of memorized examples is largely consistent with per-token accuracy. We ended up using the latter because it has a numerical range of [0, 1], which leads to scores that are more easily interpretable. We note that the intermediate-to-high memorized examples do end up containing details from news reports of specific events (Table 1 and Appendix L), and such memorized documents are found to have strong influence on the validation set documents describing related events (Table 2 and Appendix M).
>
> **Q2**: “expected” change (L32) inaccurate
>
> By “expected” change (L32) we meant “anticipated” change, instead of the formal “expectation” as in probability. We will rephrase it to avoid ambiguity.
>
>
> **Q3**: quantitative definition of common / rare is unclear; is 38k near-duplicates worth studying?
>
> The fundamental question is how to compare if two documents are “identical” or “encode the same information”. We can define common and rare by counting the occurrences via approximate textural matching, as previous work has done. In this paper, we try to quantify it by whether or not the document can be easily predicted by a language model if the model has not seen it during training.
>
> Section 4 is a study of how our metric relates to the number of near duplicates. Though there are only 38k examples that have near-duplicates, the number of duplicates goes up to hundreds for many examples. Moreover, the text-matching based near duplication detection does not capture all related documents. In Appendix D, we further studied the impact of data deduplication on memorization analysis, and found that related documents that are missed by text matching deduplication can still be identified by counterfactual memorization. Finally, we are more interested in rare examples than common ones due to the large impact on model predictions.
>
> **Q4**: common patterns (L50)?
>
> We found examples with the highest counterfactual memorization are generally unconventional text such as all-capital letters. After those artificial examples, we see news reports of specific events. Examples with low counterfactual memorization encode common information, many are templated documents.
>
> **Q5**: contradiction on difficulty and counterfactual memorization
>
> There is actually no contradiction here. The counterfactually memorized examples are mostly difficult, but the converse is not true. In fact, it is shown (L125, and Fig.1) that the **hardest** examples generally have low counterfactual memorization, because even when included in the training set, the models are having a hard time learning them. We will revise the text to make it more clear.
>
> **Q6**: no concentration of densities around x=y in Figure 4(b).
>
> Note the concentration won’t necessarily be on x=y, because the max-influence on a validation example does not necessarily equal the memorization (self-influence). Moreover, the main probability mass is on low memorization / low influence because the counterfactual memorization mostly captures examples in the tail. For those examples with large max-influence, we observe that memorization is also large. But the converse is not true, because the finite validation set might not contain a document that encodes matching information for each counterfactually memorized training document.
>
> **Q7**: Are sensitive data, such as phone numbers and usernames, "common" and frequently occurring?
>
> It depends on the context. Phone numbers of a customer service would be considered “common” as it is likely to be found in many places. Whereas personal phone numbers are generally not “common” unless it is leaked or intentionally posted online in many places.
>
>
> **Q8**: Why “counterfactual” memorization?
>
> Counterfactual here refers to the procedure to make such measurement: we estimate the model’s prediction change if a given training document is withheld from its training corpus.
>
> **Q9**: what "predicts x accurately" means? It is implausible for LM to generate an entire document perfectly.
>
> We measure the next token prediction accuracy. Following the convention in language model *evaluation*, we use teacher forcing, meaning that the prediction of the next token is always conditioned on the *true* context. So the model only needs to predict each of the next token accurately given the true context, instead of generating an entire document perfectly via decoding. Although language models are also capable of doing that for *some* training examples, as shown in previous studies of text-matching based memorization.
>
> **Q10**: T5-base has ~220M parameters
>
> We use the decoder-only architecture for the auto-regressive language modeling task, which has around half of the parameters.
>
> **Q11**: is it possible that a training sample appears in all 400 samples?
>
> When generating the subsets, each example is included with probability 0.25. The probability that none of the 400 samples contain a given example is therefore 0.75^400 which is in the order of 10^-50. So it is possible but extremely unlikely. More generally, this is a binomial distribution, and so has very small tails.

---

> > ### Comment · Reviewer_dGf9 · 2023-08-20
> >
> > Thank you for the reply, which clarified many of my doubts, and look forward to the progress during the rebuttal being added to the final paper.

---

### Official Review · Reviewer_PCjJ · 2023-07-06

**Soundness:** 3 good
**Presentation:** 3 good
**Contribution:** 3 good
**Rating:** 7
**Confidence:** 3

**Summary:**

This paper introduces a concept  "counterfactual memorization," defined as the anticipated change in a model's output when a specific training example is omitted. Experimental analysis was conducted on three widely employed text corpora in the domain of language modeling, and the phenomenon of memorization was identified in all instances. Furthermore, the authors assess the impact of each memorized training example on the validation set and generated texts. This novel approach offers direct evidence pinpointing the origin of memorization during testing. Despite these intriguing insights, the paper has notable limitations. The experiments performed lack robustness and some of the assertions made are lacking in clarity.

**Strengths:**

1.The paper well-written and  easy to follow.

2.It is of great significance to explore memorization in neural language models.

3.The proposed new definitions of can serve as references for future works, and the experimental analysis is detailed and interesting.

**Weaknesses:**

1.The paper lacks a section on related work, which should summarize and analyze the current state and challenges of relevant research.

2.This paper only uses the Transformer-based language backbone equivalent to T5-base (encoder-decoder framework). Can the experimental results be applied to other architectures and models of different scales?

3.Some typos need to be fixed. For example, “revious” (line 21) should be “Revious”. The word at the beginning of a paragraph is capitalized.

**Questions:**

1.Does the result of the model depend on the size of the subset ?  Is the result sensitive to the value of r? How did the author determine that r=0.25?

2.Some domains are represented much more frequently than others in the datasets we studied (Line 136). What are these domains, and what are their characteristics?

**Limitations:**

The authors have adequately addressed the limitations.

---

> ### Author Rebuttal · Authors · 2023-08-08
>
> Thank the reviewer for the positive review and helpful comments. Please see below for detailed clarifications.
>
> ---
>
> **Q1**: lacks a section on related work
>
> The “related work” section is put in Appendix A due to space constraint. We will rearrange the contents in the updated manuscript to make it more prominent.
>
> **Q2**: this paper only uses  T5 (encoder-decoder framework) backbone, how about other architectures and models of different scales?
>
> We actually used the decoder-only version of T5, which better matches the architecture used for training auto-regressive language models. We will emphasize this in the manuscript. We are very interested in exploring the scaling law of counterfactual memorization as we increase the model sizes exponentially. We leave it as future work as it is currently computationally very challenging to evaluate much larger models without non-trivial improvements to the estimation algorithms.
>
> **Q3**: Does the result of the model depend on the size of the subset ? Is the result sensitive to the value of r? How did the author determine that r=0.25?
>
> When r is too large (close to 1) or too small (close to 0), we would end up having a very unbalanced number of samples on the two terms of Eq (1), therefore inaccurate results. We chose r=0.25 by considering such balance factor, computational efficiency, and utility of models trained on the subset data. However, we did not “tune” this parameter very carefully because apart from very extreme values, the results are consistent within a large range of r values.
>
> **Q4**: Some domains are represented much more frequently than others in the datasets we studied (Line 136). What are these domains, and what are their characteristics?
>
> In Fig 2(a)(b), we plot the number of documents on the y-axis for each domain. We annotated a few representative domain names. For example, on RealNews, reuters.com has orders of magnitudes more documents than hotair.com, therefore is represented more frequently. Especially on the c4 dataset, we observe that domains with more documents tend to concentrate on intermediate memorization values, while smaller domains have a wide spread along the memorization spectrum.

---

> > ### Comment · Reviewer_PCjJ · 2023-08-21
> > **Official Comment by Reviewer PCjJ**
> >
> > Thank you for the clarification, and it has addressed many of my concerns. I would like to raise my score to 7. I eagerly anticipate seeing the updates from the rebuttal incorporated into the final paper.

---

### Official Review · Reviewer_9fjM · 2023-07-11

**Soundness:** 3 good
**Presentation:** 2 fair
**Contribution:** 4 excellent
**Rating:** 8
**Confidence:** 4

**Summary:**

This paper proposes a metric for whether a sequence is memorized based on the degree to which exposure during training increases the probability of producing the sample.

**Strengths:**

The proposal here is strong, and remedies a real problem in the memorization literature: a failure to consider the *inherent* predictability of any given sequence (a property which this paper calls "simplicity"). This is a significant gap that I'm happy to see work on.

The results cover basic questions such as the impact of the number of epochs. They also explain the reasoning behind choosing the number of model runs they apply their method on, and this analysis is sound.

I also like that they measured not only the impact of a given sample on the model's tendency to emit itself but also its influence on the model's tendency to emit a different sample.

**Weaknesses:**

As far as I can tell, it is hard to actually get a counterfactual metric for memorization of highly duplicated sequences because they are more likely to appear across different training subsamples. This metric inherently favors rare samples, because prior exposure to the duplicated samples will decrease the impact of each additional exposure.

This paper doesn't distinguish between "simplicity" due to duplication and simplicity due to inherently predictable sequences, like repetitions of a single character.

"the choice of data has a relatively minor effect on memorization" but the corpora used are all ones that avoid code and multilinguality, favoring natural English language text. It is not clear that this property would hold for significantly different corpora. The authors acknowledge this issue to some degree, pointing out that one of the largest differences between the datasets is likely due to the higher-quality texts in Wiki40B.

The authors only qualitatively check their conjecture that "if a validation set example x0 contains similar information, infl(x ) x0) could be large", but it should be fairly easy to quantify whether the similarity between any pair of samples increases the influence.

Minor:
- fair number of typos, please proofread before publication

**Questions:**

I'm curious about counterfactual memorization with respect to the training timeline. This paper uses training subsampling as a way of controlling whether a particular sequence has occurred, but I think it is worth asking whether that sequence was counterfactually memorized when the model actually encountered it for the first time, which may be at different points of training. Given a model trained on a known ordering of its corpus, we could consider the impact of a sample in the context of training by comparing checkpoints before and after exposure. Have you considered this question, explored to some degree by Biderman et al. (https://arxiv.org/pdf/2304.11158.pdf)? I would love to see results if so.

**Limitations:**

The corpora studied are all fairly similar (focusing on natural language English text), so the results are limited to those settings. The authors don't acknowledge this as completely as I'd like, although they do mention that higher quality text has different behavior than a corpus with malformatted text or multilingual contamination.

The authors also acknowledge that counterfactual memorization is inherently going to assign less memorization value to sequences with near-duplicates throughout the corpus, and describe this as a decision about what type of memorization they consider.

The authors also acknowledge that the models they are using are much smaller than the large language models of interest today.

---

> ### Author Rebuttal · Authors · 2023-08-08
>
> Thank the reviewer for the strong support of our paper and helpful suggestions!
>
> ---
>
> **Q1**: This paper doesn't distinguish between "simplicity" due to duplication and simplicity due to inherently predictable sequences, like repetitions of a single character.
>
> Thanks for the comments. We agree those are important questions, which we think can be formulated into two main questions: 1) how does the frequency of a phrase correlate with its simplicity; 2) how does the (distribution of) frequencies of internet based training corpus correlate with that of “real” natural languages. For 1), we agree that there are some simple patterns such as “repeated same characters” or “short sequence lengths” that tend to make a sequence easier, but that is not always the case. We did some preliminary studies of how simple metrics such as token diversity correlate with memorization, but we did not see significant patterns. In the end, we believe the frequency is probably a general indicator than the intrinsic structures for simplicity. For 2), we studied how deduplication impacts the measurement of memorization in Appendix D. We found that it is very hard to choose the appropriate threshold for text-matching based deduplication method, and the counterfactual memorization metric actually helps to identify near-duplicate documents that are exactly along the cutting threshold of edit distances. Furthermore, the ranking for high memorization examples remains consistent after deduplication.
>
> **Q2**: quantify whether the similarity between any pair of samples increases the influence.
>
> One key difference between counterfactual influence and similarity is that the former tries to capture rare memorization. For example, a sequence with a common fact could have high similarity with a validation set example. But because it is a common fact, the validation set example receives influence from many training examples containing this fact, thus the influence from one particular training example could be very diluted and end up being small.
>
> **Q3**: counterfactual memorization with respect to the training timeline?
>
> Thanks for the pointer to Biderman et al. We will include a discussion of it in the updated manuscript. One main challenge we run into here is that neural network training has large intrinsic stochasticity. Even if we retrain the network on exactly the same training set again, the prediction could end up being quite different. To increase the signal to noise ratio for stable measurement of counterfactual memorization, we train hundreds of models to average out the stochasticity (we included a study of how many models are needed in Fig. 3a). For this reason, it is very challenging to control the noise in the measurement using the training trajectory of a single model. Maybe there is a way to perform a similar averaging procedure if we consider multiple model training trajectories with different randomly shuffled data order. In the early stages of our experiments, for implementation efficiency, instead of sampling data uniformly randomly, we tried to partition the data into chunks and sampling random chunks instead. The measurement ended up capturing the artifacts of co-occurrence of documents in the same chunk. So we suspect to successfully make such measurements along the training timeline, we will need very precise control over the data orders in the data loading pipeline.
>
> **Q4**: The corpora studied are all fairly similar (focusing on natural language English text), so the results are limited to those settings.
>
> Thanks for pointing this out! We will revise the limitation accordingly.

---

> > ### Comment · Reviewer_9fjM · 2023-08-21
> > **Thanks for the response**
> >
> > I’ve engaged with another reviewer elsewhere here, but I think you’ve addressed my own comments.

---

### Official Review · Reviewer_mGti · 2023-07-20

**Soundness:** 4 excellent
**Presentation:** 4 excellent
**Contribution:** 3 good
**Rating:** 6
**Confidence:** 4

**Summary:**

This work studies memorization in neural language models. The major scientific question in this work is how to filter out common memorization in language models. To this end, this work first formulates a notion of counterfactual memorization which
characterizes how a model’s predictions change if a particular document is omitted during training. Then, this work estimates counterfactual memorization on three standard text datasets, and confirms that rare memorized examples exist in all of them. In addition, this work also identifies an inverse correlation between number of duplicates and counterfactual memorization as compared with previous definitions of memorization. Finally, this work extends the definition of counterfactual memorization to counterfactual influence, and studies the impact of memorized examples on the test-time prediction of the validation set examples and generated examples.

Overall, this paper is well-written and flows smoothly. This work begins with a detailed definition of counterfactual memorization and then estimates and analyzes counterfactual memorization of training examples in standard datasets. The experiments identify the examples sampled at high, intermediate and low memorization. Analysis covers the impact of number of models, number of training epochs. This work also shows the potential for inference attacks in large language models in a learning theoretical perspective.

**Strengths:**

1. This is an interesting paper that analyzes the counterfactual memorization in neural language models. The paper is written in a clear and concise manner. The problem is well defined, and the analysis is technically sound with insightful findings.

2. This work extends the definition of counterfactual memorization to counterfactual influence, and studies the impact of memorized examples on the test-time prediction of the validation set examples and generated examples --- thus has the potential to provide new perspectives of studying the inference attack problem in large language models.

**Weaknesses:**

1. The efficiency of the proposed approach can be improved for a practical use. Currently, identifying the memorization levels of the training examples, as well as the counterfactual influence, requires extensive computation overheads by training multiple models --- which may become a major concern for its application in large language models and with larger scale training sets.

2. The experiments can be also broaden to leveraging different backbone language models to provide more useful insights towards how existing language models vary in the effects of counterfactual memorization. This may lead to an intuition on how those language models would be sensitive to inference attacks and stimulates the development of defense approaches.

**Questions:**

Line 21: "revious" -> "previous"

**Limitations:**

The authors have provided detailed and useful descriptions about the limitation.

---

> ### Author Rebuttal · Authors · 2023-08-08
>
> Thank the reviewer for the positive review and useful feedback!
>
> ---
>
> **Q1**: computation cost may become a major concern for its application in large language models and with larger scale training sets
>
> We acknowledge this limitation of our work. Our main focus for future work along this line is to improve the computational cost of estimation.
>
> ---
>
> **Q2**: The experiments can be also broaden to leveraging different backbone language models to provide more useful insights towards how existing language models vary in the effects of counterfactual memorization. This may lead to an intuition on how those language models would be sensitive to inference attacks and stimulates the development of defense approaches.
>
> Thanks for the suggestion! We used decoder-only transformer based models which are commonly adopted for language models. It is indeed quite interesting to try out alternative architectures. We believe that the results will not be that different among the different variants of transformer-based architectures that are currently used in practice. However, we are very interested in studying the “scaling law” behaviors as we increase the models and data sizes. Though that depends on the future work of more efficient estimation methods.

---

> > ### Comment · Reviewer_mGti · 2023-08-18
> >
> > Thanks for the response!

---

### Official Review · Reviewer_AGTS · 2023-07-20

**Soundness:** 3 good
**Presentation:** 3 good
**Contribution:** 3 good
**Rating:** 6
**Confidence:** 5

**Summary:**

The study proposes two novel metrics: “counterfactual memorization” and “counterfactual influence”, the first one can measure how a model’s predictions change if a sample is omitted in training, and the later one measure how a sample in the training set influences the prediction of a validation sample. The authors also analyze the two metrics on 3 standard datasets and discover some interesting results, like “different data sources display different memorization profiles”, ”number of models needs”, “memorization does quantitatively decrease when data is repeated more often”,  the strong influencer ”also only receive tiny influence from all the rest of the training examples”.

**Strengths:**

1.	Great motivation and important research question, how to measure a sample’s influence on the trained model’s prediction and the sample’s influence on a validation sample.
2.	The counterfactual viewpoint is interesting and easy to understand. “counterfactual memorization” and “counterfactual influence” are useful metrics.
3.	The analysis and some findings are insightful for inspiring future study.
4.	The presentation is clear and easy to follow.


**Weaknesses:**

1.	I am surprise that the authors do not refer the causal inferences studies, since “counterfactual” is an important concept in statistical causality. Judea Pearl, Donald Rubin and many scholars did many great work about counterfactual and causal effect, such as structural causal model, potential outcome model [2]. From the perspective of causality, the equation 1 and 3 could be reformulated by causal effect, counterfactual inference, which are important research topic and have been studies well in statistical causality communities. So I suggest the authors can study some classical causality studies [3][4], which can help you to make the research better.
2.	I am curious the proposed two metrics on simple models, such logistic regression, SVM. Large language model (LLM) is popular and important, while the simple models are also useful and efficient for many applications. In the meantime, I think the simple models, like logistic regression, are much easier to measuring the two metrics, since the LLM is complex and have more unobserved confounders. So my second suggestion is that the authors could try some simple models.
3.	It is much clearer to summarize your analysis and findings in one table.
4.	Typos: e.g. revious in line 21.
[1] Causal inference in statistics: An overview, J Pearl
[2] Causal Inference Using Potential Outcomes, DB Rubin,2005
[3] Causality: Models, Reasoning and Inference, J Pearl 2009
[4] Causal Inference for Statistics, Social, and Biomedical Sciences, G.W. Imbens, 2015


**Questions:**

1. The authors could study some classical causality studies [3][4], which can help you to make the research better;
2. The authors could try some simple models and do the similar analysis.


**Limitations:**

Besides the limitations the authors list in section 7, the authors also mentions that it is hard to evaluate the impact of a single training example for the costly computation, I hope the authors could compare their methods with influence function in [5], which can identify training sample most responsible for a given prediction, and discuss their advantages and disadvantages.

---

> ### Author Rebuttal · Authors · 2023-08-08
>
> Thank the reviewer for the positive review and useful suggestions!
>
> ---
>
> **Q1**: pointers to classical causality studies
>
> Thanks for the pointer! We will revise the manuscript to add missing references to the classical causality studies and discuss the connections. Our formulation does not involve complex graphical models and intervention techniques from statistical causality. We are interested in how removal of a training document impacts the model’s prediction, and formulated it as a *direct measurement*. It would be interesting future work to see if any advanced techniques in causality studies could be used to improve our measurement.
>
> ---
>
> **Q2**: counterfactual memorization analysis on simple models
>
> Since we focused on language tasks in this paper, we did not try simpler models as they generally do not fit well to the sequence-prediction task. However, we agree that simpler models in general could be quite interesting, and there may be specialized algorithms that can estimate memorization efficiently for specific simple models. For example, for SVMs, if a training example is not a support vector, then removing it from the training set will have zero impact on the model’s decision boundary. Similarly, for k-nearest-neighbor classifiers, the removal of a training example will have very localized impact on the model’s prediction. For linear regression models, since the optimal weights have a closed form solution, we can also calculate the memorization more easily.
>
> ---
>
> **Q3**: It is much clearer to summarize your analysis and findings in one table.
>
> Thanks for the suggestion! We will update the manuscript and try to format the summary of findings into a table.
>
> ---
>
> **Q4**: the advantage and disadvantage comparing to influence function [5]
>
> It seems that the reviewer forgot to list the citation [5]. We assume the reviewer meant the following [5]. We note that the estimation in [5] relies on an approximation using the inverse of Hessian, which is computationally prohibitive to calculate for large models. As a result, both methods are computationally expensive, but the cost of [5] scales with the model size, and our method scales with the cost of training multiple models, which can be easily parallelized. Moreover, while the approximation used in [5] works well for simple linear models, it is shown to be very fragile in deep neural networks (e.g.[6]). We will include discussions on this in the updated manuscript.
>
> - [5] Koh, P. W., & Liang, P. (2017, July). Understanding black-box predictions via influence functions. In ICML.
> - [6] Basu, S., Pope, P., & Feizi, S. (2021). Influence functions in deep learning are fragile. ICLR.

---

> > ### Comment · Reviewer_AGTS · 2023-08-15
> >
> > Thank you for your clarifications, especially for the analysis and discussion about influence function.

---

### Official Review · Reviewer_PFxH · 2023-07-21

**Soundness:** 2 fair
**Presentation:** 2 fair
**Contribution:** 2 fair
**Rating:** 6
**Confidence:** 4

**Summary:**

This paper formulates a notion of counterfactual memorization to measure the "one-point" generalization performance of the model (they do not convey this definition as this). Equipped with this definition, the authors conduct plenty of experiments to explore the components that are related to such counterfactual memorization of large language models.

**Strengths:**

This paper is well-written, and the analysis of what kind of samples tend to be memorized is interesting. The experiments in this paper are systematic as many of the related components are considered.

**Weaknesses:**

Despite the strengths mentioned above. I think this paper has the following weakness.

1) The definition of counterfactual memorization is nothing but the generalization performance of the model on one single point. In fact, taking expectation to $x$, the counterfactual memorization becomes expected generalization. Thus, in my opinion, studying it from the perspective of distribution is exactly studying the generalization of LLM, which obviously is not a new topic.

2) The definition of counterfactual memorization is quite similar to some classical notations in generalization theory, e.g., algorithmic stability and $\epsilon$-differential privacy.

3) Here is my major concern. The performance of the model is evaluated by the 0-1 prediction loss on the task of next-word prediction (NWP). However, I think this is not a reasonable metric to measure the performance of NWP. The metric is reasonable for the classification task. However, for the NWP task, I do not think there exists a "ground-truth" label as in classification. For example, the sequence "I like eating _" followed by the words "apple" or "banana" are all valid predictions.



**Questions:**

As the authors clarified, "Both the neural language models and training sets used in this work are orders of magnitude smaller than modern standards such as GPT-3". Though conducting experiments on these LLM is impossible. My question is, does the samples that are proven to be easily memorized can not have diverse patterns during inference, even for larger models?

**Limitations:**

As the authors clarified in their paper, they have no experiments on larger-scale pre-trained model.

---

> ### Author Rebuttal · Authors · 2023-08-08
>
> Thank the reviewer for the comments and suggestions! It seems there are some misunderstandings of how our metrics relate to existing notions of generalization and how the 0-1 prediction loss is chosen. We answer those questions below and hope the reviewer could raise the rating if the answers clarify the confusion. We are happy to answer any further questions during the discussion period.
>
> ---
>
> **Q1**: relationship between counterfactual memorization and generalization.
>
> We appreciate the reviewer for observing the connection with the generalization gap in learning theory. This connection actually provides another strong motivation to the definition of counterfactual memorization. As noted by the reviewer, the standard generalization gap measures the *expected* difference of model performance on unseen and seen examples. When this gap is large, the model is said to overfit, or memorize the training set without generalization capability. By making the measurement at the example level (and averaging over multiple models, instead of averaging over test examples), we measure the same “generalization gap”, but now we can characterize whether a particular example x is memorized or not. This key difference allows us to do all the analysis in this paper, and enables the extension to measure counterfactual influence. Therefore, we respectfully disagree with the reviewer that this is identical to measuring the generalization.
>
> ---
>
> **Q2**: relationship between counterfactual memorization and algorithmic stability / $\varepsilon$-differential privacy
>
> We agree that counterfactual memorization is related to many important notions, which actually supports that our metrics are grounded on the same fundamental notion of generalization-memorization trade-off widely studied in the field.
>
> We also clarify the key differences here: the relation to algorithmic stability is similar to the relation to generalization gap as explained in the previous question. We focus on characterizing the behavior of each single example, while stability (e.g. leave-one-out stability) is formulated as a property of a learning algorithm. Moreover, while our analysis is heavily dependent on the underlying data, the study of stability generally consider *uniform* stability, which is a property of the learning algorithm that is supposed to hold for *arbitrary* data (i.e. data independent).
>
> The relation to differential privacy (DP) is similar: DP quantifies the *worst case* changes under arbitrary replacement of an example, which is a property of the learning algorithm and independent of the data (DP guarantee should hold for arbitrary data). While we focus on a characterization of each example (as opposed to the algorithm), and measure how each training example influences the trained model in relation to the other training examples, which is heavily dependent on the underlying data distribution. On the empirical side, it is known that outlier examples (which can be identified via high counterfactual memorization scores) tend to have high membership inference attack success rate, and membership inference attacks can be used to provide empirical lower bound for the DP privacy parameter.
>
> We also note that DP and stability are closely related to each other in the formulation, but it does not decrease the importance of either notion because they have very different perspectives.
>
> ---
>
> **Q3**: justification of 0-1 loss in next word prediction task
>
> It is true that for short contexts, many possible “next words” exist in the training data. But the distribution of possible next words becomes sharp as the context length increases. Moreover, such kind of ambiguity also exists in image classification. For example, many ImageNet images are real world scenes containing multiple different kinds of objects, yet the model is trained to maximize the likelihood of a single labeled class and evaluated with 0-1 loss because it is simple and effective. In language model, such ambiguity is also not *explicitly* addressed, the training objective is exactly the same (cross entropy loss) at each word (token) level. The model implicitly learns about the relation between similar words. When evaluating a language model, it is common to use perplexity, which boils down to measuring the per-token likelihood on the exact groundtruth token (again, without explicitly considering other possible words that are equally valid). We have evaluated measuring memorization using perplexity, and the ranking of memorized examples are largely consistent with the per-token accuracy measurements. However, we ended up using the latter because the per-token accuracy has a well defined numerical range of [0, 1], which leads to score values that are easier to interpret.
>
> ---
>
> **Q4**:  "does the samples that are proven to be easily memorized can not have diverse patterns during inference, even for larger models?"
>
> Could the reviewer clarify the question? What do you mean by having “diverse patterns during inference”?
>
> While we cannot make formal guarantees that all the results transfer to LLMs at GPT-3 or larger models, we believe most of the observations would still hold. In particular, manual inspection of the memorized examples (see the Appendix for a sample of them) matches the intuition, and we believe larger models will behave similarly. Our future work focuses on finding approximate solutions to the algorithm so that it can be applied to the larger models.

---

> > ### Comment · Reviewer_PFxH · 2023-08-14
> > **Response**
> >
> > Thanks for the authors' responses. The response addresses some of my concerns. But my major concern "The performance of the model is evaluated by the 0-1 prediction loss on the task of next-word prediction (NWP)" does not been addressed.
> >
> > The authors admit that there exist many possible words in the NWP task, and they said ambiguity exists in the image classification task as well. However, the main difference between the image classification task and the NWP task is that during the training and inference stage, "underlying ground truth" labels do not exist in the training stage. For example, suppose we have an image classification model which is trained to recognize the species of animal, we will not expect it to recognize the background of images, as the background labels do not involved in the training stage. However, for the NWP task, all possible words exist in the training stage (with the format of a vocabulary list), thus it is improper to compare image classification and the NWP task.
> >
> > Back to the possible words situation, I insist on my opinion that for the NWP task, we should recognize it as a generative task instead of a classification task. For example, if the sentences "I like eating apples" and "I like eating bananas" are existed in the training set. Obviously, the optimal solution of ERM objective predicts "apples" and "bananas" respectively with a probability of $0.5$, under the condition of "I like eating _" (this makes training loss become zero). Since this paper explores memorization in a theoretical manner, I highly suggest the authors consider this situation (I think commonly existed).
> >
> > By the way, I note a highly related paper https://arxiv.org/pdf/2308.03296.pdf recently published, which studies the memorization paper as in this paper. This paper studies memorization at the distribution level. The opinion conveyed in the reference seems to be opposite of this paper. I directly copy it on page 15 of the reference as follows.
> >
> > "Second, while the true training tokens are used as the inputs to the network, the “labels” for the pseudo-gradient calculation are sampled from $P(yˆ|x)$. While it may appear odd for the labels not to match the inputs in an autoregressive setting, "

---

> > > ### Comment · Reviewer_9fjM · 2023-08-14
> > > **0-1 accuracy, another reviewer's perspective**
> > >
> > > While I agree with you that perplexity presents a more complete portrait of learning, I think that it's worth noting that the influence functions paper does not claim to focus on memorization, but on generalization. In fact, I would argue that relying on perplexity risks conflating the two concepts, as it discards the question of whether the model has memorized, rather than simply learning, a datapoint. This points to a problem in the memorization literature broadly, which repeatedly fails to define memorization as a phenomenon distinct from learning and even from generalization. This paper is at least an attempt to add nuance to the current notion of memorization, which is often based on accuracy as here, on k-elicitation, or on some partial metric based on k elicitation at best. As such, I believe that it has significant value for the *memorization* literature specifically, beyond the general learning framed by the influence functions paper.

---

> > > ### Author Response · Authors · 2023-08-14
> > > **Further clarification on 0-1 prediction loss**
> > >
> > > Thanks very much for the prompt response!
> > >
> > > **Image classification**: when we say the same ambiguity exists in image classification, we meant that there could be multiple **foreground objects** in the same image, and the same image can actually be predicted as multiple labels "correctly". There are many examples of such images exist in standard image classification training sets. While we cannot post link directly, one example we could point the reviewer to is Fig. 5 of [1]. For example, under the "Mountain Bike" category, there is an image of a cat behind the bike wheel, and another image with a truck loading lots of bikes. Under such inputs / contexts, both labels are equally likely or correct. In the terminology of NLP, all possible classes (bike, cat, truck) exist in the same vocabulary and are involved in the training stage.
> > >
> > > **Common practice in the field**: We acknowledge such ambiguity exist (in both image classification and language modeling), and we do not claim comparing with the groundtruth label is the best way to handle it. However, this is a simple and effective metric that is widely adopted in both image classification (top-1 accuracy) and language modeling (validation perplexity computed by the likelihood on the groundtruth next token). We follow this convention in our study here. Moreover, as reviewer 9fjM (Thank you!) mentioned, we focus on study of memorization in this paper, previous studies in language model memorization actually mostly rely on text matching, and our study is an important step towards distinguishing memorization and generalization.
> > >
> > > **Opposite opinions**?: Thank the reviewer for a pointer to the interesting paper studying LLM generalization with influence function. Regarding the quote provided by the reviewer (emphasis added by us):
> > >
> > > > Second, while the true training tokens are used as the inputs to the network, the “labels” for the pseudo-gradient calculation are
> > > sampled from $P(\hat{y}|x)$. While it may appear odd for the labels not to match the inputs in an autoregressive setting, this is indeed the correct **sampling procedure when the goal is to approximate** $\mathbf{G}$.
> > >
> > > We note that this paper suggested such operation specifically for *the purpose of approximating* $\mathbf{G}$, which is defined as an expected value and approximated via such a *sampling procedure*. This is not directly arguing against using the groundtruth next-token as a *performance evaluation criterion*. Therefore, we do see contradiction in the messages conveyed by the two papers.
> > >
> > > **Study on generated texts**: Finally, while not a main focus of this paper, we did study the counterfactual influence on model generated text in the last paragraph of section 5 (L279). We hope this study and the clarification above address the reviewer's concern.
> > >
> > >
> > > [1] Siddiqui, S. A., Rajkumar, N., Maharaj, T., Krueger, D., & Hooker, S. (2022). Metadata archaeology: Unearthing data subsets by leveraging training dynamics. arXiv preprint arXiv:2209.10015.

---

> > > > ### Comment · Reviewer_PFxH · 2023-08-15
> > > > **Thanks for your response**
> > > >
> > > > Thanks for the author's response.
> > > >
> > > > About the image classification ambiguity, if there exist images with both cats and bikes in it. Suppose half of these images are labeled with cat and the others are labeled with bikes. Then I think the classification model will output logits with 50% weight on the cat and 50% weight on the bikes when the input is one of these images. Clearly, this example is similar to the one in the language model.
> > > >
> > > > For the definition of memorization, I agree with the proposed memorization in this paper explains some of it. But my point is that such criteria built upon 0-1 loss may not cover all possible situations in the language model. A simple example is when we say memorization, we intuitively think the language model memorize all possible combination in the training set. For example, if there are only  "I like eating apples" and "I like eating bananas" in the training set. If the language model totally memorizes all training sets. Then it will only predict "apples" or "bananas" within the condition "I like eating". Clearly, this is intuitively a memorization. However, when measured by the 0-1 loss, none of the training data "I like eating apples" nor "I like eating bananas" are memorized. Because the prediction probability on "apples" will vary from 0 to 0.5 (the probability can be arbitrarily small by increasing the categories of fruits) when  "I like eating apples" appears in the training set.
> > > >
> > > > By the way, I think memorization should be measured under the sufficiently trained model. For the above example, if the model predicts "apples" with probability 1, under the condition "I like eating". Then, I think this model is not sufficiently trained as it does not approximate the ERM.

---

> > > > > ### Author Response · Authors · 2023-08-15
> > > > > **Clarification on types of Memorization being studied**
> > > > >
> > > > > Thanks for the reviewer's clarification. We now get a better understanding of where the disagreement lies --- it seems the reviewer has a different interpretation of *memorization* from ours in the paper. The term "memorization" is very intuitive yet result in many *different* formal definitions when put into the contexts.
> > > > >
> > > > > It seems the reviewer is focusing on the situation where a LM **fully memorized** the training set, or, when the LM fully capture the probability distribution of the underlying training data. We agree this is an interesting and important topic of study. However, as far as we know, there is no clear evidence even the largest LMs nowadays have achieved this goal of *fully* memorizing the training set (i.e. achieved the Bayes optimal model). Even if we want to study this, it would be quite difficult to approach.
> > > > >
> > > > > On the other hand, most study of LM memorization (including ours) focus on the case where a LMs have not *fully* memorized the training set. In this case, one of the main question is which examples are memorized and which are not. In other words, the main subject is not how *the model* behaves when it memorized the training set, but which *training examples* are memorized by a model when trained on them.
> > > > >
> > > > > Our paper goes one step further with a goal of trying to distinguish among the memorized examples, which one are counterfactually memorized, which one are memorized via "generalization". The reviewer's example of "I like eating _" falls into the category of memorization via "generalization", because it is a generalizable, distributional wise property of the (English) language itself. Our counterfactual memorization focuses on a different kind of memorized examples that could potentially be important in the contexts of privacy and copyright. For example, if the prefix is "My social security number is _", then it is not very concerning if the LM "fully memorized" all the training examples and learns the "general rule" that the completion should be XXX-XX-XXXX where "X" could be any digits. However, it could be quite concerning if my particular SSN combination is memorized by the model and generated by the model. Our paper tries to capture this type of memorization.
> > > > >
> > > > > **Summary**: the reviewer's suggestion seems very valuable in the study of model behavior under **full memorization**. However, our study is in a **different regime** (following previous studies of LM memorization) and focus on identifying which examples are (counterfactually) memorized and which are not. In this regime, the measurement by 0-1 loss is reasonable. Please let us know if this clarifies the disagreement. Thanks again for the suggestions and engaging discussions!

---

> > > > > > ### Comment · Reviewer_PFxH · 2023-08-16
> > > > > > **Reponse**
> > > > > >
> > > > > > Yes, I think this clarifies the disagreement. For LLM, I agree with you that the 0-1 metric is suitable to measure memorization when considering copyright or privacy. I suggest the author clarify such differences at the beginning of this paper. I think my concern is addressed, and raise my score to 6.

---

> > > > > > > ### Author Response · Authors · 2023-08-16
> > > > > > > **Thank you very much**
> > > > > > >
> > > > > > > We really appreciate the time you put into the review, suggestions and discussions! Thank you for raising the score! We will make sure to make the introduction clear on the specific types of memorization we study in the paper.

---

### Official Review · Reviewer_DH4T · 2023-07-21

**Soundness:** 4 excellent
**Presentation:** 4 excellent
**Contribution:** 3 good
**Rating:** 8
**Confidence:** 3

**Summary:**

The paper studies an important problem of the "memorizing" effect in neural language models, particularly focusing on rare or isolated pieces of information in the training data that get memorized, as opposed to widely duplicated or common information.
It formulates a notion of "counterfactual memorization" that measures how a model's predictions change when a particular training example is excluded during training. It analyzes counterfactual memorization on several standard text datasets and studies patterns in what gets memorized. Examples with unconventional text formats tend to have high memorization.
It extends the definition to "counterfactual influence" which quantifies the impact of a memorized training example on the inference phase, which helps trace the source of predictions.
Overall, the paper provides a novel perspective and tools to systematically study the memorization of information in neural language models.

**Strengths:**

This paper presents a novel and significant perspective on studying information flow in neural language models.
The notion of counterfactual memorization is an original contribution to quantifying the memorization of isolated pieces of information. This moves beyond prior work that focused more on duplicated content. Extending this to counterfactual influence to trace predictions back to training examples provides a new capability.
The mathematical formulation of counterfactual memorization is clear and well-motivated.
The empirical methodology is sound, with adequate models trained to converge and sensitivity analysis on the number of models needed.
The results support the claims and provide new insights into memorization.
Overall, I find this a very solid, insightful paper with multiple novel contributions to an important problem.

**Weaknesses:**

One potential weakness of this paper is the application of such analysis remains elusive. I can imagine many interesting questions could be answered with the provided concept, e.g., what kind of stereotypes/social biases in the model are influenced by what data in the training samples?

**Questions:**

The core idea of counterfactual memorization is intuitive, but can you provide some theoretical justification for why the formulation in Equation 1 captures the memorization of information? Is there any formal argument that you can make?

**Limitations:**

Yes.

---

> ### Author Rebuttal · Authors · 2023-08-08
>
> Thank the reviewer for the strong support of our paper and useful suggestions!
>
> > **Q1**: One potential weakness of this paper is the application of such analysis remains elusive. I can imagine many interesting questions could be answered with the provided concept, e.g., what kind of stereotypes/social biases in the model are influenced by what data in the training samples?
>
> Thanks for the suggestion. We are mostly thinking about prediction attribution (finding which training examples contribute to the current prediction) as a general direction which may have many downstream applications. We agree that understanding the source of stereotypes / social biases seems to be a very interesting concrete question to consider.
>
> > **Q2**: The core idea of counterfactual memorization is intuitive, but can you provide some theoretical justification for why the formulation in Equation 1 captures the memorization of information? Is there any formal argument that you can make?
>
> One potential connection to learning theory is that Eq 1 essentially measures the "generalization gap", except that it is measured on a specific given data point, as opposed to the expectation on the (unknown) underlying data distribution in standard learning theory. The latter being large means that a model has overfitted, thus memorizing the training data without generalization capability. In our case, we modified the notion to measure the "generalization gap" of a specific example x, in order to quantify whether a particular example x is memorized. This formulation has been theoretically studied in a synthetic model, where it is proven that memorization is closely related to optimal generalization if the underlying data distribution is long tailed.
>
> [1] Feldman V. Does learning require memorization? a short tale about a long tail. InProceedings of the 52nd Annual ACM SIGACT Symposium on Theory of Computing 2020 Jun 22 (pp. 954-959).

---

> > ### Comment · Reviewer_DH4T · 2023-08-14
> >
> > Thanks for the clarification!

---

### Author Rebuttal · Authors · 2023-08-08

Thank all the reviewers for their useful comments and suggestions. We appreciate the reviewers found that our paper is clearly written, provides “strong proposal”, “sound analysis”, “useful metrics”, “novel perspective and tools”, “insightful findings” to the study of language model memorization.

We will fix the typos and minor issues pointed by the reviewers in the updated manuscript. The main questions from each reviewer will be clarified in the individual replies below.

---

### Decision · Program_Chairs · 2023-09-21

**Decision:**

Accept (spotlight)

**Comment:**

This paper initiates and establishes a quantitative framework to study the counterfactual memorization  problem of LLMs. It receives unanimous positive reviews and rebuttal has resolved some misunderstandings. The reviewers think the problem is well motivated, timely, important, and the findings are insightful and practical. This work can be strengthened by extending the studies and analysis on larger-scale pre-trained models, and elaborating on the connection between memorization and generalization. AC agrees this work is a solid contribution on the study of memorization of LLMs, and thus recommends acceptance as a spotlight.